# The pulse of a montane ecosystem: coupling between daily cycles in solar flux, snowmelt, transpiration, groundwater, and streamflow at Sagehen and Independence Creeks, Sierra Nevada, USA

James W. Kirchner[1,2,3], Sarah E. Godsey[1,4], Madeline Solomon[5], Randall Osterhuber[6], Joseph R. McConnell[7], and Daniele Penna[2,8]

[1]Department of Earth and Planetary Science, University of California, Berkeley, California, USA
[2]Dept. of Environmental Systems Science, ETH Zurich, Zurich, Switzerland
[3]Swiss Federal Research Institute WSL, Birmensdorf, Switzerland
[4]Department of Geosciences, Idaho State University, Pocatello, Idaho, USA
[5]Department of Geography, University of California, Berkeley, California, USA
[6]Central Sierra Snow Laboratory, Norden, California, USA
[7]Division of Hydrological Sciences, Desert Research Institute, Reno, Nevada, USA
[8]Department of Agriculture, Food, Environment and Forestry, University of Florence, Florence, Italy

*Correspondence to*: James Kirchner (kirchner@ethz.ch)

**Abstract.** Water levels in streams and aquifers often exhibit daily cycles during rainless periods, reflecting daytime extraction of shallow groundwater by evapotranspiration (ET) and, during snowmelt, daytime additions of meltwater. These cycles can aid in understanding the mechanisms that couple solar forcing of ET and snowmelt to changes in streamflow. Here we analyze three years of 30-minute solar flux, sap flow, stream stage, and groundwater level measurements at Sagehen Creek and Independence Creek, two snow-dominated headwater catchments in California's Sierra Nevada mountains. Despite their sharply contrasting geological settings (most of the Independence basin is glacially scoured granodiorite, whereas Sagehen is underlain by hundreds of meters of volcanic and volcaniclastic deposits that host an extensive groundwater aquifer), both streams respond similarly to snowmelt and ET forcing. During snow-free summer periods, daily cycles in solar flux are tightly correlated with variations in sap flow, and with the rates of water level rise and fall in streams and riparian aquifers. During these periods, stream stages and riparian groundwater levels decline during the day and rebound at night. These cycles are reversed during snowmelt, with stream stages and riparian groundwater levels rising during the day in response to snowmelt inputs, and falling at night as the riparian aquifer drains.

Streamflow and groundwater maxima and minima (during snowmelt- and ET-dominated periods, respectively) lag the mid-day peak in solar flux by several hours. A simple conceptual model explains this lag: streamflows depend on riparian aquifer water levels, which integrate snowmelt inputs and ET losses over time, and thus will be phase-shifted relative to the peaks in snowmelt and evapotranspiration rates. Thus, although the lag between solar forcing and water level cycles is often interpreted as a travel-time lag, our analysis shows that it is mostly a dynamical phase lag, at least in small catchments. Furthermore, although daily cycles in streamflow have often been used to estimate ET fluxes, our simple conceptual model demonstrates that this is infeasible unless the response time of the riparian aquifer can be determined.

As the snowmelt season progresses, snowmelt forcing of groundwater and streamflow weakens and evapotranspiration forcing strengthens. The relative dominance of snowmelt vs. ET can be quantified by the diel cycle index, which measures the correlation between the solar flux and the rate of rise or fall in streamflow or groundwater. When the snowpack melts out at an individual location, the local groundwater shifts abruptly from snowmelt-dominated cycles to ET-dominated cycles. Melt-out, and the corresponding shift in the diel cycle index, occur earlier at lower altitudes and on south-facing slopes, and streamflow integrates these transitions over the drainage network. Thus the diel cycle index in streamflow shifts gradually, beginning when the snowpack melts out near the gauging station, and ending, months later, when the snowpack melts out at

the top of the basin and the entire drainage network becomes dominated by ET cycles. During this long transition, snowmelt signals generated in the upper basin are gradually overprinted by ET signals generated lower down in the basin.


The gradual springtime transition in the diel cycle index is mirrored in sequences of Landsat images showing the springtime retreat of the snowpack to higher elevations, and the corresponding advance of photosynthetic activity across the basin. Trends in the catchment-averaged MODIS enhanced vegetation index (EVI2) also correlate closely with the late springtime shift from snowmelt to ET cycles and with the autumn shift back toward snowmelt cycles. Seasonal changes in streamflow

cycles therefore reflect catchment-scale shifts in snowpack and vegetation activity that can be seen from Earth orbit. The data and analyses presented here illustrate how streams can act as mirrors of the landscape, integrating physical and ecohydrological signals across their contributing drainage networks.

Key words: diel cycle; diurnal cycle; streamflow; groundwater; snowmelt; evapotranspiration; sap flow; solar radiation; phase lag.

**1 Introduction**

In mountain regions, streamflow and shallow groundwater levels often exhibit 24-hour cycles driven by either snow/ice melt or evapotranspiration. Both snowmelt and evapotranspiration cycles result from daily variations in solar flux, but are of

opposite phase (Lundquist and Cayan, 2002; Mutzner et al., 2015; Woelber et al., 2018), because melt processes contribute water to the shallow subsurface during daytime, while evapotranspiration removes it during daytime. These daily cycles have been used to investigate streamflow generation and runoff routing (Wondzell et al., 2007; Barnard et al., 2010; Woelber et al., 2018), to infer dominant processes affecting catchment water balances (Lundquist and Cayan, 2002; Czikowsky and Fitzjarrald, 2004), and to estimate temporal patterns of landscape-scale evapotranspiration (ET) and precipitation rates (Bond

et al., 2002; Kirchner, 2009; Cadol et al., 2012). The analysis of daily cycles may thus be a useful diagnostic tool in catchment hydrology, helping to characterize eco-hydrological processes at the catchment scale (Lundquist et al., 2005; Gribovszki et al., 2010).

However, in many cases it remains unclear how daily cycles in groundwater and streamflow should be quantitatively linked

to daily cycles of snowmelt and ET fluxes. How are the amplitudes or phases of groundwater cycles related to the amplitudes and phases of the snowmelt and ET cycles that drive them? How are these groundwater cycles transmitted to streamflow, and how are streamflow cycles integrated along the channel network? While these linkages have been modeled (both conceptually and numerically) based on various mechanistic assumptions (as reviewed by Gribovszki et al., 2010), empirical verification remains sparse due to the scarcity of coupled observations of snow accumulation and melt, daily ET

cycles, and fluctuations in both groundwater and streamflow at multiple locations along channel networks.

Daily groundwater cycles have been widely used to infer riparian evapotranspiration rates using various forms of a groundwater mass balance first proposed by White (1932):

$$E_G = S_y (24r + s) \tag{1}$$

where $E_G$ is the consumption of groundwater by evapotranspiration, expressed as a daily rate (in, e.g., mm d$^{-1}$), $S_y$ is specific yield (dimensionless), $r$ is the hourly rate of night-time water table rise (mm h$^{-1}$) during hours when ET is assumed to have no effect (thus reflecting a constant rate of riparian aquifer recharge), and $s$ is the net daily decline in the water table (mm

d$^{-1}$).  This approach and its many subsequent elaborations (e.g., Loheide et al., 2005; Loheide, 2008; Butler et al., 2007; Soylu et al., 2012; Fahle and Dietrich, 2014) are collectively termed the "water table fluctuation" (or WTF) method (Healy

and Cook, 2002).  The WTF method assumes that the daily cycle in ET results only in a daily cycle in groundwater levels, and not a daily cycle in streamflow, which would need to be taken into account in the groundwater mass balance (but see Gribovszki et al., 2008 for an example where this is explicitly included).  The WTF method also implies that a given rate of evapotranspiration (or a given rate of snowmelt input) should be reflected in a given rate of rise or fall in groundwater levels. The WTF method therefore implies that groundwater levels integrate snowmelt or evapotranspiration signals, and thus that

there should be a roughly 6-hour phase lag (see Sect. 3.3 below) between daily groundwater cycles and the evapotranspiration or snowmelt cycles that drive them.

Daily cycles in streamflow have also been widely used to infer evapotranspiration rates, based on summing the "missing streamflow" between the actual streamflow cycle and a line connecting daily peak flows, assumed to represent the

streamflow that would occur in the absence of ET (e.g., Tschinkel, 1963; Hiekel, 1964; Meyboom, 1965; Reigner, 1966; Bond et al., 2002; Boronina et al., 2005; Barnard et al., 2010; Cadol et al., 2012; Mutzner et al., 2015).  The missing streamflow method pre-dates all of these cited applications by decades, given that as early as the 1930's, Troxell observed that "Others have connected the points of maximum discharge during the diurnal fluctuation and assumed that the curve thus obtained would represent the probable flow of the stream if there were no losses, also that the difference between this

quantity and the actual discharge represents the transpiration-loss" (Troxell, 1936).  The latter assumption outlined by Troxell implies that evapotranspiration losses are subtracted 1:1 from streamflow and thus that they are not buffered by changes in groundwater storage.

From the two preceding paragraphs, it should be clear that WTF approaches (for inferring ET rates from groundwater cycles)

and missing streamflow approaches (for inferring ET rates from daily streamflow cycles) are founded on fundamentally incompatible assumptions.  Missing streamflow methods assume that daily cycles in ET are transmitted 1:1 to daily cycles in streamflow, implying that they must not be buffered by changes in groundwater levels (and thus that the groundwater cycles required by WTF approaches cannot exist).  Conversely, WTF approaches assume that daily cycles in ET are volumetrically equal to daily cycles in groundwater levels, implying that no part of these ET cycles can be transmitted to the stream (and

thus that the streamflow cycles required by missing streamflow methods cannot exist).  There may be conditions under which one or the other set of assumptions is approximately correct, but clearly both cannot be valid at the same time.

The timing of daily streamflow maxima and minima, and their lags relative to the daily peaks of snowmelt or ET rates, have also been widely interpreted as reflecting travel times and flow velocities through snowpacks, hillslopes, and river networks

(e.g., Wicht, 1941; Jordan, 1983; Bond et al., 2002; Lundquist et al., 2005; Lundquist and Dettinger, 2005; Wondzell et al., 2007; Barnard et al., 2010; Graham et al., 2013; Fonley et al., 2016).  These applications, like the missing streamflow method, invoke assumptions that are incompatible with those that underlie WTF approaches.  WTF approaches are based on a mass balance in which groundwater integrates ET cycles (because a given ET flux results in a given rate of change in groundwater levels).  This implies that there will be a several-hour phase lag (for the same reason that the integral of a sine

function is a cosine and vice versa) between ET cycles and both groundwater and streamflow cycles (given that streamflows are closely linked to groundwater levels).  This phase lag must be taken into account before inferring travel-time delays from observed time lags between snowmelt or ET cycles and the resulting streamflow maxima or minima.

Clarifying how groundwater and streamflow cycles are linked to the snowmelt or ET cycles that drive them will require

coupled observations of groundwater and stream stage, as well as rates and patterns of snow accumulation and melt, and

daily cycles in vegetation water uptake and its meteorological drivers. Such integrated observational studies are rare. Few studies have examined interactions between snowmelt and ET cycles, though exceptions include Lundquist and Cayan (2002), Mutzner et al. (2015) and Woelber et al. (2018). Likewise, few studies have linked daily cycles in groundwaters and streams, although exceptions include Troxell (1936), Klinker and Hansen (1964), Czikowsky and Fitzjarrald (2004),
Gribovszki et al. (2008), Szilagyi et al. (2008), Loheide and Lundquist (2009), Wondzell et al. (2010), and Woelber et al. (2018). And due to the scarcity of simultaneous spatially distributed measurements spanning mesoscale basins, the spatial aggregation of snowmelt and ET cycles across elevation gradients remains greatly under-studied.

Here we contribute to closing these knowledge gaps using detailed, multi-year ecohydrological time series, including solar
flux, snowmelt, snow water equivalent, riparian tree sap flow fluxes, stream stages (recorded at 12 sites spanning a 500-meter elevation gradient), and groundwater levels (recorded in two dozen wells), from Sagehen and Independence Creeks in California's Sierra Nevada Mountains. These time series, together with a simple conceptual model of riparian groundwater mass balance, demonstrate both the potential and the limitations of using snowmelt- and ET-induced daily cycles in streamflow and groundwater to infer catchment-scale processes. We compare these time series measurements with remote
sensing observations of the spring/summer retreat of the seasonal snowpack and the corresponding advance of photosynthetic activity, to illustrate how daily cycles in groundwater levels and stream stages mirror the spatial and temporal patterns of seasonal ecohydrological transitions at the catchment scale. The Mediterranean climate of Sagehen and Independence Creeks is characterized by heavy winter snowfall and by strong solar radiation and very little precipitation during the snowmelt and growing seasons, making it relatively easy to see how snowmelt and evapotranspiration are
reflected in daily cycles in groundwater and streamflow.

## 2 Field site and data

### 2.1 Field site

The Sagehen (pronounced "sage hen") basin is located on the east slope of California's Sierra Nevada mountain range, approximately 12 km north of the town of Truckee (Fig. 1a). Sagehen Creek is a headwater tributary that flows eastward
from the crest of the Sierra Nevada into Stampede Reservoir on the Truckee River. The catchment ranges in elevation from 2663 m on Carpenter Ridge to 1877 m at the lowermost streamflow monitoring location, where it has a drainage area of 34.7 km$^2$. The uppermost part of the catchment is a steep, glaciated cirque, and the lower catchment is a broad U-shaped valley bordered by broad rolling uplands.

The Sagehen basin has a Mediterranean climate with cold, wet winters and warm, dry summers. Monthly average temperatures recorded at Sagehen Creek Field Station between 1997 and 2009 ranged from -3.5 °C in January to 15.9 °C in July. Average annual precipitation between 1 June 1953 and 31 Dec 2010 at the same location was 850 mm, and average annual snowfall and snow depth were 515 and 33 cm, respectively. Sagehen Creek is downwind of the Sierra crest, so there is a pronounced gradient in precipitation (and particularly in snowfall) from the headwaters toward the eastern (downstream)
end of the basin, due to a combination of declining altitudes and a deepening rain shadow. Because precipitation occurs predominantly in the winter and snowfall accounts for more than 80% of the annual precipitation, the annual runoff is strongly controlled by snowmelt, which generates peak flows in late spring or early summer, with annual minima occurring in the late summer and fall (Godsey et al., 2014).

The Sagehen basin is densely vegetated, with roughly 90% covered by forests and 10% covered by meadows and shrubs. The forest is dominated by lodgepole pine (*Pinus contorta*), Ponderosa pine (*Pinus ponderosa*), Jeffrey pine (*Pinus jeffreyi*),

Douglas fir (*Pseudotsuga menziesii*), sugar pine (*Pinus lambertiana*), white fir (*Abies concolor*), red fir (*Abies magnifica*), and incense cedar (*Calocedrus decurrens*). Grassy meadows are predominantly found along the main stream. Shrub vegetation occurs on soils too poor, rocky, or shallow to support conifer forests, and also as a post-fire or post-harvest

successional stage to mixed conifer forests on deeper, more productive soils (Bailey et al., 1994).

Soils at Sagehen are deep, well-drained acidic Alfisols developed in weathered volcanic parent material. Typically, soil profiles in the Sagehen basin present a dark grayish-brown, gravelly, sandy loam from the surface to roughly 60 cm and a subsoil of yellowish-brown, cobbly, sandy, loam that extends to a depth of 115 cm (Johnson and Needham, 1966).

Lithology is dominated by Tertiary volcanic rocks, primarily Miocene-Pliocene andesitic flows (and, on the north side of the lower Sagehen basin, Pliocene basalt flows), overlying several hundred meters of Tertiary volcaniclastic deposits which in turn overlie Cretaceous granodiorites of the Sierra Nevada batholith (Hudson, 1951; Sylvester and Raines, 2017). This >400 m layer of volcanic rocks hosts a substantial groundwater aquifer, with geothermal data suggesting groundwater circulation to depths exceeding 100 m (Brumm et al., 2009). Mean groundwater ages in springs feeding Sagehen Creek have been

estimated at approximately 28 years during baseflow conditions and 15 years during snowmelt (Rademacher et al., 2005), varying from year to year in response to changes in annual snowmelt volumes and thus recharge rates (Manning et al., 2012). This groundwater system sustains flows in springs, fens, and Sagehen Creek itself during the dry season, which typically lasts from May through September. Even during peak snowmelt, cosmogenic $^{35}$S measurements indicate that over 85% of Sagehen Creek streamflow is derived from stored groundwater, with less than 15% originating as recent snowmelt

(Uriostegui et al., 2017). Quaternary colluvial, alluvial, and glacial deposits lie on top of the volcanic rocks, ranging from a few meters on most hillslopes to >15 m in the riparian zone at lower elevations (Manning et al., 2012). Measured hydraulic conductivities in the surficial deposits near the creek range from $10^{-6}$ to $10^{-4}$ m s$^{-1}$ (Manning et al., 2012), indicating the capacity to support considerable groundwater flow.

The Sagehen basin was affected by extensive timber harvesting, grazing and wildfires in the late nineteenth and early twentieth century, but there has been little change in land use since the early 1950's (Erman et al., 1988). Two access-limited dirt roads cross the catchment, which also hosts a small US Forest Service campground. The only permanent habitation is the headquarters of Sagehen Creek Field Station, and the principal human activity is research (mainly in ecology, biology and hydrology) conducted by several universities and government agencies. Recreational uses include fishing, hunting,

hiking, cross-country skiing, and snowmobiling.

In contrast to the Sagehen basin, the adjacent Upper Independence basin was deeply scoured by Pleistocene glaciers that removed the Tertiary volcanic rocks and exposed the underlying Cretaceous granodiorites over much of the catchment (Sylvester and Raines, 2017). As a result, the Upper Independence basin lacks Sagehen's extensive groundwater system.

Dry-season low flows in Upper Independence Creek are nonetheless sustained by groundwater seeping from the Tertiary volcanic deposits that ring the basin, particularly on the steep north slopes of Carpenter Peak, which retain snow cover long after the rest of the basin has melted out. The Upper Independence basin extends approximately 4 km farther west than the Sagehen basin does, and thus it likely receives somewhat more precipitation, being less affected by the rain shadow of the Sierra crest. The steep north-facing slopes of the Upper Independence basin also keep their snow cover later into the

summer than the Sagehen basin does. Roughly 50% of the Upper Independence basin consists of bare granodiorite outcrops and talus slopes, whereas the Sagehen basin is almost completely vegetated. The Upper Independence Creek basin is largely undisturbed, with no roads, no developed trails, and old-growth forest. Example ground-level views of the Sagehen and Independence basins are shown in Fig. S1.

There were no impoundments or diversions on either Sagehen Creek or Upper Independence Creek at the time of this study. Sagehen Creek has been gauged continuously since 1953 at an altitude of 1929 m and a drainage area of 27.6 km$^2$ (https://waterdata.usgs.gov/ca/nwis/inventory/?site_no=10343500&agency_cd=USGS) as part of the U.S. Geological Survey's Hydrological Benchmark Network (Mast and Clow, 2000), and LiDAR-derived digital elevation data are available from https://opentopography.org/ for both the Sagehen and Upper Independence basins (Kirchner, 2012; Huntington, 2013;

Guo, 2014).  Further background information on the Sagehen basin can be found in Mast and Clow (2000) and on the Sagehen Creek Field Station web site (https://sagehen.ucnrs.org/).

## 2.2 Field instrumentation

    The field data presented here were collected during three water years (defined as 1 October-30 September): 2005-6, 2006-7 and 2007-8.  Solar flux, air temperature, wind velocity, relative humidity, precipitation, and atmospheric pressure were

recorded by a weather station located near Sagehen Creek Field Station (Fig. 1a).  Precipitation, air temperature, snow depth and snow water equivalent (SWE) are also available from three Natural Resources Conservation Service SNOTEL (snow telemetry) stations located adjacent to the Sagehen Creek catchment, each equipped with a snow pillow (https://www.wcc.nrcs.usda.gov/snow/).  The SNOTEL stations are, in order of increasing elevation: i) Independence Creek (1968 m a.s.l.), near the confluence of Independence Creek and Little Truckee River, approximately 7 km NNW from the

Sagehen main stream gauge, ii) Independence Camp (2135 m a.s.l.), near the outflow of Independence Lake, approximately 5 km WNW from the Sagehen main stream gauge, and iii) Independence Lake (2546 m a.s.l.), on the divide between the Sagehen Creek basin and the adjacent Upper Independence Creek basin (Fig. 1a).  These SNOTEL stations lie outside the Sagehen Creek catchment but are adjacent to it, spanning roughly the same altitude range and the same range of distances from the Sierra crest.  Thus they provide a reasonable proxy for the gradient in precipitation, snow accumulation, and

snowmelt timing across the Sagehen basin.

    Sap flow was measured using Granier (1987) thermal dissipation probes (Dynamax inc.) installed in June 2005 in four trees close to the weather station and the B transect of groundwater wells (see below).  Three trees were outfitted with duplicate probes to test for consistency.  The timing and magnitude of sap flow variations were similar among the monitored trees, so

the average of all the available measurements was used for further analysis.  Because our analysis is focused on the timing of sap flow and its relationship to groundwater and streamflow fluctuations, it was not necessary to calibrate the sap flow measurements or quantitatively extrapolate them to stand-scale evapotranspiration fluxes.  The sap flow sensors were not removed and re-inserted into new sites on the tree trunks each year, but instead remained in the same sites; thus the sap flow measurements show year-to-year declines that are artifacts of the wound healing response of the trees.


    Water stage was measured by TruTrack and Odyssey capacitance water level loggers (www.trutrack.com and http://odysseydatarecording.com/, respectively) at six locations along Sagehen Creek (see Table 1 and Fig. 1): the lower culvert, the main gauge (the USGS gauging station), the B transect (approximately 120m west of the main gauge), the D transect (at Kiln meadow), the middle culvert (where the Sagehen road crosses the creek, upstream of Kiln meadow) and the

upper culvert (where the road again crosses the creek, just below its headwater cirque).  Water stage was also measured on three lateral tributaries of Sagehen Creek: one entering from the north (Kiln Creek), and two entering from the south (South Tributaries 1 and 2).  Water stage was also measured at four locations on Upper Independence Creek, of which three are used here.  The Sagehen main gauge stage recorder is co-located with the US Geological Survey gauging station, whereas all other stage recorders were installed specifically for this study (Fig. 1a).  The capacitance water level loggers were calibrated

in the lab and referenced to an arbitrary datum that differed for each stream location.  Therefore, water stage was not comparable from one location to another.  No rating curves were available to convert water stage into discharge, except at

the main gauge. Thus all stream data are presented here as stage, in millimeters relative to an arbitrary datum that varies from site to site.

Shallow groundwater level variations were monitored in 24 wells equipped with TruTrack and Odyssey capacitance water level loggers. In the 1980's, five transects of shallow groundwater wells were hand-augered to 1m, or to refusal, in the Sagehen Basin, and were sleeved with 1.5m PVC pipes (8 cm diameter), perforated over the bottom 0.5 m (Allen-Diaz, 1991). We instrumented 24 wells in the two longest transects, labeled B and D. The B transect crosses Sagehen Creek just downstream of the field station. The northern B transect consists of four wells extending 32m northward from the stream

across the seasonally wet Sagehen East Meadow, close to the weather station and the sap flow trees. The southern B transect consists of ten wells (of which the first nine were instrumented) extending 330m southward from the stream across dry and seasonally wet meadows and, in the farther reaches of the transect, lodgepole pine (*Pinus contorta*) forest (Fig. 1c). The D transect is located at Kiln Meadow, roughly 1.5 km upstream of the B transect. The D transect consists of six wells that extend 132m northward from the stream across a seasonally wet sedge meadow, and nine wells (of which five were

instrumented) that extend 280m southward from the stream across seasonally wet meadows and lodgepole pine forest (Fig. 1b).

**2.3 Field data**

The original meteorological, hydrometric, and sap flow measurements were collected at 10, 15, and 30-minute intervals. All of the records were aggregated to a consistent 30-minute time base for analysis, and all times are reported in Pacific Standard

Time. Weather and snow water equivalent (SWE) data from the three SNOTEL stations were at daily temporal resolution. The stage recorders were downloaded infrequently, and often failed; as a result, data gaps of up to a year in length are found in several of the stage records, and up to two years in some groundwater wells.

To account for the combined role of snowmelt and rainfall during the melting season, we calculated the total water input at

each of the SNOTEL stations by subtracting the net change in snow water equivalent (as measured by the snow pillow) from total precipitation over each daily time step. Thus, any precipitation that was stored as increased SWE was not counted as liquid water input to the catchment until it subsequently melted. Since there is a strong elevation gradient in SWE (see Section 3.1), we computed an area-weighted average of the total water input, assigning a weight to each SNOTEL station by defining its area of influence. We defined three elevation bands centered on each SNOTEL station, with the band limits

defined by the midpoint in elevation between each pair of adjacent stations, and by the top and the outlet of the basin (see Fig. S2). Measurements at each SNOTEL station were weighted according to the catchment area in each elevation band. Independence Creek SNOTEL station (1968 m) had a weight of 32%, Independence Camp (2135 m) had a weight of 58%, and Independence Lake (2546 m) had a weight of only of 10%, reflecting the relatively small fraction of the Sagehen Creek basin at these higher elevations. Because the resulting average water input values are used only for visualization and not for

mass balance analyses, we did not account for other factors (such as slope, aspect, and forest cover) that can also influence the spatial distribution of precipitation and snow accumulation.

To more precisely compare stream stage and groundwater level fluctuations with potential weather drivers, we estimated the rate of change of stage or groundwater level for each time step $i$ from the difference between the measurements immediately

before and after, i.e.,

$$\left(\frac{dh}{dt}\right)_i = \frac{h_{i+1} - h_{i-1}}{2\,\Delta t} \tag{2}$$

where $h$ is groundwater level or stream stage, $t$ is time, and $\Delta t$ is the sampling interval (0.5 hours). Thus the rates of change reported here are averaged over one hour, centered on each 30 minutes. To visualize daily stream and groundwater variations while excluding longer-term patterns, we also calculated water level anomalies relative to the running 24-hour average as follows:

$$h'_i \; = \; h_i - \frac{1}{48} \left( \sum_{j=i-24}^{i-1} h_j + \sum_{j=i+1}^{i+24} h_j \right) \tag{3}$$

where $h'_i$ is the detrended water level, relative to a 24-hour average composed of 48 half-hourly measurements surrounding (but excluding) $h_i$ itself.

## 3. Analysis, Results and Discussion

### 3.1 Climate forcing

Precipitation, air temperature, and SWE data at the three SNOTEL stations clearly show an elevation gradient in precipitation and snow accumulation across the Sagehen Creek catchment (Fig. 2, Tables 2 and 3). Precipitation patterns were similar among the three stations, with year-to-year Pearson's correlation coefficients for total cumulative precipitation for 29 water years (1981-2009) between 0.94 and 0.98 (p<0.01, n=29) across all pairs of sites. SNOTEL stations at higher elevations (and also closer to the Sierra crest) had somewhat higher precipitation totals, and also markedly greater seasonal snow accumulation despite a difference of less than 1 °C in average temperature across the nearly 600 m range of elevations (Tables 2 and 3). The higher-altitude stations also began accumulating snow earlier in the winter, and their melt seasons began later and lasted longer (Fig. 2).

Annual precipitation totals and peak SWE varied substantially from year to year, with larger cumulative precipitation totals and peak snow-water equivalent in water year 2005-6, followed by 2007-8 and 2006-7 (Fig. 2, Tables 2-3). Comparison with the long-term averages (Table 3) shows that precipitation at all stations was well above the long-term average in 2005-6, and well below the long-term average in the other two water years.

During the summer and early fall, intense solar fluxes and high temperatures (with daily highs often well above 30°C) created ideal conditions for high evapotranspiration fluxes. Consistent with Sagehen's Mediterranean climate, from May to October precipitation events were infrequent, sporadic, and generally small. Thus the hydrologic effects of snowmelt and evapotranspiration were minimally obscured by precipitation from late spring through early fall.

### 3.2 Climatic control on daily cycles in stream stage and groundwater level

Clearly visible daily cycles were observed in all water level records (both stream stages and groundwater levels) during rain-free periods between late spring and early autumn. Daily cycles in several stream stage records (particularly the upper culvert and the three tributary streams) became indistinct as the streams dried up; likewise the daily cycles in several groundwater wells became indistinct as the water level reached the bottom of the sensor. Our analysis of groundwater cycles will focus on the northern side of the B transect, just downstream of the field station (Fig. 1c), because records from three of the four water level sensors are complete for two full years, and because this transect is situated close to the weather station and the trees equipped with sap flow sensors.

During the snowmelt period in late spring, stream stages and groundwater levels typically reached their maxima in late afternoon and their minima shortly after dawn (Fig. 3c-d).  This temporal pattern has also been observed in previous studies

of snowmelt-induced daily cycles in streamflow and groundwater levels (e.g., Loheide and Lundquist, 2009; Lundquist et al., 2005; Lundquist and Dettinger, 2005), and has been attributed to daytime melting of the snowpack during periods of high temperatures and strong solar radiation (Fig. 3).  During the summer, the phase of the daily cycles reversed, with stream stages and groundwater levels typically reaching their maxima in the early morning and their minima late in the afternoon (Fig. 4).  This temporal pattern has also been observed in previous studies of evapotranspiration-induced daily cycles in

streamflow and groundwater levels during dry periods (e.g., Kozeny, 1935; Troxell, 1936; Dunford and Fletcher, 1947; Hiekel, 1964; Klinker and Hansen, 1964; Burt, 1979; Kobayashi et al., 1990; Lundquist and Cayan, 2002; Butler et al., 2007; Wondzell et al., 2007; Gribovszki et al., 2008; Gribovszki et al., 2010), and has been attributed to daytime riparian evapotranspiration in response to strong solar fluxes and low relative humidity (Fig. 4).  The night-time rebound in groundwater levels can be attributed to groundwater recharge delivered to the alluvial aquifer from upslope (Tschinkel,

1963).  The average groundwater level, and thus average discharge to the stream, will adjust to the balance between the average recharge from upslope and the average evapotranspiration losses.  During summer days, however, evapotranspiration losses will be substantially higher than the 24-hour average, so the short-term flux balance in the riparian aquifer will be negative and groundwater levels (and thus drainage rates to the stream) will fall during the daytime.  Conversely, at night evapotranspiration losses will be minimal, the short-term flux balance will be positive, and groundwater

levels (and thus streamflows) will rise (e.g., Troxell, 1936; Tschinkel, 1963).  Figure 5 shows a simplified schematic of the mass balance that determines the evolution of riparian groundwater storage, and thus the rise and fall in stream discharge over time.

The examples shown in Figs. 3 and 4 illustrate the fundamental role of solar radiation in driving daily fluctuations in the

stream and in groundwater.  The rate of rise and fall in groundwater levels is tightly coupled to the solar flux (top panels in Figs. 3 and 4) in both the snowmelt-dominated and evapotranspiration-dominated periods.  However, the sign of that coupling reverses between the two periods, consistent with solar radiation driving water inputs to the riparian zone during spring snowmelt, and driving water extraction from the riparian zone by evapotranspiration during mid-summer.  During mid-summer, the daily cycle in the solar flux is very tightly correlated with the sap flow flux (top panel of Fig. 6), and both

the solar flux and the sap flow flux are very tightly correlated with the rate of decrease in groundwater levels (note the inverted scale of the groundwater fluctuations in the bottom two panels of Fig. 6).  Day-to-day, and even hour-to-hour, variations in solar flux are reflected in both sap flow rates and riparian zone groundwater declines (Fig. 6).  During snow-free periods, approximately the same timing of daily cycles is observed among most of the wells, both in meadows and in adjacent forests (Fig. S3), suggesting that they reflect a local synchronous response to ET forcing.

Variations in stream stage are synchronous, or nearly so, with variations in groundwater levels (Figs. 3-4), further suggesting strong coupling between the stream and the riparian aquifer (Troxell, 1936; Cadol et al., 2012).  One can of course question whether the groundwater cycles drive the stream stage cycles, or the other way around, as has been reported in some riparian meadows (e.g., Loheide and Lundquist, 2009) and glacial forefields (e.g., Magnusson et al., 2014).  However, that possibility

can be excluded in the case of the B transect shown in Figs. 4-6, because the mean water levels in the wells are 0.2-1 m above the stream stage, and both the water levels and the amplitudes of the daily cycles increase with distance from the channel (Fig. 7).  When groundwater cycles are driven by stream stage variations, by contrast, their amplitude decreases with distance from the stream.

In July 2009, following the field measurements reported here, the US Geological Survey drilled several deeper wells adjacent to the stream channel at the B transect, the D transect, and the middle culvert (Manning et al., 2012). At the B transect, a well drilled to a depth of 10.4 m (and screened below 4.3 m depth) had a static water level of 1.5 m above the stream, and 0.6 m above the ground surface. At the D transect, a well drilled to a depth of 14.5 m (and screened below 2.3 m depth) had a static water level of 0.7 m above the stream. And just upstream from the middle culvert, a well drilled to a

depth of 13.1 m (and screened below 2.4 m depth) had a static water level of 0.3 m above the stream (see Tables A1 and A2 of Manning et al., 2012). These water levels, recorded in September 2009 under dry conditions, demonstrate that groundwater feeds the stream rather than the other way around, even under the driest conditions. These measurements also demonstrate an upward hydraulic gradient in the valley axis at all three locations, consistent with fracture flow from upslope recharging the riparian aquifer during mid-summer, thus sustaining both streamflow and plant water use.

**3.3 Dynamical phase lags between solar flux and hydrometric response**

A clear feature seen in Figs. 3, 4, and 7, and in many previous studies, is the time lag between the daily cycles of solar flux and groundwater and streamwater levels: the solar flux peaks near noon, but the water levels reach their maximum (or, during ET-dominated periods, their minimum) in late afternoon or early evening. This time lag has been widely interpreted as indicating the time it takes for a pulse of water from snowmelt (or, conversely, a pulse of water removal by ET) to reach

the channel, or to travel downstream to the measurement point (e.g., Wicht, 1941; Jordan, 1983; Bond et al., 2002; Lundquist et al., 2005; Lundquist and Dettinger, 2003, 2005; Wondzell et al., 2007; Barnard et al., 2010; Graham et al., 2013; Fonley et al., 2016). Here we show that, at least in small catchments, this is not primarily a travel-time lag, but rather a dynamical phase lag. Dynamical phase lags arise whenever one system component integrates another. In this case, because riparian groundwater integrates meltwater and evapotranspiration fluxes, daily cycles in groundwater should lag those in meltwater

or evapotranspiration by roughly six hours, even in the absence of travel-time lags, for the same reason that a sine wave input, when integrated, yields a cosine wave with a 90-degree phase lag relative to the input. Streamflows depend on riparian groundwater levels; thus, streamflow maxima and minima lag behind peak snowmelt or ET because it takes time for the effects of each day's snowmelt or ET to accumulate in the riparian aquifer (see Fig. 5).

We demonstrate this principle using a simple conceptual model of a stream and its adjacent riparian aquifer. Following the simple dynamical systems approach of Kirchner (2009) we assume that stream discharge ($Q$) depends directly on the storage ($S$) in the riparian aquifer, which is recharged by liquid precipitation ($P$), snowmelt ($M$) and groundwater flow from upslope ($G$), and is drained by stream discharge ($Q$) and evapotranspiration ($ET$). This simple dynamical system, shown in simplified form in Fig. 5, can be represented mathematically as:

$$Q = f(S) \quad \text{and} \quad \frac{dS}{dt} = P + M + G - ET - Q \ , \tag{4}$$

where storage is expressed in volume per unit riparian area, and fluxes are expressed in volume per unit riparian area per unit time. Any other consistent system of units can also be used (for example, storage in volume per unit stream length and fluxes in volume per unit stream length per time); the numerical values will differ, but the equations and the underlying concepts remain the same. Several mechanisms may link increases in riparian storage to increases in stream discharge,

including steepening of hydraulic gradients, rising water tables reaching shallower, more permeable till layers (the "transmissivity feedback" hypothesis of Bishop, 1991), increasing connectivity between local zones of mobile saturation (Tromp-van Meerveld and McDonnell, 2006), extension of flowing stream networks (Godsey and Kirchner, 2014; Van Meerveld et al., 2019), and activation of preferential flowpaths.

Directly from the form of Eq. (4), we can see that maxima or minima in riparian storage (and thus groundwater level) will
generally lag maxima in the fluxes of meltwater or evapotranspiration, for the simple reason that these fluxes directly control
the rate of change of storage ($\mathrm{d}S/\mathrm{d}t$), and storage itself integrates this rate of change over time. Thus, for example, the peak
of a daily snowmelt pulse will not correspond to the peak in storage (and thus discharge), but rather to the fastest rate of
increase of storage (and thus of discharge). The peak of storage and discharge will instead occur later, as the snowmelt pulse

is ending and the flux balance in Eq. (4) is shifting from positive to negative. One can see this behavior in Figs. 3 and 4:
groundwater levels change fastest near the peak of the solar flux (Figs. 3a, 4a, and 5b), but the groundwater levels
themselves reach their maxima (or, for ET cycles, minima) several hours later (Figs. 3c, 4c, and 5c), when the rate of
groundwater rise/fall changes sign.

Integration of a periodically cycling input implies a phase lag of roughly one-quarter cycle in the output, or roughly six hours
in the case of a daily cycle in meltwater or ET forcing. The exact value of the time lag will depend on the shape of the cyclic
forcing function and the form of the relationship between storage and discharge. For purposes of illustration, we can make
the simplifying assumption that discharge is a linear function of storage $Q = f(S) = S/\tau$, where $S$ is riparian storage relative
to the level of the stream and $\tau$ represents the characteristic response timescale of the linear reservoir. In real-world cases,

storage-discharge relationships are likely to be strongly nonlinear (e.g., Penna et al., 2011), with the characteristic response
time $\tau$ being shorter at high flows (as may occur, for example, during peak snowmelt) than during summer low flows (e.g.,
Sect. 12 of Kirchner, 2009). Nonetheless, any nonlinear storage-discharge relationship will be approximately linear over a
sufficiently narrow range of storage variations, such as one would expect for individual daily cycles of storage and
discharge. Making this assumption, Eq. (4) becomes the first-order linear differential equation

$$\tau \frac{\mathrm{d}Q}{\mathrm{d}t} = P + M + G - ET - Q \quad , \tag{5}$$

where $P$, $M$, $G$, and $ET$ may all be time-varying, and $Q$ and $\mathrm{d}Q/\mathrm{d}t$ are related to $S$ and $\mathrm{d}S/\mathrm{d}t$ by the proportionality
constant $1/\tau$. We can further assume that, at least over small ranges of riparian groundwater levels, specific yield (drainable
porosity) $S_y$ is approximately constant and thus the rate of change in groundwater level is $\frac{\mathrm{d}h_G}{\mathrm{d}t} = \frac{\mathrm{d}S}{\mathrm{d}t}\frac{1}{S_y}$. We can also assume
that, at least over small ranges of stream stage, the slope $m$ of the stage-discharge rating curve is nearly constant and thus the

rate of change in stream stage is $\frac{\mathrm{d}h_Q}{\mathrm{d}t} = \frac{\mathrm{d}Q}{\mathrm{d}t}\frac{1}{m} = \frac{\mathrm{d}S}{\mathrm{d}t}\frac{1}{m\tau}$. Thus $S_y$ and $m$ can be used to convert storage and discharge
variations into changes in groundwater levels and stream stages.

The assumptions underlying this simple model are similar to those made by Gribovszki et al. (2008) in their analysis of
riparian evapotranspiration. Our assumptions differ from those of Troxell (1936), Loheide (2008), Cadol et al. (2012), and

Soylu et al. (2012) because our analysis explicitly recognizes that the rate of discharge from the riparian aquifer to the stream
is not constant, but instead depends on riparian aquifer storage. Our analysis also differs fundamentally from those of Bond
et al. (2002), Lundquist and Dettinger (2003, 2005), Lundquist et al. (2005), Wondzell et al. (2007), and Graham et al.
(2013), who assume that water fluxes from snowmelt or evapotranspiration are added or subtracted 1:1 from streamflow
itself, rather than from a riparian aquifer that feeds the stream (which buffers and phase-lags the hydrologic signals that the

stream receives).

If the external forcing is sinusoidal, solving a linear equation like (5) is a well-known textbook problem in linear systems
theory. For example, if the combined forcing $P + M + G - ET = \bar{Q} + A\cos(\omega t)$, where $\bar{Q}$ is average discharge, $A$ is the
amplitude of the forcing cycle, and $\omega$ is its angular frequency (and thus for a daily cycle, $\omega = 2\pi$ day$^{-1}$), Eq. (5) can be

solved analytically to yield

$$Q = \bar{Q} + \frac{A}{\sqrt{1 + \omega^2\tau^2}} \cos(\omega t - \phi) \quad , \tag{6a}$$

$$S = \tau Q = \bar{S} + \tau \frac{A}{\sqrt{1 + \omega^2\tau^2}} \cos(\omega t - \phi) \quad , \tag{6b}$$

$$\frac{dQ}{dt} = \omega \frac{A}{\sqrt{1 + \omega^2\tau^2}} \cos\left(\omega t + \frac{\pi}{2} - \phi\right) \quad , \quad \text{where} \tag{6c}$$

$$\frac{dS}{dt} = \omega \tau \frac{A}{\sqrt{1 + \omega^2\tau^2}} \cos\left(\omega t + \frac{\pi}{2} - \phi\right) \quad , \quad \text{and} \tag{6d}$$

$$\phi = \arctan(\omega\tau) \quad . \tag{6e}$$

Thus, in this simplified example, streamflow will be a sinusoidal cycle that is damped by a dimensionless factor of $\sqrt{1 + \omega^2\tau^2}$, storage will be a sinusoidal cycle that is damped by a factor of $\sqrt{\frac{1}{\tau^2} + \omega^2}$ (which has dimensions of 1/time), and both storage and streamflow will be phase-shifted by an angle $\arctan(\omega\tau)$ relative to the external forcing. These sinusoidal cycles in storage and discharge, when re-scaled by factors of $\frac{1}{S_y}$ and $\frac{1}{m}$, respectively, will yield the corresponding sinusoidal

cycles in groundwater level and stream stage. If the riparian aquifer's response time $\tau$ is short (such that $\omega\tau \ll 1$), the cycle in $S$ will be small (Eq. 6b) and most of the cycle in the forcing will be transmitted directly to $Q$, so the amplitude of the cycles in $Q$ will nearly equal the amplitude of the forcing (Eq. 6a). In this case, the phase shift $\phi$ will be small (Eq. 6e) and the cycles in storage (and thus discharge) will be nearly synchronized with the forcing (Eqs. 6a and 6b). Thus, the assumptions underlying "missing streamflow" methods, as outlined in Sect. 1, are met when the aquifer's response time $\tau$ is

short ($\omega\tau \ll 1$; for a daily cycle this corresponds to $\tau \ll 4$ hours). Conversely, in the more typical case that the riparian aquifer's response time $\tau$ is long enough that $\omega\tau \gg 1$, the forcing cycles will mostly be absorbed by variations in storage (which now will mostly integrate the forcing cycles rather than transmitting them to discharge). This more typical case corresponds to the assumptions underlying water table fluctuation (WTF) methods for inferring ET from groundwater cycles (as outlined in Sect. 1). When $\omega\tau \gg 1$, cycles in the rate of rise and fall in riparian storage $dS/dt$ will have nearly the same

amplitude as the forcing (Eq. 6d), but cycles in stream discharge $Q$ will be strongly damped (Eq. 6a). In this case, the phase shift $\phi$ will approach $\frac{\pi}{2}$ (Eq. 6e) and thus cycles in storage and discharge will lag the forcing by about 90 degrees, or six hours for a daily cycle (Eqs. 6a and 6b). However, unlike the storage and discharge themselves, their rates of rise and fall (that is, $dS/dt$ and $dQ/dt$ in Eqs. 6d and 6e) will be nearly synchronized with the forcing (e.g., Fig. 6), because their phase lags $\phi - \frac{\pi}{2}$ will be small.


In less idealized cases, the forcing functions $P$, $M$, $G$, and $ET$ may be non-sinusoidal (but nonetheless periodic) functions of time. In such cases, Eq. (5) can be solved to any desired precision using Fourier methods. The solution is straightforward because the Fourier transform of the derivative operator is simply $i\omega$ – that is, $\text{F}\left(\frac{dx}{dt}\right) = i\omega\,\text{F}(x)$, where $\text{F}()$ denotes the (complex) Fourier transform , $x$ is some function of time, and $\omega$ is angular frequency – so the Fourier transforms of ordinary

differential equations are algebraic equations. For example, the Fourier transform of Eq. (5) is

$$i\,\omega\,\tau\,\text{F}(Q) = \text{F}(P + M + G - ET) - \text{F}(Q) \quad , \tag{7}$$

with the solutions

$$\text{F}(Q) = \frac{1 - i\,\omega\,\tau}{1 + \omega^2\,\tau^2} \text{F}(P + M + G - ET) \quad , \tag{8a}$$

$$\mathrm{F}(S) = \tau\, \mathrm{F}(Q) = \tau \frac{1 - \mathrm{i}\,\omega\,\tau}{1 + \omega^2\,\tau^2}\, \mathrm{F}(P + M + G - ET) \quad, \tag{8b}$$

$$\mathrm{F}\!\left(\frac{\mathrm{d}Q}{\mathrm{d}t}\right) = \mathrm{i}\,\omega\, \mathrm{F}(Q) = \mathrm{i}\,\omega \frac{1 - \mathrm{i}\,\omega\,\tau}{1 + \omega^2\,\tau^2}\, \mathrm{F}(P + M + G - ET) \quad, \quad \text{and} \tag{8c}$$

$$\mathrm{F}\!\left(\frac{\mathrm{d}S}{\mathrm{d}t}\right) = \mathrm{i}\,\omega\, \mathrm{F}(S) = \mathrm{i}\,\omega\,\tau \frac{1 - \mathrm{i}\,\omega\,\tau}{1 + \omega^2\,\tau^2}\, \mathrm{F}(P + M + G - ET) \quad. \tag{8d}$$


Equation (8) can be applied straightforwardly by 1) taking the Fourier transforms of the forcing functions, 2) multiplying the resulting complex Fourier coefficients as shown in Eq. (8) to obtain the Fourier transforms of the discharge $Q$, storage $S$, and their rates of change $\frac{\mathrm{d}Q}{\mathrm{d}t}$ and $\frac{\mathrm{d}S}{\mathrm{d}t}$, and then 3) taking the inverse Fourier transforms to obtain $S$, $Q$, $\frac{\mathrm{d}Q}{\mathrm{d}t}$, and $\frac{\mathrm{d}S}{\mathrm{d}t}$ as functions of time. This Fourier method is preferable to numerically integrating Eq. (5) because initialization is not required (the input, and the solution, go on forever in both directions) and there is no risk of numerical instability. In Fig. 8 we show the behavior of Eq. (5) assuming no precipitation ($P$), a constant groundwater inflow rate ($G$), and a reasonable range of riparian aquifer response times $\tau$ (0.2 to 5 days). We represent both snowmelt rates $M$ (dark blue curves) and evapotranspiration rates $E$ (light blue curves) using a rectified half-wave cosine function (Fig. 8a), which roughly approximates the mid-summer solar flux curve (Figs. 4-7).

The daily cycles shown in Fig. 8b-d are asymmetrical, rising or falling more steeply during the daytime than their subsequent recovery at night. They also exhibit large apparent time lags, with discharge peaks (or minima in the case of ET cycles) between 3 and 5 PM, depending on the value of $\tau$, and discharge minima (or maxima in the case of ET cycles) between 6 and 7 AM. Figure 9a shows that these time lags remain several hours long for all aquifer response times $\tau$ longer than about 0.1 day. These time lags, as well as the asymmetry of the discharge curves, have often been attributed to travel-time delays for transport through the snowpack, aquifer, or river network (Wicht, 1941; Jordan, 1983; Bond et al., 2002; Lundquist et al., 2005; Lundquist and Dettinger, 2003, 2005; Wondzell et al., 2007; Barnard et al., 2010; Graham et al., 2013; Fonley et al., 2016). Figure 8 shows instead that they can arise purely from the internal dynamics of the riparian aquifer itself, as it integrates either cyclic water inputs from snowmelt or cyclic riparian losses from ET. That is, lags between snowmelt or ET and discharge cycles can arise purely as dynamical phase lags, determined by the characteristic response time $\tau$ of the aquifer in relation to the shape and period of the cyclic forcing. These dynamical phase lags must first be taken into account before any additional lag can be attributed to the celerity of kinematic waves in snowpacks, hillslopes, or stream channels. (Jordan (1983) is the only one of the authors cited above who explicitly recognizes that the propagation speed of the daily cycles is determined by kinematic wave celerity. The others appear to assume that these cycles propagate at the speed of water movement per se, which is inconsistent with decades of work on both snowpacks (e.g., Colbeck, 1972) and streams (e.g., Beven, 1979). Like the apparent time lag, the asymmetry in the discharge cycles can arise simply because the daytime forcing is briefer, and stronger, than the night-time rebound of the riparian aquifer (see also Czikowsky and Fitzjarrald, 2004, for a similar analysis of this asymmetry based on somewhat different assumptions). Although the time-integrating behavior of the riparian aquifer is recognized by many riparian groundwater models (e.g., Loheide et al., 2005; Loheide, 2008; Soylu et al., 2012), it is almost universally overlooked in studies of daily streamflow cycles.

Figures 8 and 9a also show that over wide ranges of the aquifer response time $\tau$, and particularly for $\tau \geq 0.5$ day, the rate of change of discharge $\mathrm{d}Q/\mathrm{d}t$ (Figs. 8f-g and dotted lines in Figs. 9a-b), unlike discharge itself (Figs. 8c-d and solid lines in Figs. 9a-b), closely mirrors the cyclic forcing by snowmelt or ET. This occurs because unless $\tau$ is small compared to the period of the cyclic forcing, the variations in the term $S/\tau$ in Eq. (5) will be small compared to the variations in the forcing

by meltwater ($M$) or evapotranspiration ($ET$), with the result that cycles in $dS/dt$ (and by extension $dQ/dt$) will closely correspond to cycles in the forcing itself.  This observation implies that to track travel-time lags through the hydrologic

system, one should look for lagged correlations between cycles in solar forcing and *rates of change* in groundwater levels, stream discharges, or stream stages (rather than those levels, discharges, or stages themselves, which will be shifted by the dynamical phase lags shown in Fig. 9a).  Cross-correlating each day's cycle in $dh_Q/dt$ between the main gauge and the lower culvert, we find that the average time lag between them is 0.96±0.04 hours, for an average celerity of 3.4 km hr$^{-1}$ over the 3.3 km reach between these two points.  Changes in flow depth should propagate downstream with the celerity of a

kinematic wave, $c = dQ/dA$, where $A$ is the cross-sectional area of the channel (Beven, 1979).  Predicting kinematic wave celerity requires estimates of channel cross-sectional area across a range of discharges, which are available at Sagehen only for stage measurements at the main gauge and the B transect.  These two sites are broadly representative of the pools and riffles, respectively, which make up most of the morphology of lower Sagehen Creek, and their wave celerities imply average lag times of 2.3 and 1.0 hours, respectively, for changes in discharge to travel between the main gauge and lower

culvert.  A precise comparison is not possible, because the channel also receives synchronized snowmelt or evapotranspiration signals along the reach between these two measurement points (which have shorter lag times than one would expect for a kinematic wave to travel the full distance).  Nonetheless, these calculations suggest that the daily cycles in $dh_Q/dt$ propagate downstream as kinematic waves, as expected, with the superposition of local signals added by the riparian aquifer en route.


A further interesting consequence of the simple model in Eqs. (4) and (5) is that the peak in the rate of change in the riparian aquifer comes slightly *before* the peak in the rate of snowmelt or evapotranspiration (see Fig. 8d-f).  This model behavior mimics the time shifts shown in Fig. 6, in which the sap flow curve lags the solar flux curve by about an hour, but the peak rate of change in groundwater *leads* the sap flow curve by about an hour (and thus is nearly synchronized with the solar

flux).  This seems counterintuitive: it looks like the change in groundwater precedes the sap flow curve, and thus an effect precedes its cause.  However, it results directly from the fact that the aquifer integrates both the sap flow flux and the discharge flux, coupled with the fact that near the noontime peak, storage and thus discharge are declining over time, meaning discharge is slightly lower (and thus that groundwater storage is declining slightly slower) at noon than just before noon.  As one can see from the dotted line in Fig. 9a, this counterintuitive (but physically and mathematically correct)

negative lag can be several hours long if the riparian aquifer response time $\tau$ is much shorter than 1 day.  For more typical aquifer response times, however, this negative lag may be short enough that it is difficult to detect.

Several studies have sought to use daily cycles in streamflow to quantify riparian evapotranspiration rates, or to estimate the fraction of the catchment that can transmit ET signals to streamflow (e.g., Tschinkel, 1963; Meyboom, 1965; Reigner, 1966;

Bond et al., 2002; Boronina et al., 2005; Barnard et al., 2010; Cadol et al., 2012; Mutzner et al., 2015).  The simulations shown in Fig. 8 show that any such inferences are problematic, because the amplitudes of daily cycles in streamflow depend not only on the snowmelt or ET forcing, but also on the riparian aquifer response time $\tau$.  For example, Figs. 8c and 8d, or Figs. 8f and 8g, show daily streamflow cycles that are nearly identical, but whose amplitudes differ by a factor of five, resulting from exactly the same forcing but a factor-of-five difference in the aquifer response time $\tau$.  As the blue lines in

Fig. 9b show, the amplitudes of the daily cycles in discharge (and in the rate of change in discharge) are strongly dependent on the response time $\tau$ whenever $\tau > 0.1$ days or so.  As $\tau$ becomes larger, discharge becomes less sensitive to changes in storage, and daily cycles in riparian storage due to snowmelt or ET are reflected in smaller daily cycles in streamflow.  This is doubly problematic because the time "constant" $\tau$ will not actually be constant, but instead will vary as the catchment dries out over long recession periods, if the storage-discharge relationship is nonlinear (Kirchner, 2009).  However, for all

response times $\tau$ greater than about 0.2 days, the amplitude of daily cycles in the rate of change in riparian storage ($dS/dt$) is

very close to the amplitude in the snowmelt or ET forcing (dotted green line in Fig. 9b).  These results suggest that it may be possible to quantitatively infer riparian ET rates from daily cycles in the rates of rise and fall in riparian groundwater, but not from daily cycles in groundwater levels themselves (solid green line in Fig. 9b), or from daily cycles in streamflow (blue lines in Fig. 9b).


Although this conceptual model has been developed in the context of Sagehen Creek, which has an extensive groundwater aquifer, the mechanisms described here do not require substantial aquifer storage.  In the model, changes in discharge equal changes in storage divided by the characteristic response time $\tau$.  This directly implies that the daily range of storage also equals $\tau$ times the daily range of discharge.  At the Sagehen main gauge, where we can measure daily cycles in units of

discharge (at the other stations we lack rating curves and thus have only stage measurements), typical daily ranges of discharge during peak snowmelt were ~2-4 mm/day in 2006 (above-average SWE), 0.2-0.6 mm/day in 2007 (below-average SWE), and 0.4-1 mm/day in 2008 (roughly average SWE).  Even $\tau$ values as small as  ~0.2-0.5 days are sufficient to generate significant lags between peak snowmelt and peak streamflow, implying that these lags could be associated with storage changes of only 0.4-2 mm in 2006, 0.04-0.3 mm in 2007, and 0.08-0.5 mm in 2008 (the ET cycles, and their

associated ranges of storage, are about 1-2 orders of magnitude smaller).  This simple calculation implies that significant dynamical phase lags can be generated from small daily variations in soil water and shallow groundwater, and that a substantial groundwater aquifer is not required.

This inference can be tested by comparing daily streamflow cycles in Sagehen Creek with those in Upper Independence

Creek. The Upper Independence basin is dominated by glacially scoured granodiorites (Sylvester and Raines, 2017) and lacks the volcanic and volcaniclastic deposits that host Sagehen's extensive groundwater aquifer.  Despite this sharp contrast in hydrogeology, Figs. 10 and 11 show that snowmelt and ET cycles are strikingly similar in Upper Independence Creek and Sagehen Creek.  Streamflow cycles lag the solar flux curve by slightly more at the Sagehen main gauge (Figs. 10f and 11f) than at the other four stations shown in Figs. 10 and 11, reflecting the fact that the main gauge is farther downstream from its

most distant headwaters (7.9 km, compared to 2.6-3.9 km for the other four stations) and integrates over a larger drainage area (27.6 km$^2$ vs 4.7-7.7 km$^2$ for the other stations), and thus accumulates commensurately larger kinematic wave lags.  The daily cycle amplitudes also differ, due to differences in drainage areas and channel cross-sections among the different stations.  Nevertheless, the clear conclusion from Figs. 10 and 11 is that the shapes of the daily cycles, and their phase lags relative to the solar flux, do not differ substantially between the granitic, glacially scoured Upper Independence basin and the

groundwater-dominated Sagehen basin.  This strongly suggests that similar mechanisms shape the streamflow cycles in both basins, despite the marked differences between their geological settings.

### 3.4 Correlations between solar flux and changes in water levels: the diel cycle index

The analyses presented in Sects. 3.2 and 3.3 clearly show that rates of change in groundwater levels and stream stages are coupled to solar flux forcing through two different mechanisms – snowmelt and evapotranspiration – that have opposite

effects.  If forcing by solar flux drives snowmelt, groundwater levels and stream stages rise during the day, and decline at night.  Conversely, if forcing by solar flux drives evapotranspiration, groundwater levels and stream stages rise at night, and decline during the day.  The very close coupling between solar flux and water level response (particularly in groundwater; see Fig. 6) suggests that the correlation between solar forcing and rates of change in water levels could be used to indirectly measure how much those water levels are influenced by snowmelt (thus resulting in positive correlations) or

evapotranspiration (thus resulting in negative correlations).

Figure 12 illustrates the concept.  For each day, we calculated the Pearson product-moment correlation coefficient between the solar flux in each 30-minute period and the simultaneous rate of change in stream stage.  The two upper plots in Fig. 12 show two sample days, one near peak snowmelt (showing a clear positive correlation with solar flux), and the other in mid-summer (showing a clear negative correlation with solar flux).  We excluded any days when the total solar flux was less than 80% of the clear-sky value for that day of the year, because one would not expect a clear correlation with solar forcing on days with heavy cloud cover.  Days when more than 5 mm of precipitation fell were also excluded, as were days when more than 5 mm of precipitation fell on the previous day, as a precaution against spurious correlations that might arise from the catchment's storm runoff response.  As Fig. 12 shows, these daily correlations provide a dimensionless index that expresses the relative influence of snowmelt (correlation ≈+1) and evapotranspiration (correlation ≈-1) as drivers of groundwater and streamflow fluctuations.  We therefore call these correlations the "diel cycle index", as a more efficient shorthand for "daily correlations between solar flux and the rate of rise and fall in stream stage or groundwater level".  (The term "diel" refers to 24-hour cycles, whereas the frequently used alternative term "diurnal" strictly refers only to daytime, just as "nocturnal" refers to night-time.)

## 3.5 Destructive interference between snowmelt and evapotranspiration cycles

Several circumstances can result in diel cycle index values near zero.  When catchments are dry or frozen, stream stages can decline to the point that stage fluctuation measurements are dominated by noise (from instrument limitations, surface waves, eddies, and so forth) and thus correlations with solar flux may be weak.  Overcast and rainy periods can also lead to confounded results, which is why they are excluded by the filters described above.  Last but not least, during the transition between snowmelt-dominated and evapotranspiration-dominated periods, the stream will feel the offsetting effects of both snowmelt and evapotranspiration, as illustrated schematically in Fig. 13.  The stream will integrate both snowmelt cycles (e.g., from higher altitudes and north-facing slopes that are still snow-covered) and evapotranspiration cycles (e.g., from lower altitudes and south-facing slopes that have already melted out), and because these two cycles have opposite phases, they will destructively interfere (see Fig. 13).

As the melt season ends, the snowpack will contract and become fragmented, and thus the stream and the groundwater system will be fed by a declining snowmelt flux (blue line in Fig. 13), making the snowmelt cycles weaker.  As spring gives way to summer, evapotranspiration fluxes will increase as temperatures and solar fluxes both rise, strengthening the evapotranspiration cycles over time (red line in Fig. 13).  From the observer's perspective, it will appear as if the snowmelt cycle disappears and then the evapotranspiration cycle grows to take its place (bottom panel in Fig. 13).  But what is actually occurring instead is that both cycles are present simultaneously, one becoming weaker and the other becoming stronger, and cancelling one another when they are of equal strength.  Thus in settings where both processes are active, we should keep in mind that we will always observe their net effects, and not just whichever process is dominant (see also Mutzner et al., 2015).

## 3.6 Differing transitions between snowmelt and evapotranspiration cycles in groundwater and streamflow

Figure 14 shows how the diel cycle index evolves over time at the B transect at Sagehen Creek.  During the winter and early spring, the diel cycle index generally ranges between about 0.5 and 1, indicating that intermittent snowmelt is the main driver of daily streamflow cycles.  Conversely, during the summer when the sap flow measurements indicate active transpiration, the diel cycle index is generally close to -1 in groundwater and roughly -0.7 to -1 in the stream.  In April and May of 2007 one can see that the diel cycle index transitions from positive to negative values later, and more slowly, in the southern B transect of groundwater wells (on the north-facing side of the valley) than in the northern B transect (on the south-facing side of the valley), reflecting longer-lasting snow patches on the north-facing slopes.

The most striking contrast, however, is between the transition in the diel cycle index values in the groundwater wells, which
respond to the local balance between snowmelt and evapotranspiration forcing, and in the stream, which responds to the
integrated effects of that forcing over its contributing area (Fig. 14). The groundwater wells promptly transition from diel
cycle index values of ≈+1 (snowmelt) to ≈-1 (evapotranspiration) roughly simultaneous with the disappearance of the
snowpack at the altitude of the B transect (indicated by the first of the two vertical dashed lines each year). The daily cycle
amplitudes, shown in gray in Fig. 14, are very small for several days during this transition, consistent with the destructive
interference described in Fig. 13. Simultaneous with this abrupt transition in groundwater cycles, the stream's diel cycle
index begins a gradual two-month transition toward evapotranspiration-dominated values, which only becomes complete
when the snowpack disappears at the top of the basin (indicated by the second of the two vertical dashed lines each year).
This gradual shift in dominance from snowmelt to evapotranspiration presumably reflects the gradual retreat of snow cover
toward the top of the basin, and the corresponding gradual advance of active photosynthesis and transpiration as the
snowpack vanishes.

This conceptual model is supported by the timing of diel cycle index transitions in the gauged subcatchments as well. As
Fig. 15 shows, the transition from snowmelt-dominated cycles toward evapotranspiration-dominated cycles takes place later
at the higher-elevation gauges at Sagehen Creek. At the lower culvert, the diel cycle index shifts from positive to negative in
late May, for example, but the same transition does not take place at the upper culvert until mid-July (Fig. 15). The latest
transition of all is at Independence Creek. Although the gauge at Independence Creek is 300 m lower than the Upper
Culvert, a sizeable fraction of the Independence Creek basin is at substantially higher altitudes, and includes steep north-
facing slopes that hold snow relatively late into the summer, explaining the greater persistence of positive diel cycle index
values. One sees similar contrasts between north-facing and south-facing tributaries to Sagehen Creek (see Fig. S4). Kiln
Creek, which faces south, transitions from positive to negative diel cycle index values two to three weeks earlier than South
Tributaries 1 and 2, which span similar elevations but face north. Distinct snowmelt cycles also begin earlier, and peak
snowmelt discharge occurs about 3 weeks earlier, in south-facing Kiln Creek compared to the two north-facing tributaries
(Fig. S4).

In Figs. 14, 15, and S4, one can see that the evapotranspiration cycles in streamflow are often less distinct than those in
groundwater, where diel cycle index values approach -1. There are several possible explanations. First, in groundwater,
snowmelt and evapotranspiration cycles are often of roughly equal amplitude, but as the daily stage anomalies in Figs. 14,
15, and S4 show, evapotranspiration cycles in streamflow are often much smaller than snowmelt cycles are. This may reflect
the decreasing sensitivity of discharge to changes in storage as the catchment dries out (see Sect. 3.3). But whatever their
origins, the smaller-amplitude stream stage cycles in mid-summer are more vulnerable to confounding by measurement noise
than the larger-amplitude snowmelt cycles. Also, as streamflow declines, so will the kinematic wave speed at which
discharge cycles are propagated through the channel network, increasing the destructive interference between signals
generated close to the measurement point and far from it, and thus making the measured cycles less distinct (Wondzell et al.,
2007). Finally, if low flows delay the evapotranspiration cycles sufficiently, they may accumulate significant phase lags
relative to the solar flux that drives them, reducing their correlation with solar flux even if the lagged correlation remains
strong. One could, of course, expand the diel cycle index to include lagged correlations (e.g., Bond et al., 2002; Barnard et
al., 2010) as a means of detecting travel-time lags, but we have not done so here in the interests of simplicity, and also to
avoid "cherry-picking" correlations at the lags that make them strongest.

It is important to note that in larger catchments (e.g., Lundquist and Cayan, 2002; Lundquist and Dettinger, 2003), diel cycles may accumulate significant time lags in the channel network, leading to snowmelt cycles with diel cycle index values near zero (if the channel lag time is roughly 6 hours) or even diel cycle index values that are negative (if the channel lag time is roughly 12 hours). Thus, diel cycle index time series like Figs. 12, 14, and 15 should be interpreted with caution in larger catchments. Because channel lag times can lengthen during the summer as snowpacks retreat to higher elevations

(Lundquist and Cayan, 2002; Lundquist and Dettinger, 2003) and streamflows decline (leading to slower kinematic wave propagation speeds), snowmelt cycles alone could potentially result in diel cycle index values that gradually transition from positive to negative. Such scenarios can be distinguished from true snowmelt-evapotranspiration transitions by inspecting the stream stage time series themselves. In transitions from snowmelt to evapotranspiration cycles, such as those shown in Figs. 12, 14, and 15, the amplitude of the streamflow cycle will nearly vanish as the diel cycle index approaches zero. By

contrast, if the transition in the diel cycle index is caused by growing transmission lags of a snowmelt cycle, the streamflow cycle amplitude will not reach a minimum as the diel cycle index approaches zero. It may also be possible to distinguish snowmelt and evapotranspiration cycles based on their asymmetry rather than their phase (Lundquist and Cayan, 2002), with snowmelt cycles being characterized by rapid rises and gradual falls, and evapotranspiration cycles being characterized by rapid falls and gradual rises; we have not tested that approach here.

**3.7 Remote sensing evidence of snowpack retreat and expansion of photosynthetic activity**

During the late spring and early summer of 2006, images are available from Landsat 5 for five cloud-free intervals that illustrate the progressive retreat of the winter snowpack and the expansion of photosynthetic activity in the Sagehen and Independence basins (Fig. 16). The left-hand panels of Fig. 16 show the Normalized Difference Snow Index (NDSI), which compares the green and mid-infrared bands to identify snow (shown as white in these false-color images) by its much higher

reflectance in the visible spectrum than at mid-infrared wavelengths (Dozier, 1989; Riggs et al., 1994). The right-hand panels show the Normalized Difference Vegetation Index (NDVI), which compares the red and near-infrared bands, identifying photosynthetically active vegetation (shown as green in these images) by its higher reflectance in the near-infrared than in the visible spectrum (Tucker, 1979).

The spatial evolution of the snow and vegetation indices in Fig. 16 mirrors the temporal evolution of the diel cycle index values at the various gauging stations (Figs. 14, 15, and S4). In early April, when all three SNOTEL stations are near their peak snowpack accumulation (Fig. 2), almost the entire landscape is snow-covered and there is little evidence of photosynthetic activity (Fig. 16a,b). Shortly thereafter, the lowest-elevation stage recorders (the lower culvert and main gauge) begin to exhibit strong snowmelt cycles (Fig. 15), as the lowest elevations and south-facing slopes begin to melt. The

north side of the B transect melts out in early May (Fig. 16c,d), and the diel cycle index abruptly shifts to reflect evapotranspiration forcing, although the stream still reflects the ongoing snowmelt in the surrounding catchment (Fig. 14). By mid-May (Fig. 16e,f), south-facing Kiln Creek has melted out and exhibits evapotranspiration cycles (Fig. S4), but South Tributaries 1 and 2, which face north, still have significant snow cover and exhibit snowmelt cycles. By early June (Fig. 16g,h), only the middle culvert, upper culvert, and Independence Creek catchments have significant snow cover and retain a

strong snowmelt signature in their daily cycles. At the B transect stream, the main gauge, and the lower culvert, this snowmelt cycle has been largely obscured, or even reversed, by evapotranspiration cycles from the increasing vegetation activity at all but the highest elevations (Figs. 14 and 15). By early July, snow remains only in the steep terrain at the perimeter of the Independence Creek basin and the cirque above the upper culvert (Fig. 16i), and only Independence Creek's diel cycle index retains a clear snowmelt signature. The daily cycles in the upper culvert are disappearing as it dries up, and

the daily cycles in the other Sagehen stage recorders are showing increasing dominance by evapotranspiration (Figs. 14, 15, and S4). Thus from May through July, the daily cycles in streamflow reflect the gradual retreat of the snow-covered area to

the higher elevations (Fig. 16, left panels), and its replacement by a gradually expanding area of strong photosynthetic activity (Fig. 16, right panels).

The remote sensing images in Fig. 16 are visually compelling, but leave open the question of whether we can more quantitatively link the spatial dynamics of snow and vegetation to the daily cycling in streamflow. Landsat imagery provides high spatial resolution, but is available only every 8 or 16 days, and if an individual image is obscured by cloud cover, the gap between usable images becomes even wider. The MODIS (Moderate Resolution Imaging Spectroradiometer) satellites, by contrast, provide nearly daily coverage, but at only 500-m resolution. Therefore we extracted average values of MODIS

snow and vegetation indices for each of the catchments and subcatchments, to track their seasonal evolution in greater detail. We calculated NDSI, the Normalized Difference Snow Index, as (GREEN-SWIR)/(GREEN+SWIR), where GREEN is band 4 and SWIR (shortwave infrared) is band 6, directly from daily surface reflectance data from the MODIS Terra and Aqua satellites, file series MOD09GA and MYD09GA (level 2G-lite, collection 6: Vermote et al., 2015). Terra and Aqua are identical satellites on nearly identical orbits, but Terra passes over northern hemisphere mid-latitudes in the morning, and

Aqua passes over northern hemisphere mid-latitudes in the afternoon, so solar illumination of the surface differs between the two. Pixels with cloud cover in the surrounding 1 km square (or in any adjacent 1 km square) were excluded. For each day, we created a 7-day composite (from 3 days before the day in question to 3 days after) of non-excluded values at each pixel, and took the median of these 7 values. We then averaged these pixel median values over the drainage basins for each gauging station. Compared to the NDSI snow cover product provided in the file series MOD10A1 (Riggs et al., 2016), this

calculation yields much lower scatter. It also does not artificially clamp low-NDSI values to zero snow cover (as the MOD10A1 NDSI snow cover product does), and thus preserves greater sensitivity to partial snow cover in complex terrain.

We followed a similar approach in calculating EVI2, the two-band Enhanced Vegetation Index, 2.5·(NIR - RED)/(NIR + 2.4*RED + 1) (Jiang et al., 2008), directly from daily surface reflectance data from the MODIS terra and aqua satellites.

Here NIR (near-infrared) is MODIS band 2 and RED is MODIS band 1. EVI2 is designed to closely mimic the original 3-band Enhanced Vegetation Index (Huete et al., 2002), but with greater stability and less sensitivity to clouds and snow. Because EVI2 is relatively insensitive to thin cloud cover, and because we are interested primarily in its behavior during the summer when clouds are rare, we did not exclude cloudy pixels as we did with our NDSI calculations. Summertime EVI2 values were almost identical whether cloudy pixels were excluded or not, but temporal coverage was better when clouds

were not excluded. All remote sensing images were processed using Google Earth Engine. We then averaged the Terra and Aqua daily values for NDSI and EVI2, and interpolated them with a Loess robust local smoothing curve to average out short-term noise and fill in missing values (see Fig. S5 for examples).

Figure 17 shows the seasonal patterns in these MODIS snow and vegetation indices, superimposed on the diel cycle index

for four gauging stations at Sagehen and Independence Creeks. Note that the scale for the vegetation index, EVI2, is reversed because we expect greater vegetation activity (and thus higher EVI2 values) to be associated with negative values of the diel cycle index. The snow index (light blue curve) exhibits a marked decline during the snowmelt season, shortly before the early-summer shift from snowmelt-dominated streamflow cycles (diel cycle index ≈+1) to evapotranspiration-dominated cycles (diel cycle index ≈-1). In late summer, however, the evapotranspiration signal in streamflow weakens and

the diel cycle index returns to neutral or positive values, and this transition occurs several months before the seasonal snowpack begins to accumulate again. Thus, although the disappearance of the seasonal snowpack could plausibly explain the shift from snowmelt to evapotranspiration cycles in streamflow, the accumulation of the seasonal snowpack comes too late to explain the shift back toward snowmelt cycles. Instead, the seasonal strengthening and then weakening of the evapotranspiration cycles coincides more closely with the summertime increase and then decrease in photosynthetic activity,

as reflected in the vegetation index EVI2 (green curves in Figs. 17c-17f) and the seasonal rise and fall in sap flow rates (Fig. 17b).  The correspondence between the vegetation index and the diel cycle index is very close, particularly at Independence Creek and the Sagehen main gauge.  Note that the same scale was used for the vegetation index at all three nested Sagehen gauges, but a different scale was used for Independence Creek because the large fraction of bare rock in the Upper Independence basin (see Fig. S1) limits the vegetation index's summertime maximum.


The similarities between the summertime rise and fall in the vegetation index EVI2 (green curves in Figs. 17c-17f) and the strengthening and weakening of the evapotranspiration cycles in the stream stage measurements (dark blue dots in Figs 17c-17f) suggest a cause-effect relationship.  The mechanistic connection between these disparate measurements remains unclear, however.  Sap flow rates in the four monitored trees peak sharply in July, almost simultaneously with the peak in the

EVI2 vegetation index at Sagehen (at Independence, EVI2 peaks a few weeks later).  The sap flow measurements peak more sharply than the solar flux curve does (Fig. 17b), suggesting that other limiting factors are also at work.  Recent sap flow measurements on hillslopes at 2365 m in the Sagehen basin imply that early-season sap flow rates are limited by low temperatures, and late-summer sap flow rates are limited by low soil moisture (Cooper et al., 2020).  Soil moisture is unlikely to be limiting for our sap flow trees, however, since the water table remained close to the ground surface during the

entire period of our study.  Nonetheless, the sap flow rates in these trees decline to half of their peak values by mid-September (in the wet year, 2006) or mid-August (in the drier years of 2007 and 2008).  Although the sap flow rates in these trees decline markedly, the nearby groundwater wells continue to show ET-dominated cycles until at least October (Fig. 14). Indications of autumn snowmelt first appear in the streamflow's diel cycle index values in October and November, before the seasonal snowpack becomes established at the Sagehen and Independence catchments (Fig. 17c-17f).  Nonetheless, hourly

data from the Independence Lake and Independence Camp SNOTEL sites show intermittent early-afternoon increases in soil moisture at 5cm depth beginning in early October (data not shown), suggesting that transient snow accumulation and melt could potentially also explain the autumn onset of snowmelt cycles, before the seasonal snowpack becomes established.

The upland soils of the Sagehen basin become very dry several weeks after snowmelt ends, so it is likely that moisture

limitations are reflected in the seasonal decline in the MODIS vegetation index.  It seems unlikely, however, that transpiration rates in the uplands, far from the channel network, would be reflected in the daily streamflow cycles shown in Figs. 14, 15, and 17.  The groundwater wells, which are all situated near the valley axis relatively low in the basin, generally show clear evapotranspiration cycles (diel cycle index ≈-1) until October, when the streams have already lost their clear evapotranspiration signal and have reached neutral diel cycle index values.  The reason for this divergence between the

streamflow cycles and groundwater cycles remains unclear.  One possibility is that by late summer, stream flows have become so low, and thus the variations in stream stage have become so small, that any correlations with external drivers like solar flux become indistinct.  A further consideration is that if the relationship between storage and discharge is nonlinear (and thus the aquifer response time $\tau$ in Eq. (5) becomes larger under drier conditions), discharge will become progressively less sensitive to ET-driven storage changes as the catchment dries up (Fig. 9b), and any evapotranspiration signal in

streamflow will become weaker.  Finally, as streamflow declines, so will the kinematic wave celerity that controls how quickly discharge signals are transmitted downstream, increasing the phase lag (and thus decreasing the correlation with solar flux) at downstream gauging stations, and also creating greater potential for destructive interference as local signals are added to the stream from the riparian aquifer along the way.

**4. Summary and outlook**

The analysis presented here is based on a catchment-scale hydrological monitoring network comprising a weather station, 3 snow telemetry (SNOTEL) stations, 6 sap flow sensors, 12 stream stage recorders distributed along the main stem and selected tributaries, and 24 groundwater level recorders in two transects of shallow groundwater wells (Figs. 1 and 2). This array of instrumentation allowed us to quantitatively explore the linkages between solar forcing, snowmelt, sap flow, and daily cycles in riparian groundwater levels and stream stage.


From late spring through autumn, diurnal forcing by snowmelt and evapotranspiration (ET) generate measurable daily cycles in riparian groundwater and stream stage (Sect. 3.2; Figs. 3 and 4). Snowmelt and ET are both driven by the diurnal pulse of solar flux, but generate riparian groundwater cycles of opposite sign. Snowmelt adds a pulse of water to the riparian aquifer during daytime; thus snowmelt-driven daily cycles are characterized by groundwater levels and streamflow that rise

throughout the day and decline during the night. By contrast, ET extracts water from the riparian aquifer during daytime; thus ET-driven daily cycles are characterized by groundwater levels and streamflows that decline throughout the day and rise at night, as the riparian aquifer is recharged by groundwater seepage from the surrounding uplands (Fig. 5). Daily cycles in riparian groundwater levels are typically much larger than daily cycles in stream stage, and they increase in amplitude with distance from the stream, implying that groundwater cycles are driving streamflow cycles rather than the other way around

(Fig. 7).

Because the riparian aquifer integrates both additions from snowmelt and subtractions from ET over time (Eq. 4), peak groundwater levels and peak streamflow (or minima, in the case of ET) occur in the late afternoon or evening, as the riparian aquifer shifts between positive and negative flux balance, rather than mid-day when the rate of snowmelt or ET is the highest

(Fig.5). Maxima (or, for ET-driven cycles, minima) in streamflow thus lag the peak solar flux by several hours (Figs. 3 and 4). In a catchment as small as Sagehen, this is primarily a dynamical phase lag, rather than a transit-time delay (Fig. 8 and 9). Solar forcing and sap flow rates are closely synchronized with the rates of increase/decrease in groundwater levels and streamflows (Fig. 6), not with the groundwater levels and streamflows themselves.

A simple mass-balance model of the riparian aquifer (Sect. 3.3; Eqs. 4-5) reproduces the essential features of the relationship between daily cycles in solar flux, groundwater levels, and stream stages. (Because storage and discharge are proportional for any given aquifer response time $\tau$, the shapes of the daily cycles in Fig. 8 describe variations in both groundwater levels and streamflows.) During snowmelt, when groundwater levels are high and aquifer response times may be quite short, daily cycles in groundwater and stream stage are asymmetrical, as observed in Fig. 3 and modeled in Figs. 8b and 8e. Later in the

summer when groundwater levels are lower and aquifer response times are likely to be longer, both the observed and modeled cycles are more symmetrical (Figs. 6 and 8f-g). In both the model and the observations, water level maxima (during snowmelt) and minima (during ET cycles) occur near the end of the day, rather than at mid-day when solar forcing is greatest (Figs. 3, 4, 5c, 8b-d, and 9a). Likewise the corresponding water level minima (during snowmelt) and maxima (during ET cycles) occur early in the day, rather than near midnight, in both the model and the observations (Figs. 3, 4, 5a,

8b-d, and 9a). And in both the model and the observations, the maximum rate of groundwater rise (during snowmelt) or decline (during ET cycles) occurs near mid-day, and slightly precedes the peak in the solar flux or sap flow rates (Figs. 6, 5b, 8e-g, and 9a). Finally, the night-time trend in the rate of rise or fall in groundwater in the model (Figs. 5d and 8e-g) mirrors the night-time trend in the groundwater observations (Fig. 6). All of these features are both predicted by the model and observed in the measurements, suggesting that this simple model plausibly represents the major dynamics shaping daily

cycles in groundwater and streamflow.

This simple mass-balance model implies that the amplitude of streamflow cycles depends not only on the amplitude of sap flow or snowmelt forcing, but also on the response time $\tau$ of the riparian aquifer (Figs. 8b-g and 9b). Because this response time is typically unknown (and may also vary with catchment wetness: see Kirchner, 2009), the amplitude of daily streamflow cycles cannot be straightforwardly interpreted as a quantitative estimator of riparian ET rates. Conversely, unless this response time is shorter than about 0.2 days, the amplitude of daily cycles in shallow groundwater will quantitatively reflect daily cycles in ET rates (Fig. 9b), suggesting that groundwater fluctuations can be used to monitor evapotranspiration over time, if the specific yield of the aquifer can be estimated (e.g., Loheide, 2008). This conceptual model also applies to catchments that lack extensive riparian aquifer storage, as indicated by the striking similarities between daily streamflow cycles in the glacially scoured granitic Upper Independence basin and the groundwater-dominated Sagehen basin (Figs. 10 and 11).

As the snowpack shrinks and becomes patchy in late spring and early summer, diurnal snowmelt pulses become weaker while diurnal ET pulses become stronger, with daily cycles in streamflow and groundwater reflecting the net effects of these two drivers. The relative dominance of snowmelt vs. ET can be quantified by the diel cycle index, which measures the correlation between the solar flux and the rate of rise or fall in stream stage or groundwater level (Sect. 3.4; Fig. 12). The diel cycle index will be close to +1 and -1 when streamflow cycles are dominated by snowmelt and evapotranspiration, respectively. During the transition from snowmelt-dominated to ET-dominated cycles in streamflow, daily cycles in the stream will temporarily vanish (diel cycle index =0) when the signals from these two drivers cancel each other out (Sect. 3.5; Fig. 13).

During snowmelt, as the snowpack melts out at an individual location, the local groundwater shifts abruptly from snowmelt-dominated cycles to ET-dominated cycles (Sect. 3.6; Fig. 14), indicating that the local groundwater cycles mostly reflect the local balance between snowmelt and ET forcing. By contrast, the springtime transition in streamflow daily cycles takes months, beginning when the snowpack melts out near the stage recording station, and ending with melt-out at the highest elevations in the catchment or sub-catchment (Fig. 14, 15, S4). These transitions occur later where the snowpack persists longer, at higher elevations (Fig. 15) and in north-facing sub-basins (Fig. S4). During these transitions, one can observe snowmelt cycles in the stream's upper reaches simultaneously with ET cycles in its lower reaches, due to overprinting of the snowmelt signals by ET signals generated below the snow zone (Fig. 15).

These gradual transitions from snowmelt-dominated cycles to ET-dominated cycles reflect the gradual retreat of the snowpack to higher and higher altitudes, and the corresponding upward advance of photosynthetic activity in the basin. Sequences of Landsat images confirm both the pattern and general timing of these progressive shifts in snow cover and photosynthetic activity (Sect. 3.7; Fig. 16). The springtime shift in the diel cycle index follows, by several weeks, the springtime decline in the basin-averaged MODIS normalized difference snow index (blue lines in Fig. 17) as the snow-covered area in the basin contracts. However, the autumn shift in the diel cycle index away from ET-dominated values ($\approx$-1) toward neutral or positive values precedes, by several months, the late-autumn establishment of the seasonal snowpack (Fig. 17a) and the corresponding rise in the MODIS snow index. By contrast, the basin-averaged MODIS enhanced vegetation index (green lines in Fig. 17) rises and falls in close synchrony with the diel cycle index in both springtime and autumn, particularly at the Sagehen main gauge (Fig. 17d) and Independence Creek (Fig. 17f). This result demonstrates that the diel cycle index closely reflects seasonal patterns of vegetation activity during snow-free periods.

More broadly, the analysis presented here illustrates how streams and groundwaters can serve as "mirrors of the landscape", and in particular how streams can integrate ecohydrological signals over their drainage networks. Those signals are

particularly clear at Sagehen and Independence creeks, because of their Mediterranean climate and snow-dominated hydrologic regime. Nonetheless, the analysis presented here could be replicated, with modifications, in many different landscapes, including catchments that are not snow-dominated, or where snowmelt and the growing season do not overlap, as they do at our study sites. The primary measurements (time series of water levels in shallow wells and streams) are relatively straightforward and inexpensive to collect, and the focus on daily cycle timing rather than amplitude means that

stream stage data can be used directly; discharge rating curves are not required. In larger basins, one could use the kinematic wave equation to account for the celerity with which discharge fluctuations propagate downstream (see Sect. 3.3) in order to interpret the timing of daily stream stage cycles. However, it will be harder to account for the dispersion that results from daily cycle signals being added all along the channel, with different time lags to the gauging station; thus interpretation of the shape of daily streamflow cycles in large basins may be challenging. Despite such complications, comparisons of daily

water level cycles across contrasting landscapes and scales should help in further clarifying how hydrologic signals are transported and mixed across landscapes and down channel networks.

**Code and data availability**

An archive of the data underlying this study is available at https://doi.org/10.16904/envidat.155. This archive includes 30-minute time series of the weather variables, sap flow fluxes, groundwater levels, and stream stages, as well as daily time series of temperature, precipitation, and snow water equivalent at the SNOTEL stations, diel cycle index values for groundwater levels and stream stages, and MODIS normalized difference snow index (NDSI) and enhanced vegetation index (EVI2) values averaged over selected subcatchments. The archive also includes the Google Earth Engine scripts that were

used to extract the MODIS data.

**Author contributions**

JWK conceived and led the study. The hydrological field measurements were made by JWK, SEG, MS, and RO, and the sap flow measurements were led by JM. JWK analyzed all of the time series and remote sensing images, and developed the conceptual model. JWK and DP drafted the figures and wrote the paper. All authors discussed the results and contributed to

finalizing the paper.

**Competing interests**

The authors declare that they have no conflict of interest.

**Acknowledgments**

We thank the University of California, Berkeley for its support of Sagehen Creek Field Station and the Central Sierra Snow

Laboratory, the US Geological Survey for its support of streamflow measurements at Sagehen, and the USDA Natural Resources Conservation Service for its support of the SNOTEL measurement network. We thank Beth Boyer for the loan of several TruTrack stage recorders, and Sagehen Creek Field Station managers Jeff Brown and Faerthen Felix for logistical support and many helpful discussions. This study was partly supported by a California Water Resources Center grant to JWK. The sapflow measurements were supported by the National Science Foundation's SAHRA (Sustainability of Semi-

Arid Hydrology and Riparian Areas) Science and Technology Center.

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

**Tables**

Table 1. Elevations and drainage areas of the stream stage recorders at Sagehen and Independence Creeks.

| Stage recorder | Elevation (m a.s.l.) | Drainage area (km²) |
|---|---|---|
| Lower culvert | 1877 | 34.1 |
| Main gauge | 1929 | 27.6 |
| Stream at B transect | 1931 | 27.4 |
| Stream at D transect | 1976 | 17.4 |
| Middle culvert | 2061 | 4.7 |
| Upper culvert | 2447 | 0.3 |
| Kiln Creek | 2001 | 2.2 |
| South Trib. 1 | 2105 | 4.1 |
| South Trib. 2 | 2117 | 1.3 |
| Independence Creek IND-1 | 2134 | 7.7 |
| Independence Creek IND-2 | 2158 | 5.7 |
| Independence Creek IND-4 | 2207 | 4.8 |

Drainage area estimates are subject to up to 1 km² uncertainty for all Sagehen gauges downstream of the D transect, because the northern boundary of the lower basin is topographically indistinct. Elevations of gauges may also vary by several

meters, depending on the topographic data source that is used.

Table 2. Cumulative precipitation, maximum SWE, and average temperature recorded at the three SNOTEL stations for the water years 2005-2006, 2006-2007, and 2007-2008.

| SNOTEL station | Elevation (m a.s.l.) | Cumulative precipitation (mm/yr) | | | Maximum SWE (mm) | | | Average temperature (°C) | | |
|---|---|---|---|---|---|---|---|---|---|---|
| | | 2005-2006 | 2006-2007 | 2007-2008 | 2005-2006 | 2006-2007 | 2007-2008 | 2005-2006 | 2006-2007 | 2007-2008 |
| Independence Creek | 1968 | 1224 | 511 | 559 | 356 | 287 | 450 | 6.3 | 6.1 | 5.8 |
| Independence Camp | 2135 | 1168 | 518 | 505 | 526 | 305 | 490 | 5.9 | 6.1 | 5.7 |
| Independence Lake | 2546 | 1732 | 856 | 907 | 1694 | 772 | 945 | 5.3 | 5.7 | 5.2 |


Table 3.  Long-term average annual (water year) precipitation, SWE, and temperature recorded at the three SNOTEL stations.  The observation period is reported in brackets.

| SNOTEL station | Elevation (m a.s.l.) | Average annual precipitation (mm/yr) | Average annual maximum SWE (mm) | Average annual temperature (°C) |
|---|---|---|---|---|
| Independence Creek | 1968 | 831 (1981-2010) | 361 (1980-2014) | 5.3 (1991-2013) |
| Independence Camp | 2135 | 865 (1981-2010) | 468 (1978-2014) | 4.9 (1983-2013) |
| Independence Lake | 2546 | 1206 (1981-2010) | 1120 (1978-2014) | 4.6 (1995-2013) |


**Figures**

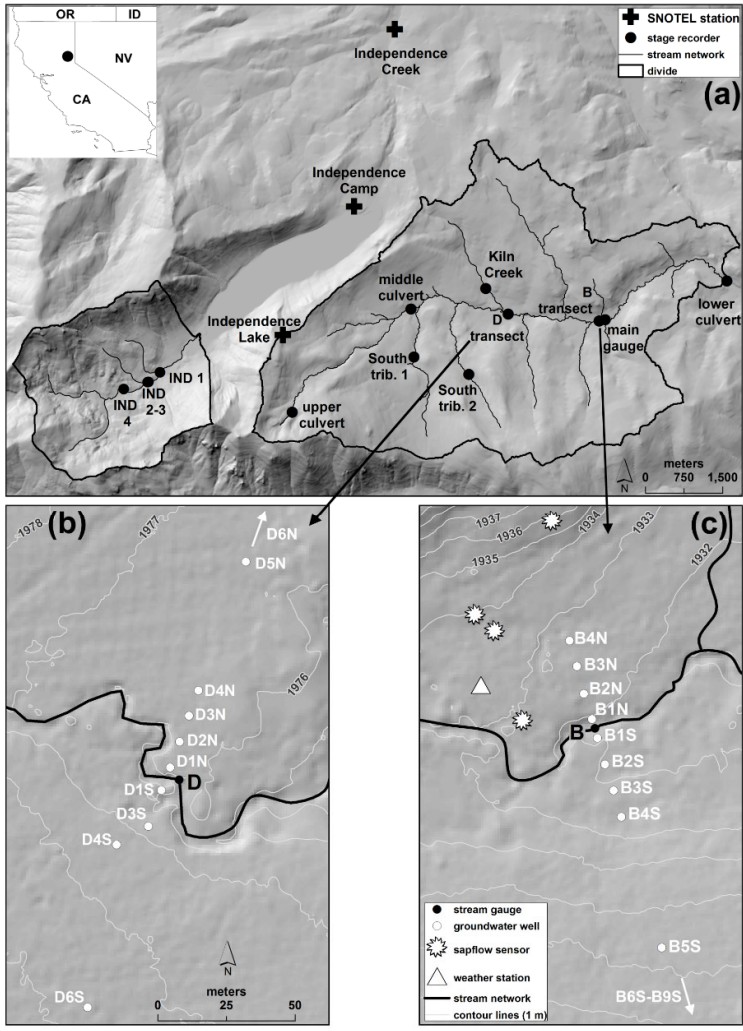

**Figure 1.  (a) Map of Sagehen Creek and Independence Creek catchments showing locations of stage recorders and SNOTEL stations, with inset map showing location in California.  (b) Map of the D transect of shallow groundwater wells.  (c) Map of the B transect of shallow groundwater wells, also showing locations of the weather station and the trees where sap flow was measured.**


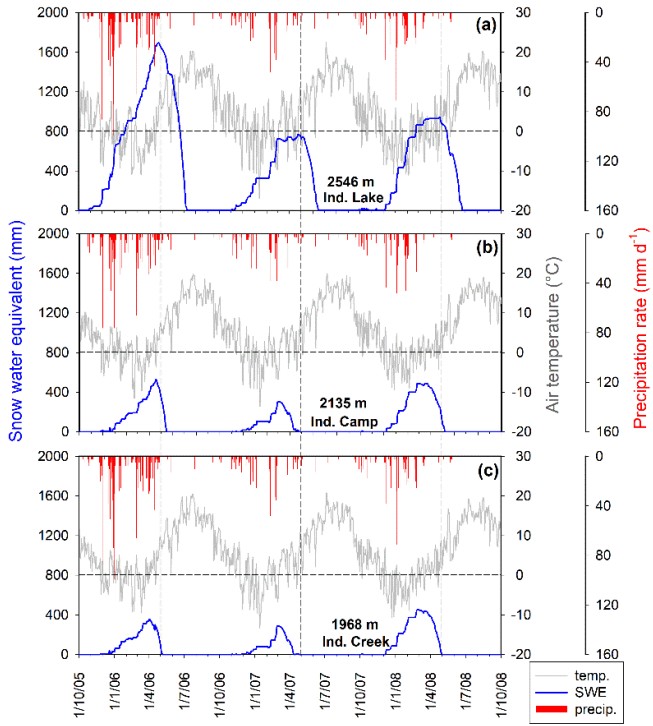

**Figure 2.** Daily time series of snow water equivalent (SWE), average air temperature, and precipitation at the three SNOTEL stations for the three water years 2005-6, 2006-7 and 2007-8. Vertical dashed lines indicate May 1st for all years. Horizontal dashed lines indicate 0°C. Seasonal snowpack volumes and melt timing vary substantially among the three SNOTEL stations, which span almost the entire elevation range of the Sagehen basin.


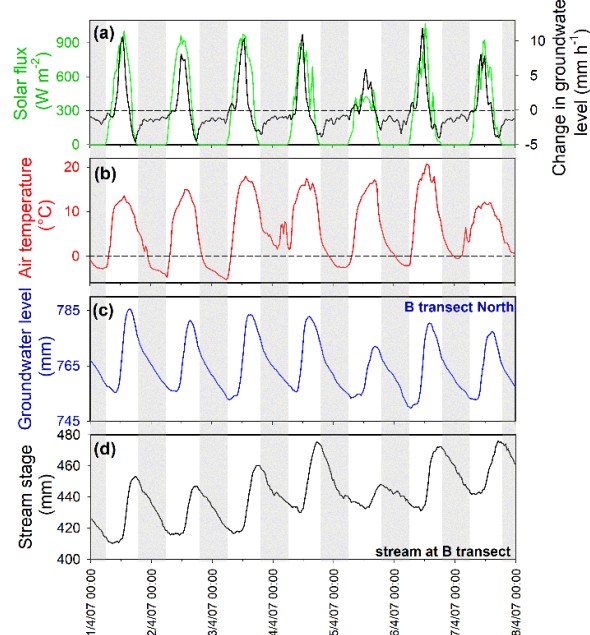

**Figure 3. Daily cycles in solar flux (a), air temperature (b), groundwater level (c), and stream stage (d) at the B transect during a snowmelt-dominated period in early April of 2007. Groundwater levels and stream stage are measured relative to arbitrary datum elevations. Vertical gray bars indicate hours between sunset and sunrise (in early April approximately between 19:00 and 06:00). Groundwater level is average of wells B1N, B2N, and B3N; well B4N records were lost due to data logger failure during this period. The black curve in panel (a) shows the rate of change in the groundwater level, which is tightly coupled to the solar flux. The mid-day peak in the solar flux (a) coincides with the greatest rate of increase in groundwater levels; the groundwater levels themselves (c) peak several hours later, in late afternoon, as the solar flux declines and the rate of change in groundwater level shifts from positive to negative. Groundwater levels (c) then decline throughout the night as the riparian aquifer continues to drain into the stream, reaching a minimum in mid-morning, when the solar flux again becomes intense enough that snowmelt exceeds the rate of riparian aquifer drainage, raising the rate of change in groundwater levels (a) above zero. Day-to-day variations in solar flux are reflected in the amplitude and timing of the daily cycles in both groundwater levels and stream stage.**



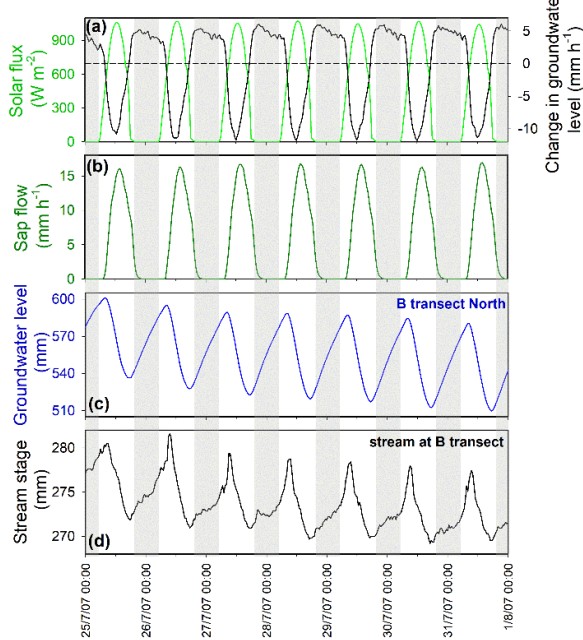

**Figure 4. Daily cycles in solar flux (a), sap flow (b), groundwater level (c), and stream stage (d) at the B transect during an evapotranspiration-dominated period in late July 2007. Groundwater levels and stream stage are measured relative to arbitrary datum elevations. Vertical gray bars indicate hours between sunset and sunrise (in late July approximately between 19:30 and 05:00). Groundwater level is the average of wells B1N, B2N, B3N, and B4N. The black curve in panel (a) shows the rate of change in the groundwater level, which is almost perfectly anti-correlated with the solar flux and sap flow. Groundwater level (c) and**
**stream stage (d) do not reach a minimum at mid-day, when solar flux and sap flow are highest, and thus groundwater levels are declining fastest (a). Instead, the groundwater level and stream stage reach their minimum in early evening, when solar flux and sap flow have decreased and the rate of change in groundwater levels crosses through zero (a). Groundwater level and stream stage then rise during the night (presumably in response to refilling of the riparian aquifer by groundwater drainage from upslope), reaching a peak in mid-morning when solar flux and sap flow rise enough to offset this groundwater influx, turning the**
**flux balance in the riparian zone (and thus the rate of change in groundwater levels) negative (a).**

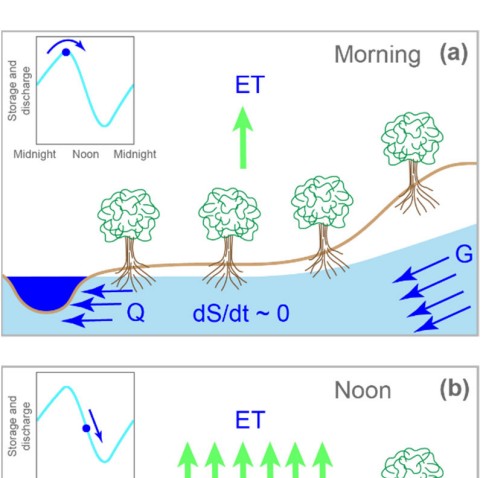

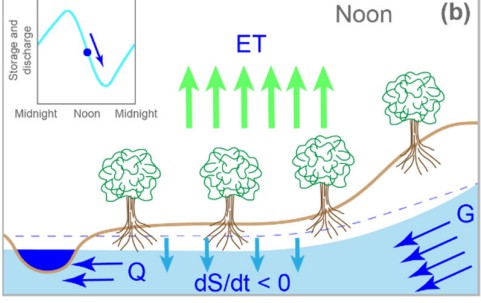

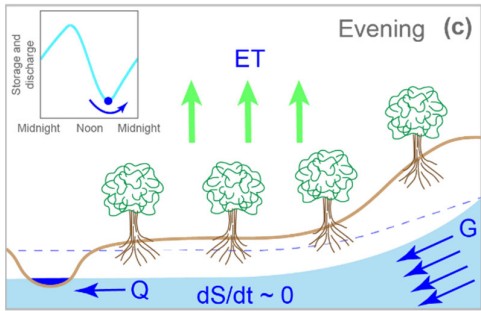

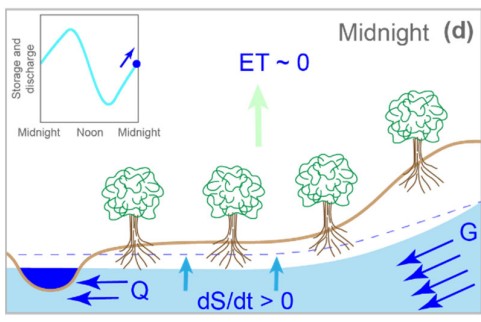

**Figure 5. Visualization of groundwater-stream coupling that leads to lagged evapotranspiration cycles in groundwater levels and streamflow (snowmelt cycles are similar but reversed). Streamflow is supplied by drainage from riparian groundwater, and this drainage rate is faster at higher levels of riparian groundwater storage (S). Riparian groundwater storage changes at a rate dS/dt that depends on the flux balance between streamflow (Q), evapotranspiration (ET), and groundwater recharge from surrounding uplands (G). The relative magnitudes of these fluxes in each panel are indicated by the number of arrows; upland recharge (G) is constant but the other fluxes vary from panel to panel. Inset figures show the corresponding phases of the daily cycle in streamflow and groundwater levels. In the morning (a), groundwater storage and streamflow reach their maximum and begin to decline as the evapotranspiration rate rises enough, relative to the difference between groundwater recharge and discharge, that the riparian aquifer reaches equilibrium and begins to decline. Around noon (b), high evapotranspiration fluxes lead to a strongly negative flux balance and a rapid draw-down of groundwater storage, and thus a rapid decline in streamflow (the dashed line indicates the morning high-stand of groundwater levels and stream stage, as a reference). Toward evening (c), riparian groundwater and stream stage reach their minimum and begin to rise when evapotranspiration rates and streamflows decline enough that the riparian aquifer reaches equilibrium and begins to refill. During the night (d) riparian groundwater levels (and thus stream stages) slowly rebound, because evapotranspiration is nearly zero and upland recharge exceeds stream discharge.**

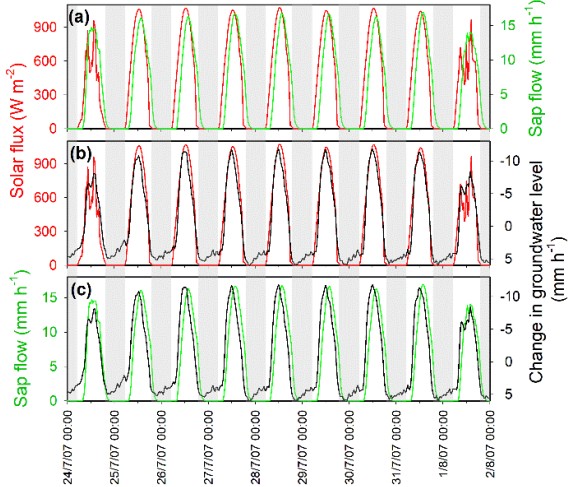

**Figure 6.** Daily cycles in solar flux (red), sap flow (green), and change in groundwater level (black) at transect B North (average of wells B1N, B2N, B3N and B4N) for ten days in mid-summer 2007. Note that the scale of the change in groundwater level is inverted, such that peaks correspond to maximum rates of decrease in groundwater levels. Rates of decline in groundwater levels are very closely synchronized with solar flux (b) and sap flow (c). Peak rates of decline in groundwater levels slightly precede the peaks in sap flow (c), consistent with model predictions (see Figs. 8 and 9). Variations in rates of groundwater rise and fall reflect day-to-day, and even sub-daily, variations in solar flux and sap flow.

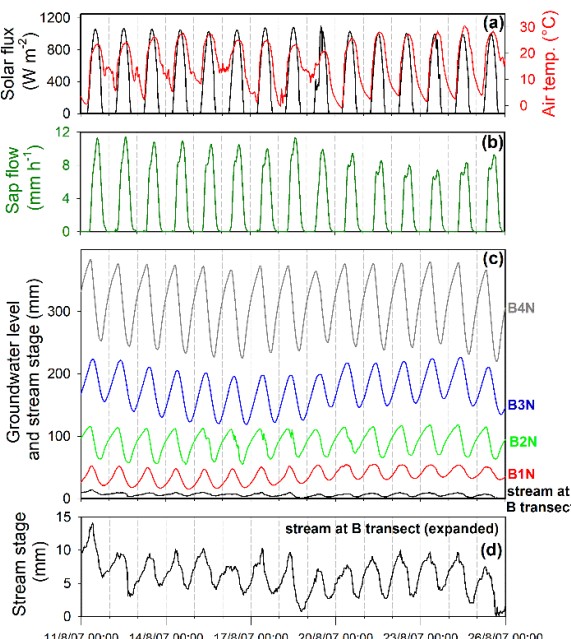

**Figure 7.** Time series of solar flux (a), air temperature (a), sap flow (b), groundwater levels in the four wells that comprise the northern B transect (c), and water level in Sagehen Creek adjacent to the B transect (d), for 15 days in August 2007. Vertical lines indicate midnight. The raw water level data have been shifted vertically to accommodate all the time series in panel (c); real-world groundwater elevations in B1N, B2N, B3N, and B4N averaged roughly 200, 1050, 1080, and 950 mm above the water level in Sagehen Creek, respectively, during the period shown here. Daily cycle amplitudes decrease in the order [B4N > B3N > B2N > B1N > stream] as one approaches the channel, indicating that daily cycles in groundwater are driving cycles in streamflow, rather than vice versa.

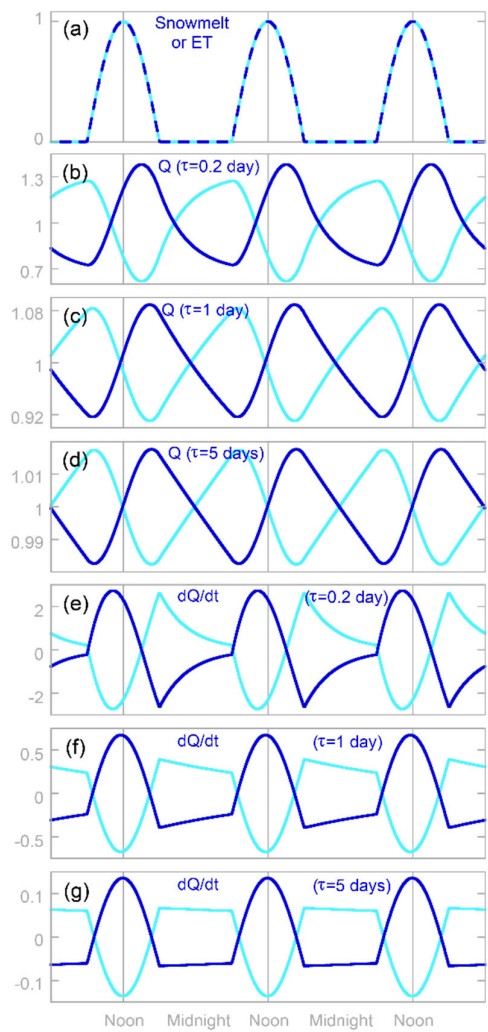

**Figure 8. Hypothetical daily pulses of snowmelt or evapotranspiration (a) and resulting daily cycles in discharge $Q$ (b-d) and rate of change in discharge $dQ/dt$ (e-g), calculated by integrating Eq. (5) for different values of the riparian storage response time $\tau$. Daily cycles driven by snowmelt and evapotranspiration are shown in dark and light blue, respectively. Note the differences in the vertical axis scales; daily cycles are markedly smaller for larger values of $\tau$. Temporal patterns in riparian storage are identical to those in discharge, because discharge is proportional to storage. Discharge maxima (or minima for ET) come 3-5 hours after noon (b-d), and discharge minima (or maxima for ET) come 6-7 hours after midnight (b-d), because riparian storage integrates water additions from snowmelt (or removals by ET) over time. These are not travel-time lags, because Eq. (5) does not simulate transport and its associated delays. Instead, these lags arise simply because changes in riparian storage accumulate over time. Peaks in $dQ/dt$ (or minima for ET) come slightly before the daily peak in snowmelt or ET (e-g), reflecting the change in the aquifer's drainage rate to streamflow as aquifer storage increases (or, under the influence of ET, decreases) as the day progresses. Nonetheless, these maxima or minima in $dQ/dt$ occur within 30 minutes of peak solar flux for $\tau \geq 1$ day (f-g), indicating that they are not greatly time-shifted by the typical dynamics of riparian storage.**

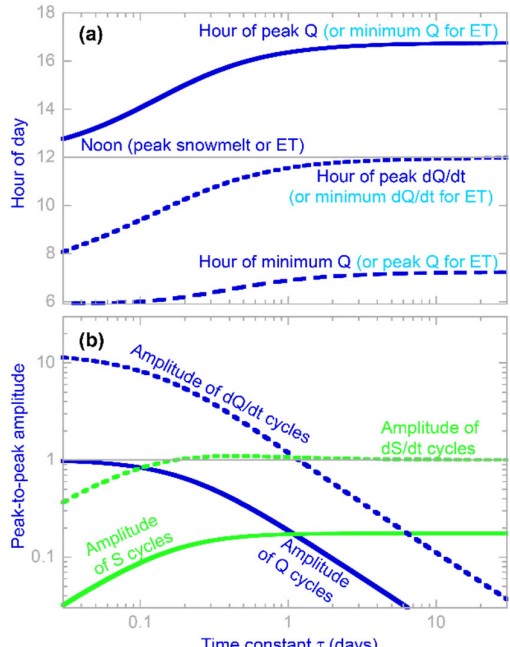


**Figure 9.** Timing (a) and amplitude (b) of daily cycles in streamflow ($Q$) and riparian aquifer storage ($S$) in response to daily cycles in evapotranspiration (ET) or snowmelt, as predicted by Eq. (5) across a 1000-fold range of the riparian aquifer response time $\tau$. In Eq. (5), changes in discharge and storage are proportional to one another; therefore their cycles obey the same timing and would plot identically in panel (a). Peak discharge (or minimum discharge for ET cycles) occurs in mid- to late afternoon,

rather than noon, and minimum discharge (or peak discharge for ET cycles) occurs in early morning, rather than midnight (see panel a), despite the fact that Eq. (5) includes no transport processes or transport delays. These apparent travel time lags are instead dynamical phase lags, created by the riparian storage integrating its inputs and outputs over time. Thus these lags are determined by the characteristic response time $\tau$ of the aquifer, the period of the cyclic forcing (1 day), and the shape of the forcing cycle, rather than by transport distances or kinematic wave velocities. For aquifer response times of $\tau \approx 1$ day or longer,

the peak in the rate of change of discharge closely coincides with the peak in the snowmelt or ET forcing. Panel (b) shows peak-to-peak amplitudes of daily cycles compared to the amplitude of the ET or snowmelt forcing (=1 on this scale). The amplitudes of cycles in discharge (and in the rate of change in discharge) are strongly dependent on the aquifer response time $\tau$ unless $\tau$ is much less than 1 day. Conversely, for aquifer response times larger than about 0.2 days, the amplitude of the cycle in the rate of change in storage ($dS/dt$) closely resembles the amplitude in the snowmelt or ET forcing.


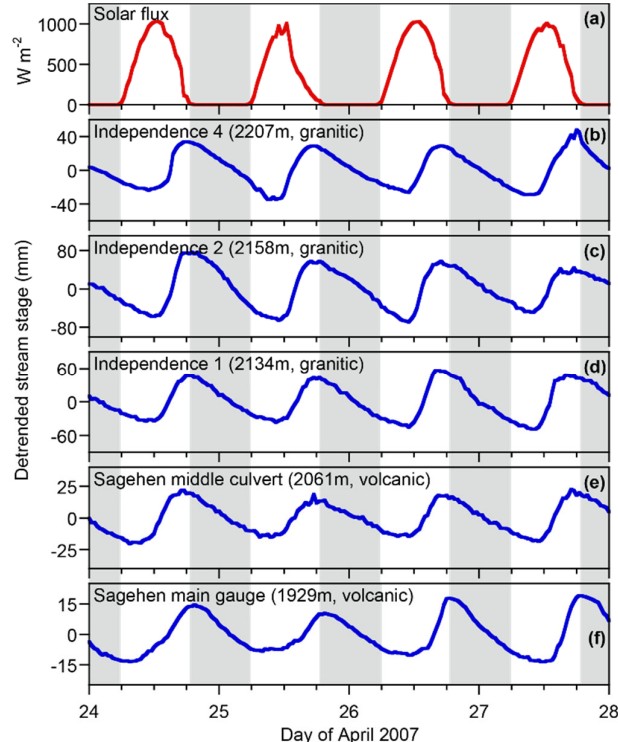

**Figure 10. Snowmelt-driven daily cycles in stream water levels measured in April 2007 at three locations along Upper Independence Creek, underlain by glaciated granodiorites, and two locations along Sagehen Creek, underlain by thick volcanic and volcaniclastic deposits. Stream stages were detrended using Eq. (3). The shapes and phases of the daily cycles are similar, and all exhibit similar lags relative to the solar forcing, despite the marked geological differences between the two catchments.**


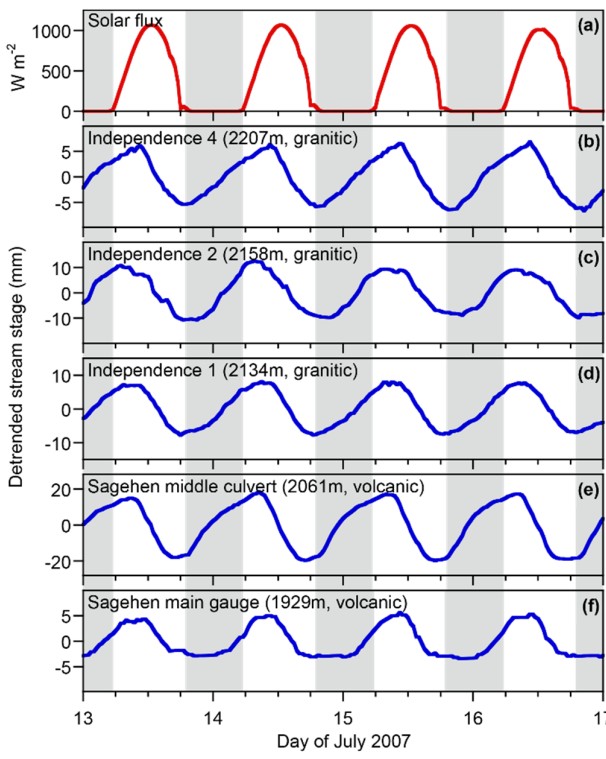

**Figure 11. Evapotranspiration-driven daily cycles in stream water levels measured in July 2007 at three locations along Upper Independence Creek, underlain by glaciated granodiorites, and two locations along Sagehen Creek, underlain by thick volcanic and volcaniclastic deposits. Stream stages were detrended using Eq. (3). The shapes and phases of the daily cycles are similar, and all exhibit similar lags relative to the solar forcing, despite the marked geological differences between the two catchments.**


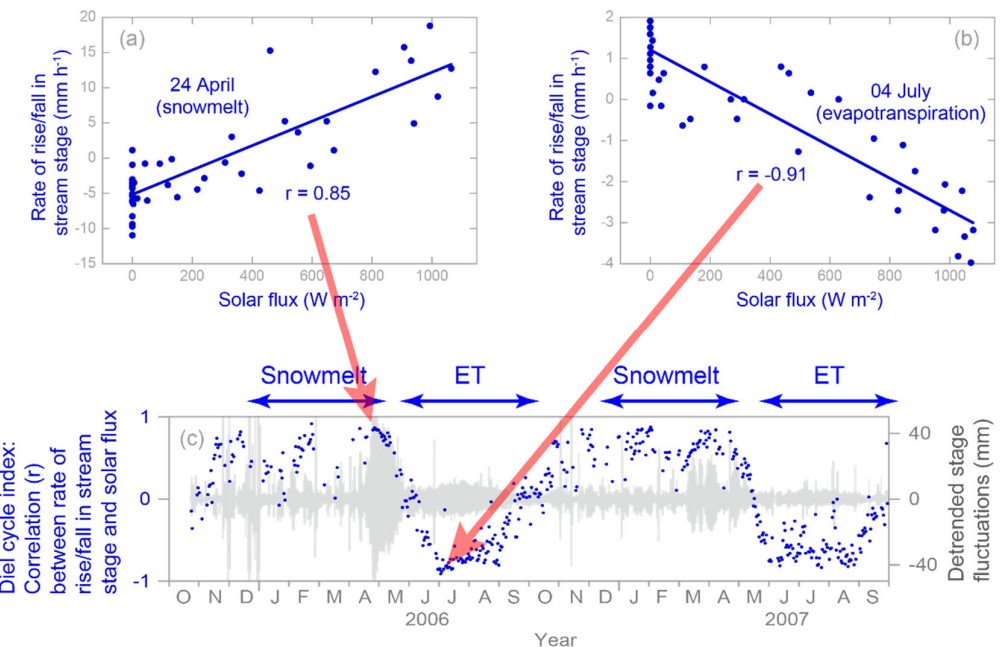

Figure 12. Correlations between solar flux and rates of rise and fall of water levels (Sagehen Creek, B transect) during two example days, one when the catchment was snow-covered and the stream exhibited a strong snowmelt cycle (24 April 2007), and another when the catchment was snow-free and the stream exhibited a strong evapotranspiration cycle (4 June 2007). In the lower plot, the correlation coefficients (blue dots) for each day indicate the relative dominance of snowmelt or evapotranspiration as generators of daily cycles in Sagehen Creek, while the gray shading shows the amplitude of the detrended daily stage fluctuations.


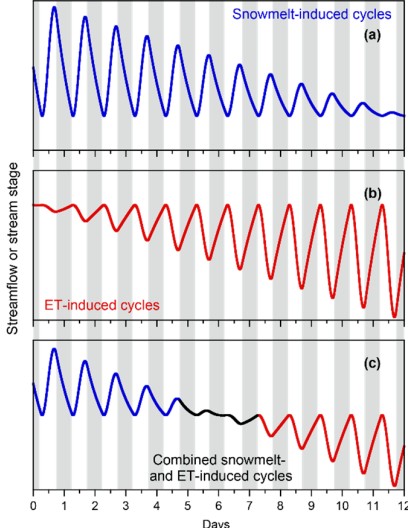

Figure 13. Mixing of hypothetical snowmelt (a) and evapotranspiration (b) cycles in streamflow or groundwater (c) during a transition period between late spring and early summer. Major ticks correspond to midnight. Vertical gray bars indicate night-
time hours between 18:00 and 06:00.

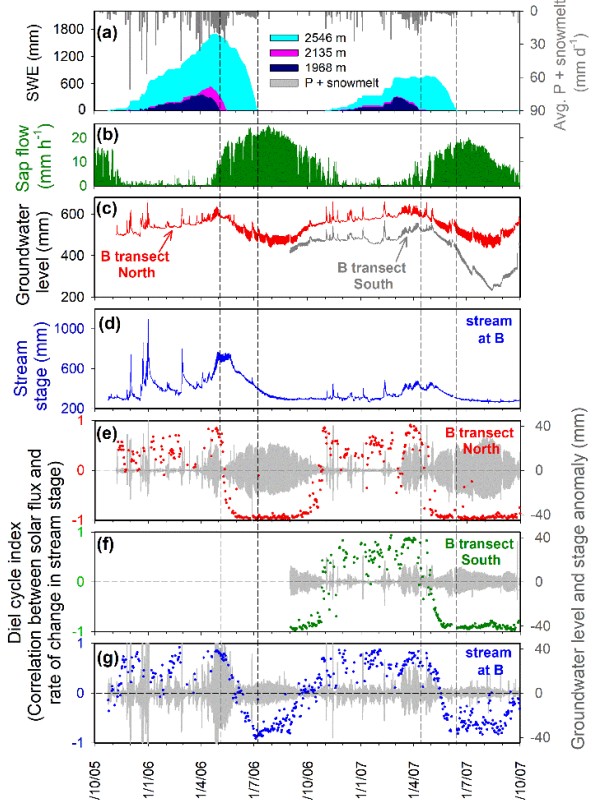

**Figure 14. Seasonal transitions between snowmelt- and evapotranspiration-dominated daily cycles in groundwater levels and stream stage at Sagehen Creek transect B. Daily time series of average net water input (liquid precipitation plus snowmelt) and SWE at the three SNOTEL stations (a) are shown with time series of sap flow (b), average groundwater levels (c) at the B transect North (average of wells B1N, B2N, B3N) and B transect South (average of wells B1S, B2S, B6S, B7S, B8S, B9S), and stream stage at the B transect (d) for two water years (2005-6 and 2006-7). Bottom panels show detrended fluctuations in groundwater (e-f) and stream stage (g) in gray, overlain by dots showing each day's diel cycle index (the correlation between solar flux and rise in groundwater level and stream stage; see Fig. 12). Snowmelt cycles generate positive correlations, and ET cycles generate negative correlations. The first vertical dashed line each year marks the date that the snowpack melts out at the lowest SNOTEL station (at 1968 m, close to the altitude of the B transect). The second vertical dashed line each year marks the date that the snowpack melts out at the highest SNOTEL station (at 2546 m, near the top of the Sagehen basin). The horizontal dashed lines in panels (e)-(g) indicate correlation coefficients of zero. Almost simultaneously with melt-out at the lowest SNOTEL site, groundwater at B transect North (e) shifts abruptly from snowmelt-dominated cycles (diel cycle index ≈+1) to evapotranspiration-dominated cycles (diel cycle index ≈-1). Groundwater at B transect South (f) shifts somewhat more gradually, because that side of the valley faces north and retains patches of snow somewhat longer. Simultaneously with the melt-out at the lowest SNOTEL site and the shift in groundwater daily cycling at the B transect, daily cycles in Sagehen Creek (g) begin a gradual two-month transition from snowmelt- to evapotranspiration-dominated daily cycles, as the snow-covered area shrinks toward the top of the basin. The transition to evapotranspiration-dominated daily cycles becomes complete almost simultaneously with melt-out of the snowpack at the highest SNOTEL site (indicated by the second dashed line each year).**

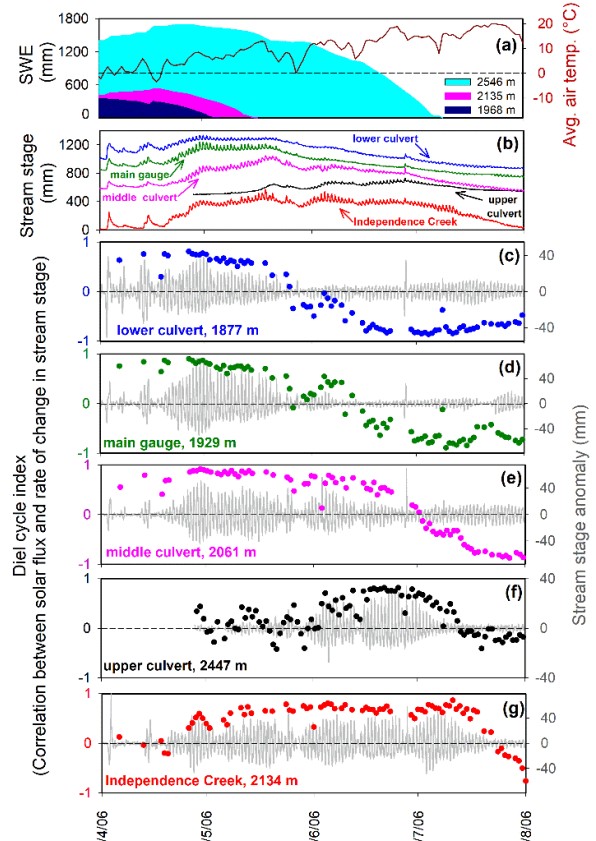


**Figure 15. Transition from snowmelt to evapotranspiration-dominated daily cycles in summer 2006 at several locations along Sagehen Creek and at Independence Creek gauge IND-01. Trends in snow water equivalent at the three SNOTEL stations (a) document the timing of melt-out across a range of altitudes. Stream stage time series (b) show streamflow response to seasonal melt patterns (each stage record has an arbitrary datum). Panels (c)-(g) show detrended daily stream stage fluctuations at**
**individual measurement stations (gray lines), and daily correlations between solar flux and the rate of change in stream stage (colored dots). Correlations near 1 and -1 indicate snowmelt- and evapotranspiration-dominated daily cycles, respectively. The transition from snowmelt- to evapotranspiration-dominated cycles occurs later and faster at higher elevations, consistent with the progression of melt-out from lower to higher altitudes. At Independence Creek, the snowmelt cycle lasts relatively longer and ends relatively later, reflecting the larger fraction of higher elevations at the Independence Creek basin, and particularly the steep**
**north-facing slopes that hold snow relatively late into the summer.**

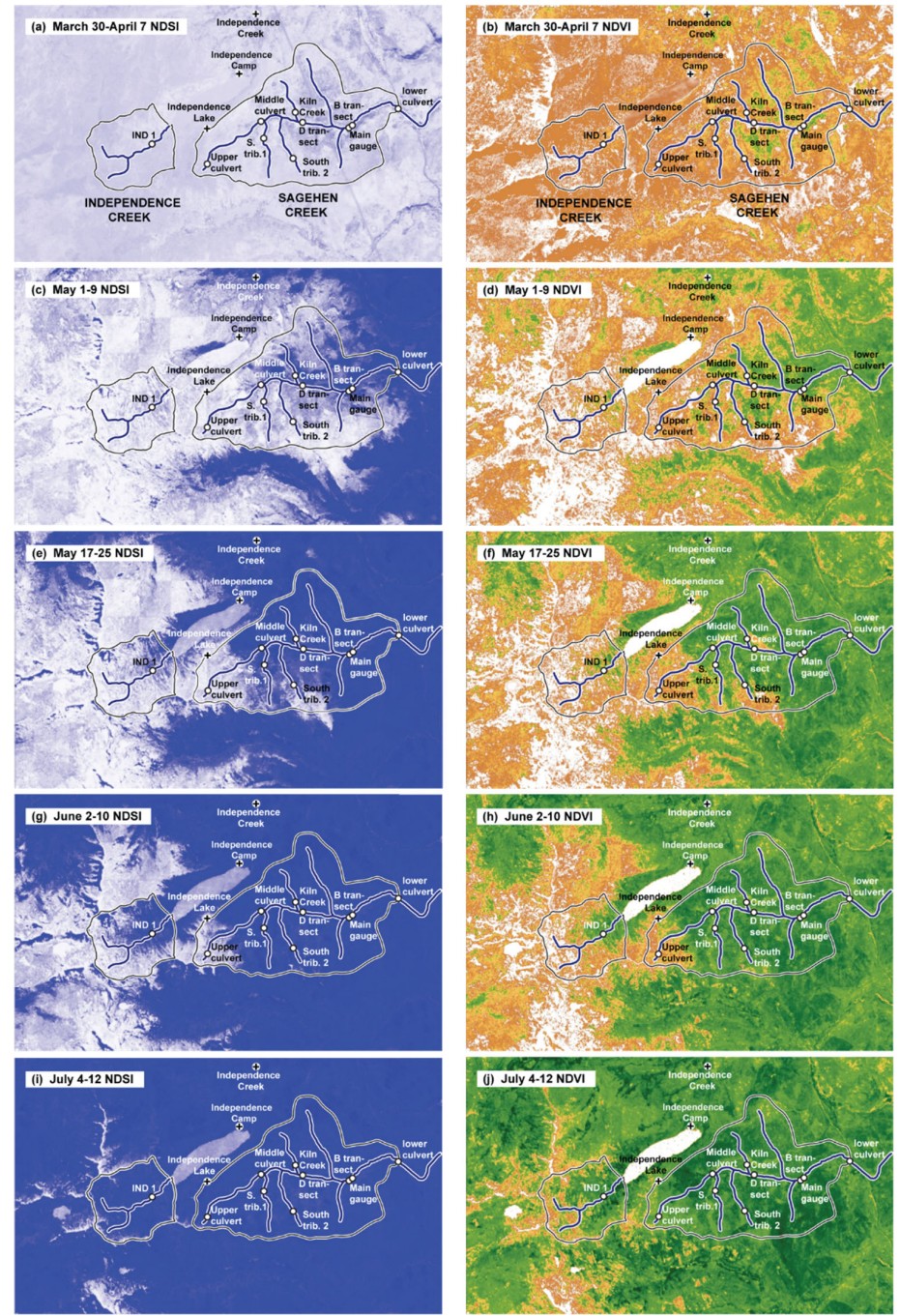

**Figure 16.** Evolution of snow cover and photosynthetic vegetation at Sagehen and Independence Creeks during spring and summer 2006, as visualized by Landsat 5 Thematic Mapper NDSI (Normalized Difference Snow Index) and NDVI (Normalized Difference Vegetation Index) 8-day composites. In the left-hand plots, white and dark blue indicate high and low snow index values, respectively. In the right-hand plots, high values of the vegetation index are indicated by dark green and low values are indicated by orange and white. Almost the entire landscape is snow-covered in early April (panels a and b), when peak snowpack accumulation is recorded at the three SNOTEL stations (Fig. 2), and shortly before the lowest stage recorders (lower culvert and main gauge) begin to exhibit strong snowmelt cycles (Fig. 15). As the melt season progresses, the lower elevations and south-facing slopes melt out quickest. In early May (panels c and d), the north side of the B transect melts out, leading to an abrupt shift to evapotranspiration-driven cycles in groundwater (Fig. 14), although the adjacent stream stage recorder still shows strong snowmelt cycles due to contributions from snowmelt upstream. From early May to early July, the snow-covered area gradually retreats to the higher elevations and is replaced by a gradually expanding area of strong photosynthetic activity (panels c through j). This is consistent with the gradual transition from snowmelt cycles to evapotranspiration cycles in stream stage at the B transect, the main gauge, the middle culvert, and the lower culvert (Figs. 14 and 15). South-facing Kiln Creek melts out by mid-May (panels e and f), but South Tributaries 1 and 2, which face north, do not melt out until early June (panels g and h), consistent with the daily stream stage cycles shown in Fig. S4. By early June (panels g and h), only the upper Sagehen catchment is still snow-covered, and the snowmelt cycle is offset by evapotranspiration-driven cycles at the lowest stage recorders (Fig. 15). By early July (panels i and j), only the uppermost ridgelines retain snow cover, the snowmelt cycle is disappearing at all of the stage recorders, and the B transect, main gauge, and lower culvert stage recorders show evapotranspiration-dominated cycles, with the middle culvert following by mid-July (Figs. 14 and 15).

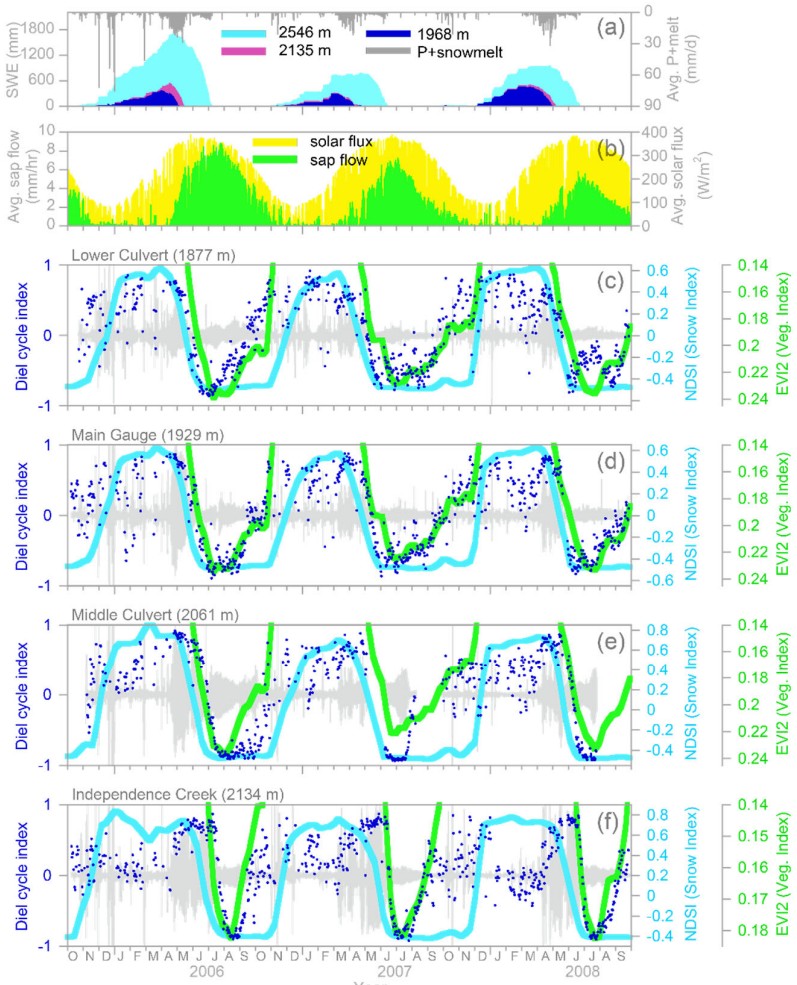

**Figure 17.** Seasonal patterns in the diel cycle index (correlation between solar flux and rate of change in water level) at Independence Creek and three nested catchments at Sagehen Creek (c-f), compared to seasonal patterns in daily MODIS NDSI (Normalized Difference Snow Index, blue curves), MODIS EVI2 (two-band Enhanced Vegetation Index, green curves), snowpack accumulation and melt at three SNOTEL stations (a), and daily average sap flow in B transect trees (b). The early-summer shift from snowmelt-dominated streamflow cycles (diel cycle index ≈ +1) to evapotranspiration-dominated cycles (diel cycle index ≈ -1) coincides with the retreat of the seasonal snowpack (a), an increase in evapotranspiration rates (b), a decrease in the MODIS snow index (blue curves), and an increase in the MODIS vegetation index (green curves, note reversed scale). The late-summer and autumn shift from evapotranspiration-dominated cycles toward snowmelt-dominated cycles precedes snowpack accumulation (a) and the seasonal increase in the MODIS snow index by several months, but coincides with a decline in sap flow rates (b) and a decrease in the MODIS vegetation index. The MODIS snow index and vegetation index curves are Loess robust smoothing fits to daily MODIS data averaged over the contributing area to each gauging station (see text and Fig. S5). The vegetation index scale is inverted to better show its relationship with the diel cycle index, and values below 0.14 are not shown. Gray shading shows daily stream stage anomalies (deviations from daily running means). Inter-annual decreases in sap flow measurements are artifacts of wound healing around the sap flow sensors (see text).