# Peer review of "The pulse of a montane ecosystem: coupling between daily cycles in solar flux, snowmelt, transpiration, groundwater, and streamflow at Sagehen and Independence Creeks, Sierra Nevada, USA"

_Hydrology and Earth System Sciences, 2020_

## Referee Comment (RC1) · Anonymous Referee #1 · 15 Mar 2020

The manuscript presents a thorough study of the daily cycles of different hydrological variables during rainless periods in different seasons, reflecting diurnal extraction of shallow groundwater by evapotranspiration and diurnal additions of meltwater during snowmelt. Basis of the study is extensive dataset of diurnal cycles of stream water level, groundwater levels, sap flow measurements, snow characteristics (snow water equivalent) and several other hydrometeorological conditions in the studied catchments (two snow-dominated headwater catchments in California's Sierra Nevada mountains).

In the introduction section authors clearly present the basis of the topic. They explain

two ż̇contrastingⁿ concepts, namely, the water table fluctuations (WTF) and missing streamflow approach, which have been traditionally used for analysis of the observed daily cycles. Furthermore, the question of time lags between the daily minimum and maximum ET and snowmelt rates vs. daily stream discharge and groundwater levels cycles is thoroughly explained and gives a reader (even those without strong theoretical background) good insight into the discussed topic.

The manuscript is long; however, I see no other option to present the discussed topic in a such holistic way. Namely, the authors have successfully covered different aspects of the research: (1) extensive field measurements; (2) theoretical background upgraded by a simple, but innovative conceptual model of the riparian groundwater mass balance and (3) presentation of the remote sensing data to support the concept. All 3 parts combined give excellent and holistic picture of the discussed topic.

By using stream levels data directly, authors presented a way how the problem of rating curve uncertainty can be avoided in analysis of daily cycles of various hydrological variables which are usually obtained through different hydrological measurements. This can be problematic especially in the range of extreme values (in this case during low-flow conditions).

In my view, the paper suits well in aims and scope of the HESS journal and I strongly support the publication of the manuscript. I would like to congratulate the authors for their excellent work. Bellow I provide some specific comments which are more or less technical.

Specific comments: Line 45: Maybe the role of specific sub-catchment characteristics (the low-land part of the catchment becomes ET dominated before the headwater (high-land part) could be more directly highlighted in the abstract. In the transition period, the temporal prevalence of snowmelt over the ET and vice-versa, strongly depend on the local topography characteristics. This aspect is thoroughly discussed in the manuscript.

[Figure]

Line 144: I suggest changing the sentence: . . .making these basins, in terms of climate characteristics, ideal. . .

Line 235: If I understand correctly, the "absolute" sap flow values were not directly used (the sap flow instruments were not calibrated) since the temporal changes (daily cycles) of sap flow were used in the study. Could lack of the calibration (drift) also influence the timing of the sap flow daily cycle?

Line 271: The reported data gaps occurred during the observed 3 years?

Line 279: Could the elevation bands limits centered on SNOTEL stations be also shown in Fig. 1 for illustration purposes?

Figure 6: Groundwater levels shown are absolute values? I would suggest slightly changing the Fig. 6 caption sentence (line 1060): . . .averaged roughly 200, 1050, 1080, and 950 mm above the Sagehen Creek water level. . .

Line 369: Is this correct? What could be an explanation for a "static water level of 0.6m above the ground surface"?

Lines 405-408: Could authors support their statement with measurements shown in Figs. 3 and 4? This would enable readers to understand the underlying processes more easily.

Line 422: What kind of water level? Groundwater, streamflow or some kind of "general water level" as an indicator of hydrological state of the catchment?

Line 435: What is "A" in the combined forcing $P + M + G - E = A\cos(wt)$?

Line 515: Were the average lag times for the two stations assessed from low flow discharge conditions (where the diel signal is evident) or using wider range of discharge conditions?

Figure 9c (lower plot): If the cloudy days and rainy periods were excluded form the dataset, how can detrended stage fluctuations data be continuous (or are they only

seemingly continuous)?

Lines 600-601: Since DEMs from Lidar data are available for studied catchments, could authors make their statement more tangible (quantifiable) in terms of the approx. % of slopes (catchment) facing north/south direction?

Lines 841-843: I would suggest mentioning also the problem related to the fact that diurnal cycles in stream water level in larger catchment (especially during the ET dominated periods) is pretty much undetectable (or unrecognizable) by the waters stage measuring equipment. Of course, this could be related to various lag times to the gauging stations as mentioned by the authors.

––––––––––––––––––––––––––––

---

## Author Comment (AC1) · 23 Mar 2020

**We appreciate Anonymous Referee #1's thoughtful comments on our manuscript. Below we respond (in bold type) to the referee's specific comments (in normal type).**

The manuscript presents a thorough study of the daily cycles of different hydrological variables during rainless periods in different seasons, reflecting diurnal extraction of shallow groundwater by evapotranspiration and diurnal additions of meltwater during snowmelt. Basis of the study is extensive dataset of diurnal cycles of stream water level, groundwater levels, sap flow measurements, snow characteristics (snow water equivalent) and several other hydrometeorological conditions in the studied catchments (two snow-dominated headwater catchments in California's Sierra Nevada mountains). In the introduction section authors clearly present the basis of the topic. They explain two contrasting concepts, namely, the water table fluctuations (WTF) and missing streamflow approach, which have been traditionally used for analysis of the observed daily cycles. Furthermore, the question of time lags between the daily minimum and maximum ET and snowmelt rates vs. daily stream discharge and groundwater levels cycles is thoroughly explained and gives a reader (even those without strong theoretical background) good insight into the discussed topic.

The manuscript is long; however, I see no other option to present the discussed topic in a such holistic way. Namely, the authors have successfully covered different aspects of the research: (1) extensive field measurements; (2) theoretical background upgraded by a simple, but innovative conceptual model of the riparian groundwater mass balance and (3) presentation of the remote sensing data to support the concept. All 3 parts combined give excellent and holistic picture of the discussed topic.

By using stream levels data directly, authors presented a way how the problem of rating curve uncertainty can be avoided in analysis of daily cycles of various hydrological variables which are usually obtained through different hydrological measurements. This can be problematic especially in the range of extreme values (in this case during low-flow conditions).

In my view, the paper suits well in aims and scope of the HESS journal and I strongly support the publication of the manuscript. I would like to congratulate the authors for their excellent work. Bellow I provide some specific comments which are more or less technical.

**We thank the referee for these supportive comments. We understand that the paper is long but, like the reviewer, we think it makes sense to present the whole analysis in one place.**

Specific comments: Line 45:

Maybe the role of specific sub-catchment characteristics (the low-land part of the catchment becomes ET dominated before the headwater (high-land part) could be more directly highlighted in the abstract. In the transition period, the temporal prevalence of snowmelt over the ET and vice-versa, strongly depend on the local topography characteristics. This aspect is thoroughly discussed in the manuscript.

**We will change the sentence, "Streamflow, however, integrates these transitions over the drainage network" to say instead, "Melt-out, and the corresponding shift in the diel cycle index, occur earlier at lower altitudes and on south-facing slopes, and streamflow integrates these transitions over the drainage network."**

Line 144: I suggest changing the sentence: …making these basins, in terms of climate characteristics, ideal…

**We will change this to more clearly express the link that needs to be made: "making it relatively easy to see how snowmelt and evapotranspiration are reflected in daily cycles in groundwater and streamflow."**

Line 235: If I understand correctly, the "absolute" sap flow values were not directly used (the sap flow instruments were not calibrated) since the temporal changes (daily cycles) of sap flow were used in the study. Could lack of the calibration (drift) also influence the timing of the sap flow daily cycle?

**Your understanding is correct. Changes in calibration (mostly due to wound healing in the trees) should affect the amplitude of the observed signals but should not introduce a measurable phase lag.**

Line 271: The reported data gaps occurred during the observed 3 years?

**Yes, in some cases. Some of the stage recorders had frequent failures, and since they were typically downloaded only a few times per year, the resulting data gaps are sometimes quite long. The results that are shown in the paper are obtained from the sites and time intervals without substantial data gaps.**

Line 279: Could the elevation bands limits centered on SNOTEL stations be also shown in Fig. 1 for illustration purposes?

**Yes, but this would tend to make it harder for readers to see other essential information in the figure. We will instead prepare a supplementary figure that shows these elevation bands, along with contour lines (some readers find that contour lines are easier to interpret than grayshade, or the other way around).**

Figure 6: Groundwater levels shown are absolute values? I would suggest slightly changing the Fig. 6 caption sentence (line 1060): …averaged roughly 200, 1050, 1080, and 950 mm above the Sagehen Creek water level…

**We will change the caption to make it clear that the quoted average elevations are above the water level in Sagehen Creek.**

Line 369: Is this correct? What could be an explanation for a "static water level of 0.6m above the ground surface"?

**Yes, this is correct. The explanation comes a few lines later: "These measurements also demonstrate an upward hydraulic gradient in the valley axis at all three locations, consistent with our hypothesis of groundwater flow from upslope recharging the riparian aquifer during mid-summer, thus sustaining both streamflow and plant water use." We will modify this statement so that it is clearer that by "groundwater flow" we mean bedrock fracture flow.**

Lines 405-408: Could authors support their statement with measurements shown in Figs. 3 and 4? This would enable readers to understand the underlying processes more easily.

**We will add the statement: "One can see this behavior in Figs. 3 and 4: groundwater levels change fastest near the peak of the solar flux (Figs. 3a and 4a), but the groundwater levels themselves reach their maxima (or, for ET cycles, minima) several hours later (Figs. 3c and 4c), when the rate of groundwater rise or fall changes sign."**

Line 422: What kind of water level? Groundwater, streamflow or some kind of "general water level" as an indicator of hydrological state of the catchment?

**Sorry, we mean riparian groundwater levels and will modify the text accordingly.**

Line 435: What is "A" in the combined forcing P + M + G – E = Acos(wt)?

**A is the amplitude of the assumed sinusoidal forcing. We will add this to the text.**

Line 515: Were the average lag times for the two stations assessed from low flow discharge conditions (where the diel signal is evident) or using wider range of discharge conditions?

**The calculations behind the statement in line 515 were made under average discharge conditions, but the results are not sensitive to the flow regime that is used. The cross-correlation results presented in line 510 are for all three years of data combined.**

Figure 9c (lower plot): If the cloudy days and rainy periods were excluded form the dataset, how can detrended stage fluctuations data be continuous (or are they only seemingly continuous)?

**Cloudy and rainy days were not excluded from the data set, but rather from the daily correlation analysis shown in Figs. 9a and 9b. That is why blue dots (indicating diel cycle index values) are missing for some of the days in Fig. 9c (and also for some days in Figs. 11, 12, 14, S3, and S4).**

Lines 600-601: Since DEMs from Lidar data are available for studied catchments, could authors make their statement more tangible (quantifiable) in terms of the approx. % of slopes (catchment) facing north/south direction?

**It is difficult to reliably estimate the catchment area that drains to the two well transects (since they are aligned roughly perpendicular to the stream, and thus parallel to the groundwater flowpath). Nonetheless, at the B transect the stream flows almost exactly from west to east. Thus almost all of the terrain north of the stream faces south, and almost all of the terrain faces north.**

Lines 841-843: I would suggest mentioning also the problem related to the fact that diurnal cycles in stream water level in larger catchment (especially during the ET dominated periods) is pretty much undetectable (or unrecognizable) by the waters stage measuring equipment. Of course, this could be related to various lag times to the gauging stations as mentioned by the authors.

**We are reluctant to say this because we present no evidence indicating that diurnal cycles are actually undetectable in larger catchments. It seems reasonable that this may be the case, but we would not want to assume it.**

---

## Referee Comment (RC2) · Anonymous Referee #2 · 28 Mar 2020

The manuscript presents a comprehensive study of the hydrological cycle in two montane catchments in the Sierra Nevada, USA. The analysis uses a large dataset to explain how groundwater and streamflow daily fluctuations are dynamically related to transpiration and snowmelt daily cycles forced by solar radiation. A simple and elegant model is used to explain these relationships.

As I understand, the main result of the study is to have identified that in small catchments the links between the daily fluctuations of streamflow (Q) and both transpiration (T) and snowmelt are mediated by the groundwater storage in the riparian zone.

[Figure]

Therefore, the lags appearing between the daily cycles of streamflow and their forcing variables are not due to travel times, but are associated with the dynamics of the whole system with groundwater acting as a buffer that dampens and delays the response of streamflow. This shows that methods to estimate T (or evapotranspiration, ET) using series of Q are not feasible unless characteristics of the riparian aquifer are also known.

Although the manuscript addresses a topic certainly interesting for the readers of HESS, I found it extremely difficult to read. The manuscript is very verbose and I often found myself lost in long explanations about concepts that were not really of interest or strictly relevant.

Therefore, my suggestions and detailed comments listed below are mainly directed to shorten and hopefully improve the readability of the manuscript.

- Title: already the title seems long. Could it be shortened into something like "The pulse of a montane ecosystem: relating daily cycles of hydrological variables".

- Abstract: this is also very long. I would try to shorten it to make the key messages of the study clear to readers.

- Line 31: "...transiently achieves mass balance." This is not clear to me. The mass balance should be always satisfied.

- L34: I would not use here "time constant", because that related to the simple model presented in Eq. 5, which assume $\tau$ to be constant to obtain an exact solution of the equation. However, as I understood reading the manuscript, the riparian aquifer might have a response time that is not constant.

- L90-91: is integro-differential the correct term here?

- L108: I believe that the WTF method as defined by White (1932) did not account for Q because the observations were done in a desertic environment where Q was not relevant.

- L129: Gribovszki

- Section 2.1: I would erase the pronunciation of the catchments and historical information that is not necessary to understand the analyses presented later on in the manuscript. I don't think information about potential evapotranspiration is provided for the catchments, and rainfall and temperature are not given for the Independence basin. It would be good if the description of the two catchments followed the same structure to facilitate the reading. L191-195 can be erased.

- Section 2.2: a lot of details can be removed (e.g., precise location of gages). A lot of this information is already in Fig. 1 (latitudes and longitudes could be reported in the figure or tables instead of the text). I would move the description of the sapflow measurements (L232-239) at the end of this section. At the moment, the description starts with weirs and bores, switches to sapflow, and then goes back to bores. L241-245 can be erased.

- L256: "To account for the combined..."

- L345-346: I do not think it is correct to say that solar radiation drives streamflow and groundwater fluctuations. There is an indirect relationship, as also stated at L557-558.

- L385-389: it is not really clear what an integro-differential system is in this context.

- L390-415: this part is rather long and it seems that is repeated more precisely after Eq. 5. I would just introduce Eq. 4, say that $Q$ is assumed to be a linear function of $S$ (i.e., $Q = f(S) = S/\tau$) and then write Eq. 5. I would avoid mentioning that the solution is well known (erase L434) and provide the solutions in Eq. 6. I think it should be better to say that it is assumed that the period considered is without $P$; I do not think it is reasonable to assume $P$ with a daily cycle as $M$, $G$, and $E$. Should there be a mention of the initial conditions for these solutions? I understand that the point is to look at cycles and the initial transient is not important; however, in Fig. 7, I found it strange that the initial values of Q were different.

[Figure]

- L460-479: I would erase this part. In most cases the inversion of the Fourier transform will be done numerically; therefore, one can just solve Eq. 5 numerically to start with. The point about the lags is clear from Eqs. 6 and their discussion.

- L501-504: the references to the lines in Fig. 8 do not seem correct.

- L572-573: I would erase this phrase.

- Subsection 3.5: I am not sure this is so important to deserve a full subsection.

- Subsection 3.7: I do not think this subsection is really necessary. I found that it was not adding much to what already presented and supported by the data. I would recommend to cut this part out.

- L724-725: the mismatch between the peaks in radiation and sapflow is not surprising. Vapor pressure deficit (VPD) is usually the variable that mostly drives transpiration, and I believe that VPD would likely explain the timing of the transpiration peak during the year (that's because there appears not to be water limitation).

- L773: I would use "changes in storage" instead of "mass balance".

- L779-781: erase?

- Figures: the captions of most figures are very long. Because the figures are explained in detail in the text, I would try to reduce the length of the captions, where a brief description of what the figures show should be enough.

- Figure 2: this figure is repeated in a different format in Figs. 11, 12, and 14. I would have these data in a single figure without repetitions.

- Figure 10: if Subsection 3.5 is reduced or removed, perhaps this figure can be removed as well.

-Figure 11: because sapflow and groundwater are related in this figure, I wonder whether it would be better to report the depth to the water table from the surface to

show that the water table is within reach of the root system. In the caption, it is said that signals were detrended but it is not explained how.

- I would consider removing Figs. 13 and 14 along with Subsection 3.7.

---

## Author Comment (AC2) · 29 Mar 2020

**We appreciate Anonymous Referee #2's comments on our manuscript. Below we respond (in bold type) to the referee's specific comments (in normal type).**

The manuscript presents a comprehensive study of the hydrological cycle in two montane catchments in the Sierra Nevada, USA. The analysis uses a large dataset to explain how groundwater and streamflow daily fluctuations are dynamically related to transpiration and snowmelt daily cycles forced by solar radiation. A simple and elegant model is used to explain these relationships.

As I understand, the main result of the study is to have identified that in small catchments the links between the daily fluctuations of streamflow (Q) and both transpiration (T) and snowmelt are mediated by the groundwater storage in the riparian zone.

Therefore, the lags appearing between the daily cycles of streamflow and their forcing variables are not due to travel times, but are associated with the dynamics of the whole system with groundwater acting as a buffer that dampens and delays the response of streamflow. This shows that methods to estimate T (or evapotranspiration, ET) using series of Q are not feasible unless characteristics of the riparian aquifer are also known.

Although the manuscript addresses a topic certainly interesting for the readers of HESS, I found it extremely difficult to read. The manuscript is very verbose and I often found myself lost in long explanations about concepts that were not really of interest or strictly relevant.

**We understand that the paper may appear long by contemporary standards, in which results from a project are often salami-sliced into as many different papers as possible. Instead of splitting our results into three or four separate papers, however, we have chosen to present them together because they are interconnected. In any case, the manuscript is not that long compared to some others in HESS. It is about the same length as Kirchner and Allen (2020, https://doi.org/10.5194/hess-24-17-2020), for example, and much shorter than Kirchner (2019, https://doi.org/10.5194/hess-23-303-2019).**

**We disagree with the characterization of the manuscript as "very verbose". Clarity and completeness often require explaining things rather than just asserting them. We also need to allow for the fact that individual readers will have stronger background in some areas than in others. We are aiming at readers who may have some background in one or another aspect of these systems, but may not be experts in all of the topics that we are covering. Readers' points of interest may also differ. Thus each reader may find some "concepts that [are] not really of interest or strictly relevant", but different readers will have different opinions on which specific concepts those are.**

Therefore, my suggestions and detailed comments listed below are mainly directed to shorten and hopefully improve the readability of the manuscript.

- Title: already the title seems long. Could it be shortened into something like "The pulse of a montane ecosystem: relating daily cycles of hydrological variables".

**We will think about this. Titles always represent a trade-off between the need to be explicit and the need to be concise. "Relating daily cycles of hydrological variables", in our view, is too cryptic (which variables? relating how?). There is also the need to cover the important keywords for search purposes, which is why we have identified the fluxes and storages (snowmelt, transpiration, groundwater, and streamflow), the study sites, and the region (Sierra Nevada).**

- Abstract: this is also very long. I would try to shorten it to make the key messages of the study clear to readers.

**We will see what we can do. The problem here is that the paper presents many interconnected results. These go well beyond the one that the reviewer has focused on (that riparian aquifer dynamics imply that we cannot infer ET rates from streamflow cycles). If that were the only punchline of the paper, the abstract could end at line 34. But the paper also shows that groundwater cycles reflect the relative dominance of snowmelt vs. ET, and streamflow cycles integrate these signals over the contributing catchment (lines 36-**

47).  We also show that temporal patterns of streamflow cycles are quantitatively consistent with the spatial evolution of snowmelt and ET during the transition from winter to summer and vice versa, as viewed from LANDSAT and MODIS (lines 49-54).  And for the abstract to be comprehensible to the reader, the results need to be (briefly) explained, not just asserted.  We will see what can be cut or condensed, but we don't want to sacrifice comprehension for the sake of brevity.

- Line 31: "...transiently achieves mass balance." This is not clear to me. The mass balance should be always satisfied.

**This depends on whether "mass balance" means "inputs balance outputs plus change in storage" (which is of course always satisfied), or "inputs balance outputs" (which is not always the case).  We meant the latter.  We can change "solar forcing declines enough that the riparian aquifer transiently achieves mass balance" to say instead "solar forcing declines enough that inputs transiently balance outputs in the riparian aquifer".**

- L34: I would not use here "time constant", because that related to the simple model presented in Eq. 5, which assume τ to be constant to obtain an exact solution of the equation. However, as I understood reading the manuscript, the riparian aquifer might have a response time that is not constant.

**That is correct.  Although "time constant" is sometimes used to refer to a characteristic response time that may not be strictly constant, "response time" would be better here.  We will change it.**

- L90-91: is integro-differential the correct term here?

**We are not referring to an integro-differential equation (that is, an equation with both integrals and derivatives).  But, as stated, there is an integro-differential relationship between groundwater levels and ET: groundwater storage integrates ET, and ET is the derivative of groundwater storage, assuming other fluxes are trivial.  Integrals introduce 90-degree phase lags, hence we should expect a roughly 6-hour phase lag between daily ET or snowmelt cycles and the resulting groundwater cycles (again, assuming there are no other fluxes; drainage to streamflow complicates this picture, as we describe later).  Nonetheless, if the term is confusing we can remove it, and say, "…the WTF method implies that groundwater levels integrate snowmelt or evapotranspiration signals…"**

- L108: I believe that the WTF method as defined by White (1932) did not account for Q because the observations were done in a desertic environment where Q was not relevant.

**That is correct, but the WTF method is often applied in many situations where Q (or drainage to deeper aquifers) is not zero.  In any case, we believe that the statement we made is correct: "Missing streamflow methods assume that the daily cycle in ET results only in a daily cycle in streamflow, and not a daily cycle in groundwater levels, whereas WTF approaches assume the exact opposite. One can of course question whether either set of assumptions is realistic, but they certainly cannot both be correct."**

- L129: Gribovszki

**Sorry!  We do know how his name is spelled, and we have no idea how that typo got there.  We will fix it.**

- Section 2.1: I would erase the pronunciation of the catchments and historical information that is not necessary to understand the analyses presented later on in the manuscript. I don't think information about potential evapotranspiration is provided for the catchments, and rainfall and temperature are not given for the Independence basin.

**Rainfall and temperature are not reported for the Independence basin because they are not measured there.  (Upper Independence creek is a remote area with no roads and no trails, and we have only measured stream stage variations there.  It is nonetheless interesting for our purposes because its bedrock is much less permeable than Sagehen's.)  Potential evapotranspiration is not reported because it is not directly measured, and its calculation is assumption-dependent.  We mention the pronunciation of "Sagehen" because colleagues have mentioned to us in the past that they don't know how to say it (and because for**

many readers, hearing words in their heads is important to comprehension).  In any case, it's just three added words.  Is saving three words really that important?

It would be good if the description of the two catchments followed the same structure to facilitate the reading.

**This is not feasible because we have a lot more that we need to say about Sagehen than about Independence.**

L191-195 can be erased.

**We disagree, because if this information is not provided, some readers may wonder about the possibility of human influences on the phenomena that we report.**

- Section 2.2: a lot of details can be removed (e.g., precise location of gages). A lot of this information is already in Fig. 1 (latitudes and longitudes could be reported in the figure or tables instead of the text). I would move the description of the sapflow measurements (L232-239) at the end of this section. At the moment, the description starts with weirs and bores, switches to sapflow, and then goes back to bores. L241-245 can be erased.

**Precise locations of gauges are not reported; the lat/long coordinates are those of the snow telemetry (SNOTEL) stations, but we can remove them.  We disagree that "a lot of this information is already in Fig. 1".  Although of course the _locations_ of the various sensors are shown in that figure, the _relevance_ of those locations will not be obvious to readers – for example, that the SNOTEL sites span the same altitude range, and the same distances from the Sierra crest, as the Sagehen catchment does – so this needs to be spelled out in the text.**

**The reviewer is simply incorrect in stating that "At the moment, the description starts with weirs and bores, switches to sapflow, and then goes back to bores".  Lines 219-230 describe the weather stations and SNOTEL station (boreholes are not mentioned there at all, and the main gauge is mentioned only as a reference point for the locations of the SNOTEL stations).  Next, lines 232-239 describe the sap flow measurements, lines 241-253 describe the water stage recorders and the stream gauging stations, and lines 255-266 describe the borehole transects.**

- L256: "To account for the combined..."

**Line 256 does not say anything like that.  Line 276 does say, "To take account of the combined…".  We don't see any difference between "to take account of" and "to account for", besides that one is four characters shorter than the other.  Nonetheless, we can change it.**

- L345-346: I do not think it is correct to say that solar radiation drives streamflow and groundwater fluctuations. There is an indirect relationship, as also stated at L557-558.

**It is broadly acknowledged in the literature of this field that these daily cycles are ultimately derived from the daily cycle in solar radiation.  It is not a 1:1 relationship, but that is not what "driver" means.**

- L385-389: it is not really clear what an integro-differential system is in this context.

**OK, we will change that to "Dynamical phase lags arise whenever one system component integrates another."**

- L390-415: this part is rather long and it seems that is repeated more precisely after Eq. 5.

**We are trying to build understanding.  We want to explicitly link our analysis to the simple dynamical systems approach (lines 390-398), explain how the relationship between storage and discharge arises mechanistically (lines 398-401), give the reader an intuitive understanding of how the phase lag arises from storage integrating its input fluxes (lines 403-409), and then explain that we will be now be analyzing a simple specific example of this more general system (lines 410-415).**

I would just introduce Eq. 4, say that Q is assumed to be a linear function of S (i.e., Q = f(S) = S/τ ) and then write Eq. 5.

**There is a big difference between just assuming that Q is a linear function of S (whereupon readers will wonder, "but what if it isn't?"), and making the point that we make here, namely that even if Q is a nonlinear function of S, it will be approximately linear over small ranges of Q and S, so our analysis still works.**

I would avoid mentioning that the solution is well known (erase L434) and provide the solutions in Eq. 6.

**We need to mention that the solution is well known, or else readers will think that we are claiming that this is an original result, when instead it can be found (in one form or another) in almost any textbook on linear systems analysis.**

I think it should be better to say that it is assumed that the period considered is without P; I do not think it is reasonable to assume P with a daily cycle as M, G, and E.

**We are not assuming that P has a daily cycle, but rather that P+M+G-E has a daily cycle.  Obviously this criterion is met if P is zero and M+G-E has a daily cycle.**

Should there be a mention of the initial conditions for these solutions? I understand that the point is to look at cycles and the initial transient is not important; however, in Fig. 7, I found it strange that the initial values of Q were different.

**There are no initial conditions, and there is no transient.  That is not how Fourier methods work.  Equation 6 gives _exact analytical_ solutions that are valid at _all values of time_, from –infinity to +infinity.  That's because Fourier methods assume that the cyclic input repeats forever, with no beginning and no end.  Equation 6 is neither derived nor solved using numerical integration, so it does not require initialization and there is no transient.**

**What the reviewer calls "the initial values of Q" in Fig. 7 are not initial values at all; they are just the values of the solution at midnight (and midnight, every night, is exactly the same, because the cyclic input and the cyclic output go on forever in both directions).  Since these are not initial values, it is not "strange" that they are different.  They are different because different values of tau yield different amplitudes and phases in Eq. 8.**

- L460-479: I would erase this part. In most cases the inversion of the Fourier transform will be done numerically; therefore, one can just solve Eq. 5 numerically to start with.

**Yes, but if you solve Eq. 5 by numerical integration, you have to worry about initial conditions and transients (and the comments immediately above illustrate how these can become confusing), and you have to worry about numerical stability.  More importantly, if you numerically integrate, you get a numerical answer but you don't get insight, whereas from Eq. 6 or Eq. 8 you can directly see how the amplitudes and phases depend on each of the parameters.  (And although this is not important here, Fourier methods can be massively more efficient than numerical integration, which is why they have been extensively used in global circulation models.)**

The point about the lags is clear from Eqs. 6 and their discussion.

**Of course, but only for an individual cosine wave.  In general, the different frequencies are damped and phase-shifted by different amounts.  Thus if you have only one cosine wave (Eq. 6), your solution is another cosine wave.  The input and the output have the same shape, just with a phase shift and change of amplitude.  But if you have an input that is not a pure sinusoid, then _the output cycle will have a different shape_ from the input cycle.  Thus the apparent lag between the peaks will be different than they would be for a pure sinusoid.**

- L501-504: the references to the lines in Fig. 8 do not seem correct.

**Good catch!  We changed the figure but forgot to update the text.  We'll fix it.**

- L572-573: I would erase this phrase.

**We think it is important, because many colleagues have said to us, "What's diel?  You mean diurnal, right?".  This indicates to us that the terminology needs to be explained.**

- Subsection 3.5: I am not sure this is so important to deserve a full subsection.

**Regardless of the question of importance, the material does not fit with the other subsection headings.  And we do think it is important.  Many colleagues have interpreted the vanishing of the daily cycle as indicating that the snowmelt cycle has ceased (and then later, the evapotranspiration cycle has begun).  Instead, the daily cycle vanishes because the snowmelt and evapotranspiration cycles are canceling each other out.**

- Subsection 3.7: I do not think this subsection is really necessary. I found that it was not adding much to what already presented and supported by the data. I would recommend to cut this part out.

**This section may not be necessary for what the reviewer considers to be the "main result" of the paper.  But in our view, that is not the only important result.  Another important result is that the diel cycle index reflects the spatial variation in snowmelt and ET throughout the drainage basin, and the only way to directly visualize (and quantitatively verify) this result is through remote sensing.  To the best of our knowledge, an analysis like this has never been done before.  Really, who knew that stage fluctuations in streamflow reflected spatial patterns of snow cover and photosynthetic activity that you can see from outer space?**

- L724-725: the mismatch between the peaks in radiation and sapflow is not surprising. Vapor pressure deficit (VPD) is usually the variable that mostly drives transpiration, and I believe that VPD would likely explain the timing of the transpiration peak during the year (that's because there appears not to be water limitation).

**Although we don't go into this here (because we can't do everything, and the topic is not central to the paper), the data do not support the reviewer's conjecture.  VPD remains high until September or October in most years (because the late summer and early fall are hot and dry), whereas sap flow rates begin declining in July.  Controls on seasonal patterns in sap flow are discussed extensively in Cooper et al. (in review), which we cite here.**

- L773: I would use "changes in storage" instead of "mass balance".

**If the reviewer really doesn't like "mass balance" we would suggest "flux balance" instead.  "Changes in storage" isn't wrong, but it gives the wrong focus: on the trends in storage rather than the relationships between the input and output fluxes.**

- L779-781: erase?

**We think it is important to point this out, because otherwise readers could look at Fig. 7 and ask, "yes, but what about storage?"**

- Figures: the captions of most figures are very long. Because the figures are explained in detail in the text, I would try to reduce the length of the captions, where a brief description of what the figures show should be enough.

**The figure captions are written this way as part of a deliberate communication strategy.  Minimalist figure captions often lead to unnecessary workload and confusion for the reader, who must jump back and forth between the figure and the text (perhaps several pages away) in order to understand what the figure says.  Furthermore, many readers scan papers by looking at the figures without reading the text, meaning that the figures should be able to stand on their own.**

**Putting interpretations in figure captions can be a great help to readers, who can thereby get a sense of what the figures _mean_ rather than just what they _are_.  Experience has shown that authors often think that their figures will be self-evident (which of course they are _for the authors_, who already know what they are**

**trying to say), and fail to comprehend how divergent a reader's understanding may be. Thus it is a smart communication strategy to lean in the direction of over-explaining rather than under-explaining.**

- Figure 2: this figure is repeated in a different format in Figs. 11, 12, and 14. I would have these data in a single figure without repetitions.

**The whole point of Figs. 11, 12, and 14 is to let readers see the relationships between seasonal patterns of snow accumulation and melt, and patterns in daily cycles of in streamflow. Readers cannot see these connections if the things that they are supposed to connect are in different figures, many pages apart from one another.**

- Figure 10: if Subsection 3.5 is reduced or removed, perhaps this figure can be removed as well.

**For the reasons that we explained above, we think it is important to keep subsection 3.5.**

-Figure 11: because sapflow and groundwater are related in this figure, I wonder whether it would be better to report the depth to the water table from the surface to show that the water table is within reach of the root system. In the caption, it is said that signals were detrended but it is not explained how.

**Unfortunately we don't know the depth of the rooting zone. The detrending procedure is documented in Eq. 3 at the end of section 2.3.**

- I would consider removing Figs. 13 and 14 along with Subsection 3.7.

**As we explained above, section 3.7 presents a novel analysis that draws connections that have never been drawn before. We believe that this section, and the corresponding figures, should be kept.**

---

## Referee Comment (RC3) · Jessica Lundquist (Referee) · 16 May 2020

Review Summary: Overall, I'm very happy to see this paper. The authors have done a nice job using an integrated and well-measured field site to present the inter-relations between multiple aspects of diurnal cycles in both streams and groundwater in a setting experiencing both snowmelt and evapotranspiration. This is a solid contribution to the field, and I recommend it be published after revisions, particularly addressing my major comments, as follows:

[Figure]

1. While the authors have done a wonderful job integrating and presenting their results, most of what they show is not new. Lundquist and Cayan (2002), see Figures 12-14, clearly illustrate the presence of both snowmelt and ET driven diurnal cycles in river basins. Lundquist and Dettinger (2003), which I have also attached here, with citation below*, as it's hard to find, takes this concept further (see Figures 5 and 6) by using the diurnal cycle switch to highlight inter annual variations in water supply and climate. The paper here builds nicely on this work, but it would be better to present the information as a development and illustration of already published ideas rather than a new idea.

2. At multiple points in the paper, the authors seem to dismiss earlier literature as missing key physical concepts and as being incomplete. At times the tone is dismissive and gives the impression of lacking respect for the earlier work. The paper would be a much stronger contribution if the authors instead addressed why the earlier work took different approaches than here. In many cases, this can be addressed by the different hydrogeologic settings of the basins, which fundamentally changes how the different processes interact and which matter the most. The Tuolumne studies (including many of the papers by Lundquist and by Loheide) are in a granitic basin with very shallow soils, which is quite different from the groundwater dominated Sagehen basin. This fundamentally changes the role of diurnal fluctuations in groundwater on the overall stream signal. (In the detailed comments below, I have called out places in the paper where this contrast could be addressed.)

3. With regards to 2 above, the paper lightly addresses comparisons and contrast between Sagehen and Independence Creek. These could be strengthened through better consideration of dominant terms in different hydrogeologic settings and with further discussion of how these two sites relate to the sites in the literature. Sarah Godsey, the second author, has a nice paper on how geology relates to low flow sensitivity to snow across the Sierra, and it seems like this could be a nice tie in with this study and a discussion on hydrogeologic setting.

If the authors have questions for me regarding these comments or would like the dis-
cuss, I can be reached at Jessica Lundquist, jdlund@uw.edu. I apologize for my time delay in getting this posted. *Citation: Lundquist, J. D. and M. D. Dettinger, 2003. Linking diurnal cycles in river discharge to interannual variations in climate. Proceedings, AMS 17th Conference on Hydrology. Long Beach, California. available at: https://ams.confex.com/ams/annual2003/webprogram/Paper55265.html

Specific Comments Follow:

The paper has a whole has a very nice literature review, but the intro seems to diminish, rather than highlight the work that went before.

line 91: What is an "integrodifferential relationship" ? This is confusing.

Lined 105-109: I think the Loheide and Lundquist paper is a link here. These two assumptions are compatible if the stream and the groundwater levels essentially rise and fall at the same time. Most papers state that ET flux variations are only true in this very linked riparian zone. I don't follow the argument that they must be separate hypotheses.

Lines 115-120: Again, I must beg to differ here. The Lundquist papers focused on the early (snowmelt-dominated) season, in a granite-lined basin with a meadow/riparian system whose groundwater levels responded essentially in synch with the streamflow levels. Again, it's not incompatible, but it's also very nuanced. I think a better way to discuss this would be that the ideas may be system specific and not directly transferable across systems. I think most people are making simplifications that matter for their systems without explicitly discussing other possible systems. So yes, it makes sense to bring them all together, but the "incompatible" statements don't seem right to me.

Line 130: Loheide and Lundquist (2009) had observations as well. Also, with regards to "few studies have examined things together", it seems to me that there are few diurnal cycle studies in general, but it seems like about as many have looked at both as have looked at one.

Upper Independence Basin is more similar to the Tuolumne watershed (compare and contrast your results with the literature).

A fair bit of the literature is also concerned with how much of the riparian area actually takes part in diurnal fluctuations. — can you address this issue?

line 240: Given the sharp rain-shadow gradient in these areas, I would recommend using the 800-m PRISM normals for distributing the Snotel rather than elevation weights (different locations at the same elevation can get quite different amounts of snow). However, I doubt that this would change any of your main results here, so this comment is mainly for future reference rather than a requirement to redo your precipitation mapping for this particular paper.

line 335: also in Lundquist and Cayan 2002

line 360: This discussion is relevant to your "incompatibility" argument, see notes above.

Your Fig. 9 is in L&C 2002, see their Fig 14. This is also in Lundquist and Dettinger 2003, a preprint from a conference (https://ams.confex.com/ams/annual2003/webprogram/Paper55265.html, also attached here). See Figures 5 and 6, which essentially show what you are getting at here.

Fig. 10: You're using straight sinusoids. We know that they're assymmetric. See Lundquist and Cayan 2002.

Line 429: You mean Lundquist and Dettinger (2005) here (not Lundquist and Cayan 2005).

Line 430: Again, I think it's worth comparing and contrasting how the assumptions made in these different systems really relate to the underlying geology. In a granitic system like Tuolumne, there isn't much of a riparian aquifer (unlike in Sagehen, with deep soils) Section 2.2 in Lundquist et al. 2005 discusses the hillslope/riparian flow

paths. Loheide and Lundquist 2009 goes on to show that for the Tuolumne system, the riparian groundwater levels are driven by the stream water levels and not vice versa. Again, you are correct that Sagehen should be modeled differently, but your paper as a whole would be a stronger contribution if you put your results in the context of the varying hydrogeology represented in the literature.

Line 584: This is illustrated in Lundquist and Dettinger 2003, see Figure 5.

Line 620: Also, Independence Greek has more granitic geology and less groundwater reserves. It makes sense in the hydrogeologic context that this would have a snowmelt-dominated signal longer.

Lines 847-850: Data do not appear to be available at this time. Please do check that everything is publicly available and clearly interpretable (with readme files, metadata, etc) before final acceptance of the publication.

Please also note the supplement to this comment:
https://www.hydrol-earth-syst-sci-discuss.net/hess-2020-77/hess-2020-77-RC3-supplement.pdf
* * *
[Figure]

**Supplement:**

**J2.4    LINKING DIURNAL CYCLES OF RIVER FLOW TO INTERANNUAL VARIATIONS IN CLIMATE**

Jessica D. Lundquist* and Michael D. Dettinger
Scripps Institution of Oceanography and USGS, La Jolla, California

**ABSTRACT**

Many rivers in the Western United States have diurnal variations exceeding 10% of their mean flow in the spring and summer months. The shape and timing of the diurnal cycle is influenced by an interplay of the snow, topography, vegetation, and meteorology in a basin, and the measured result differs between wet and dry years. The largest interannual differences occur during the latter half of the melt season, as the snowline retreats to the highest elevations and most shaded slopes in a basin. In most basins, during this period, the hour of peak discharge shifts to later in the day, and the relative amplitude of the diurnal cycle decreases. The magnitude and rate of these changes in the diurnal cycle vary between years and may provide clues about how long-term hydroclimatic variations affect short-term basin dynamics.

**1. INTRODUCTION**

In the Western United States, over half of the water supply is derived from mountain snowmelt, where the snow provides a natural reservoir that delays runoff and provides water in the spring and summer when it is needed most.  However, while the population continues to grow, an alarming change has been noted in recent decades in late season runoff. April-July runoff from mid-elevation Sierra Nevada watersheds has decreased by 10% in favor of other periods of the year, and spring and summer snowmelt has declined markedly (Dettinger and Cayan 1995).  Whether this is long-term natural variability or the forerunner of anthropogenic climate change is unknown, but the trends and short-term variations are well-correlated with regional temperature change and have generated much interest in the climate community (Cayan et al. 2001).  In the long run, it is estimated that, in response to projected global warming of 3°C, the spring-summer snowmelt would be diminished by about one-third (Roos 1987), and winter floods would likely increase.  Thus, understanding snowmelt processes, from modeling surface energy balances to determining the timing and magnitude of snowmelt runoff to better understanding climatic change, is crucially important.

Snowmelt processes are spatially complex and thus difficult to forecast and incorporate in large-scale hydrologic and atmospheric models.  Much of the difficulty arises because snow occurs in patches of nonuniform depth and density, particularly in mountainous regions. In situ measurements of the snowpack are both difficult to make and not necessarily representative of region-wide characteristics.  Satellite images and geographical information systems offer broader spatial coverage, but this data, which is often infrequent in time, is still difficult to relate to the actual snowmelt and river discharge originating from a basin. Fortunately, hourly measurements of river discharge provide another widely available, but as yet unutilized, source of information, providing direct information on basin output at a fine temporal scale.  Each hour, the United States Geological Survey (USGS) monitors thousands of rivers that drain mountainous regions with marked spring snowmelt periods (Slack and Landwehr, 1992).  Many of these, including the Merced River at Happy Isles (Figure 1), exhibit diurnal cycles comprising over 10% of the daily flow each spring (Lundquist and Cayan, 2002).  The shape and timing of these diurnal cycles relate directly to the rate and location of snowmelt and to the pathways travelled by meltwater.  Thus, changes in the diurnal cycle from such basins can provide information about snowmelt and runoff generation during the course of a snowmelt season and from year to year.

[Figure]

**Figure 1:** *The amplitude of the diurnal cycle (hatched red line, right axis) is about 10% of the total streamflow (solid line, left axis).*

* *Corresponding author address:*  Jessica D. Lundquist, Scripps Institution of Oceanography MC-0213, University of California, San Diego, 9500 Gilman Dr., La Jolla, CA 92093-0213; e-mail: jlundquist@ucsd.edu

**2. PATTERNS AND VARIATIONS IN DIURNAL CYCLE CHARACTERISTICS**

Diurnal cycle characteristics and patterns during 1996 to 2000 were analyzed at 50 snowmelt-dominated river gages. Some characteristics, like diurnal-cycle timing, amplitude, and shape, appear to be dominated by basin characteristics (fixed between years but varying between basins) and others appear dominated by meteorological and snow characteristics (varying more between years than between neighboring basins). Early in the season, when most basins are covered in snow, the size and topography of the basins appear to be more important than the depth of the snowpack. During the height of the melt season, the diurnal characteristics are consistent from year to year in most basins. Later in the season, as the snowpack retreats and the basin dries out, the difference between wet and dry years becomes much more apparent.

**2.1 Timing of diurnal flow maxima**

Textbooks (Davar 1970; Singh and Singh 2001), numerical models of the percolation of snowmelt water through a snowpack (Colbeck 1972; Dunne et al. 1976), and localized, small-basin observations (Jordan 1983; Bengtsson 1982; Caine 1992) all report that the hour of day of maximum flow becomes earlier as the snowpack thins and matures, reflecting shorter times for meltwater to travel from the snow surface to the base of the snowpack. However, most

[Figure]

*Figure 3:* Merced River discharge measurements (a) contrast a very dry year (1992) with a very wet year (1998). During both years, the hour of peak flow (b) shifts to later in the day as flows decline. This shift is much more rapid for 1992. A line with the 1992 slope is drawn next to the 1998 slope for comparison.

USGS gages monitor watersheds larger than those that have been examined in these local process studies, and most gages are located downstream, at elevations well below the snowfield. Grover and Harrington (1966) state that the peaks and troughs of the diurnal cycle occur later below a snowfield than at its edge, with a delay that depends on the distance from the snowfield and the stream's flow velocity. Lundquist and Cayan (2002) show that, during the peak melt season, most rivers show only small or inconsistent changes in the hour of peak flow. Rather, the most consistent change of peak timing in snowfed watersheds is the shift of maximum flows to later in the day near the end of the melt season (Figure 2). This shift almost always occurs during the period of declining flows, and reflects increasing travel times as the snowline retreats to the highest, farthest reaches of the basin (Grover and Harrington 1966). This timing shift suggests that, at the medium- to large-basin scales, near the end of the season, snowmelt distribution is more important than snow depth in affecting diurnal timing. The hour of peak flow shifts to later in the day more rapidly in dry years (Figure 3), and the slope of this shift correlates inversely with the duration of the melt season, as measured by the number of days with a discernible diurnal cycle caused by snowmelt.

**2.2 Diurnal amplitude/relative amplitude**

Daily variations of the amplitude of the diurnal flow cycle are highly correlated with daily temperatures and

[Figure]

*Figure 2* Average shift in hour of maximum flow from beginning to end of July, 1996-2000. Black circles show shifts to later in the day, white to earlier in the day. Crosses show stations with no significant change. Circle sizes show rate of change, ranging from 0.1 to 0.75 hours per day.

with total discharge in most basins (such as the Merced, Figure 1), reflecting both the rates of melt and the area over which melt is occurring. Relative amplitude — measured as the diurnal amplitude divided by the average daily discharge — is largest when most meltwater reaches the gage within one day of melting and is smallest when most meltwater arrives as part of the recession curve several days after melting (Collins and Young 1981). In most rivers, the relative amplitude correlates directly with total discharge (such as the Merced River, Figures 4a-d), suggesting that a greater fraction of meltwater travels via fast pathways to the gage during times of higher streamflow. However, if the relative amplitude is written as a linear function of discharge ($A/Q = kQ$, where $A$ = amplitude of the diurnal cycle, $Q$ = mean daily discharge, and $k$ = the best-fit slope), the slope ($k$) varies by a factor of five between wet and dry years in the Merced River (Figure 4d).

Figures 4a, 4b, and 4c illustrate how the drier year, 1994, has a larger relative amplitude than the wetter year, 1998. The larger relative amplitude reflects a larger portion of discharge travelling along fast pathways during the dry year. This suggests that, in the Merced Basin, subsurface flows are a greater percentage of the total runoff during wet years than dry years.

The Merced River occupies a glaciated basin, dominated by shallow soils and impermeable granite. Out of 34 river basins examined in the Sierra Nevada and Rocky Mountains, 20 exhibit increases in the slopes of relative amplitude vs. discharge relations in drier years. The basins that do not show such increases generally do not have a significant correlation between discharge and the relative amplitude. These basins have different types of rock and soils and probably have larger groundwater flow components.

[Figure]

**Figure 4:** *Relationship between discharge and relative amplitude for the Merced River at Happy Isles, in California. (a) In a dry year, 1994, the relative amplitude correlates with the discharge. (b) In a wet year, 1998, the relative amplitude also correlates with the discharge, but while the discharge is much larger in 1998 than 1994, the relative amplitude is the same size or smaller. (c) Plotting relative amplitude as a function of discharge for the two years illustrates that, at every daily flow rate, the relative amplitude is greater during the dry year. (d) For all eight years analyzed here, the slope of the relative amplitude vs. discharge relation decreases as the total snow water for the season increases. The total snow water is calculated as the sum of the discharge during the season with a discernible diurnal cycle (dates pictured in 4a and 4b). This sum is normalized by basin area to provide an estimate of the average depth of the year's snowpack.*

**2.3 Diurnal cycle shape**

Diurnal cycles in some streams switch from snowmelt-dominated characteristics in the spring to evapotranspiration/infiltration-dominated characteristics in the late summer (Lundquist and Cayan, 2002). This shift is most obvious in the shape of the diurnal cycle, as shown for the Little Bighorn River at Stateline near Wyola, Montana (Figure 5). Lundquist and Cayan (2002) found that where water is added diurnally, as in snowmelt, the diurnal cycle is characterized by a sharp rise and gradual decline (Figure 5b). Where water is removed diurnally, as by evapotranspiration or infiltration, the diurnal cycle is characterized by a gradual rise and sharp decline (Figure 5c). The change between these two shapes signifies a shift from wet (gaining water) to dry (losing water) conditions. The difference between the number of hours the hydrograph falls and the number of hours the hydrograph rises each day provides a simple numerical measure of the diurnal cycle's shape. Positive numbers indicate longer decays than rises and reflect snowmelt-domination. Negative numbers indicate longer rises and reflect evapotranspiration or infiltration. The date this index changes sign varies from year to year (Figure 6a), depending on the year's snowpack volume and thus the year's soil-moisture reserves. Figure 6b shows the date of this shift in the Little Bighorn River as a function of total annual runoff from 1996 to 2000. (1997 is excluded because frequent rainstorms in July obscured the diurnal cycle so that its characteristics could not be determined.) Overall, the Little Bighorn shifts between the two regimes later in the summer during wetter years.

**3. DISCUSSION, CONCLUSIONS, AND FUTURE WORK**

How will watersheds respond as temperatures warm and snowpacks diminish in size? Which areas of a basin will be driest, and how will ecosystems respond? Will snowmelt become more gradual (all winter long), and thus will streamflow derive more from groundwater and subsurface flows? Alternatively, will soils be depleted of moisture, resulting in smaller recharge rates and smaller subsurface components of flow? Will the snowmelt-dominated period each year become longer (but earlier) or just shorter?

The diurnal cycles of streamflow can help answer these questions. Conceptual modelling is needed to work out the details, but the observations provide clear indications that seasonal evolution of these diurnal cycles is a sensitive indicator of year-to-year differences in western snowmelt conditions. Contrasts between diurnal cycles in wet and dry years show that long-term changes and trends in western snowmelt would likely be evident in a carefully crafted set of indices based on the diurnal cycle characteristics. During drier years, the travel time between the melt source and the stream gage changes rapidly, perhaps indicating a more rapid retreat of the snowline away from the gage than in wet years. The

[Figure]

**Figure 5:** *(a) The 1996 hydrograph for the Little Bighorn River, illustrating how the diurnal cycle changes as snowmelt forcing gives way to evapotranspiration/ infiltration forcing. Periods illustrated in (b) and (c) were fit to a line, which was then subtracted out to accentuate the diurnal fluctuations.*

relative amplitude of diurnal oscillations is smaller for a given discharge during dry years, suggesting that inflows from slower pathways, such as subsurface reservoirs, are depleted. Drier conditions also lead to earlier shifts from a snowmelt-dominated diurnal cycle (with a rapid rise and slow decay) to an evapotranspiration/infiltration-dominated diurnal cycle (with a slower rise and rapid decay).

Further work is needed to model how wetter and drier basin conditions will yield these diurnal cycle changes. Longer time-series and comparisons between basins with different soils, rocks, and vegetation will also help provide a clearer picture of the range of diurnal cycle responses to interannual variations in climate.

**4. REFERENCES**

Bengtsson, L. 1982: Groundwater and meltwater in the snowmelt induced runoff. *Hydrol. Sci. J.,* **27**, 147-158.

Caine, N. 1992: Modulation of the diurnal streamflow response by the seasonal snowcover of an alpine basin. *J. Hydrology,* **137**, 245-260.

Cayan, D. R., S. A. Kammerdiener, M. D. Dettinger, J. M. Caprio, and D. H. Peterson, 2001: Changes in the onset of spring in the Western United States. *Bull. Am. Met. Soc.,* **82**, 399-415.

Colbeck, S. C., 1972: A theory of water percolation in snow. *J. Glaciol.,* **11**, 369-385.

Collins, D. N., and G. J. Young, 1981: Meltwater hydrology and hydrochemistry in snow- and ice-covered mountain catchments. *Nordic Hydrol.,* **12**, 319-334.

Davar, K. S. 1970: Section IX: Peak Flow – Snowmelt Events. *Handbook on the Principles of Hydrology.* D. M. Gray, Ed., Water Information Center, Inc., 9.1-9.25.

Dettinger, M. D. and D. R. Cayan, 1995: Large-scale atmospheric forcing of recent trends toward early snowmelt runoff in California. *J. Climate,* **8**, 606-623.

Dunne, T., A. G. Price, and S. C. Colbeck, 1976: The generation of runoff from subarctic snowpacks. *Water Resour. Res.,* **12**, 677-685.

Grover, N. C., and A. W. Harrington, 1966: *Stream flow: measurements, records and their uses.* Dover Publications, Inc. 363 pp.

Jordan, P. 1983: Meltwater movement in a deep snowpack 1. Field observations. *Water Resour. Res.,* **19**, 971-978.

Lundquist, J. D. and D. Cayan, 2002: Seasonal and spatial patterns in diurnal cycles in streamflow in the Western United States. *J. Hydrometeorology,* **3**, 591-603.

Roos, M., 1987: Possible changes in California snowmelt runoff patterns. *Proceedings of the 4th Annual PACLIM Workshop,* Pacific Grove, CA., 22-31.

Singh, P. and V. P. Singh, 2001: *Snow and Glacier Hydrology.* Water Science and Technology Library, Kluwer Academic Publishers, 742 pp.

Slack, J. R., and J. M. Landwehr, 1992: Hydro-climatic data network (HCDN): A U. S. Geological Survey streamflow data set for the united states for the study of climate variations, 1874-1988. *USGS Open-File Report* **92-129**, 193 pp.

[Figure]

[Figure]

**Figure 6:** *(a) Shifting of the flow in the Little Bighorn River from a longer diurnal decay time (snowmelt-dominated) to a longer rise time (evapotranspiration/infiltration-dominated), as the recent snowmelt season draws to a close. (b) The date of the shift as a function of the total discharge for each year. In general, drier years shift earlier in the summer.*

---

## Author Comment (AC3) · 3 Jun 2020

We thank Jessica Lundquist for her comments on our manuscript. Below we respond (in bold type) to Prof. Lundquist's specific comments (in normal type).

Review Summary: Overall, I'm very happy to see this paper. The authors have done a nice job using an integrated and well-measured field site to present the inter-relations between multiple aspects of diurnal cycles in both streams and groundwater in a setting experiencing both snowmelt and evapotranspiration. This is a solid contribution to the field, and I recommend it be published after revisions, particularly addressing my major comments, as follows:

**We thank Prof. Lundquist for these supportive comments.**

1. While the authors have done a wonderful job integrating and presenting their results, most of what they show is not new.

**We disagree with the claim that "most of what they show is not new".**

**Specifically, Figures 3, 4, and 5 illustrate the strong hour-by-hour coupling between the solar flux and the rate of rise and fall of riparian groundwater tables, during both snowmelt and ET cycles. Most of this is new. Our Figure 5 somewhat resembles Figure 6 of Loheide (2008), but that only concerns evapotranspiration cycles, and compares derived estimates of ET and potential ET, not solar flux and the rate of groundwater rise/fall (although these are obviously related).**

**Our conceptual model analysis presented in Section 3.3 and illustrated in Figures 7 and 8 is also new. Although the conceptual model makes broadly similar assumptions to those of Gribovzski et al. (2008), as we acknowledge on line 426, we apply our model in different ways and reach new conclusions. In particular, our analysis leads to two conclusions that are new, and in our view, significant:**

> **1) The commonly observed lags between peak snowmelt or peak ET and the daily peak or trough in streamflow are largely dynamical phase lags, not travel-time lags, at least in small catchments. This result challenges the assumptions underlying decades of prior work, including Wicht (1941), Jordan (1983), Bond et al. (2002), Lundquist et al. (2005), Lundquist and Dettinger (2005), Wondzell et al. (2007), Barnard et al. (2010), Graham et al. (2013), and Fonley et al. (2016); see the manuscript for the full citations.**

> **2) The amplitude of the daily cycle in streamflow cannot be quantitatively linked to the daily ET or snowmelt flux, unless the time constant tau of the near-stream groundwater system is quite short. This result also calls into question over 50 years of prior studies, including Tschinkel (1963), Meyboom (1965), Reigner (1966), Bond et al. (2002), Boronina et al. (2005), Barnard et al. (2010), Cadol et al. (2012), and Mutzner et al. (2015).**

**We also note that this conceptual model also explains the asymmetry in snowmelt and ET cycles in streamflow, which were pointed out by Lundquist and Cayan (2002).**

**The diel cycle index developed in Section 3.4 and illustrated in Figure 9 is also new. This provides a new tool for characterizing seasonal transitions between snowmelt and evapotranspiration cycles in small basins.**

**In Section 3.5 and Figure 10 we observe that, as snowmelt cycles give way to ET cycles, the amplitude of daily cycling in the stream nearly vanishes. We also infer that this results from destructive interference between the snowmelt and ET signals originating from different parts of the catchment. These points are also, to the best of our knowledge, new.**

**We also observe that the transition between snowmelt and evapotranspiration cycles occurs differently in groundwaters and streamflow, due to the fact that groundwater cycles mostly reflect local forcing and streamflow integrates that forcing over the drainage network (Section 3.5 and Figures 11 and 12). This transition occurs earlier or later at different points along the drainage network (Figures 12 and S3),**

**reflecting differences in snow accumulation and melt (and also in the onset of evapotranspiration) from place to place depending on altitude and aspect.  This is all, to the best of our knowledge, new.**

**We also show that the spatial pattern in daily streamflow cycles is consistent with the spatial evolution of snow cover and vegetation activity as seen from space (Figures 13 and 14).  This is also, to the best of our knowledge, new.**

**These new observations and inferences comprise almost the entire manuscript, whether measured by number of figures or length of text.  Thus the claim that "most of what they show is not new" is factually incorrect.**

Lundquist and Cayan (2002), see Figures 12-14, clearly illustrate the presence of both snowmelt and ET driven diurnal cycles in river basins.

**Of course, and we say almost exactly this in the second sentence of the manuscript, "Both snowmelt and evapotranspiration cycles result from daily variations in solar flux, but are of opposite phase (Lundquist and Cayan, 2002…"  We certainly do not want to give the impression that we are claiming to have discovered snowmelt and ET cycles in streamwater (indeed, these were known in the literature for decades before Lundquist and Cayan's work, as section 2 of their paper makes clear).**

Lundquist and Dettinger (2003), which I have also attached here, with citation below*, as it's hard to find, takes this concept further (see Figures 5 and 6) by using the diurnal cycle switch to highlight inter annual variations in water supply and climate. The paper here builds nicely on this work, but it would be better to present the information as a development and illustration of already published ideas rather than a new idea.

**Figure 5 of Lundquist and Dettinger (2003) is a verbatim copy of the previously mentioned Figure 14 of Lundquist and Cayan (2002).  Figure 6 of Lundquist and Dettinger (2003) makes the point that snowmelt cycles switch to ET cycles earlier in drier years, based on what appears to be a preliminary analysis of daily cycle asymmetry in four years at one river.  We don't think it is correct to say that our manuscript "builds nicely on this work", given that neither the methods nor the questions are similar: we use the diel cycle index rather than daily cycle asymmetry to measure the transition from snowmelt to evapotranspiration cycles, and we do not focus on year-to-year variations in the timing of these transitions, but rather the spatial evolution of those transitions in a catchment context.**

2. At multiple points in the paper, the authors seem to dismiss earlier literature as missing key physical concepts and as being incomplete. At times the tone is dismissive and gives the impression of lacking respect for the earlier work.

**We certainly do not mean to be dismissive or disrespectful toward prior work.  However, in some cases it is unfortunately necessary to point out where the assumptions or conclusions of these previous studies are contradicted by our data, or by accepted physical principles of water flow in hydrologic systems.  For example, nearly every paper that discusses the propagation speed of snowmelt or ET cycles assumes that this propagation speed equals the flow velocity of the water itself, but that is simply not correct.  It has been known for decades that changes in flow rates in hydrologic systems propagate at the kinematic wave speed, not the bulk flow rate.  We have tried to point this out as gently as we can, but we cannot avoid the fact that it needs to be said.**

The paper would be a much stronger contribution if the authors instead addressed why the earlier work took different approaches than here.

**We do not want to speculate about why earlier studies took the approaches that they did.  Obviously it is a different situation if the papers themselves reveal their motives, but that is rarely the case.**

In many cases, this can be addressed by the different hydrogeologic settings of the basins, which fundamentally changes how the different processes interact and which matter the most. The Tuolumne

studies (including many of the papers by Lundquist and by Loheide) are in a granitic basin with very shallow soils, which is quite different from the groundwater dominated Sagehen basin. This fundamentally changes the role of diurnal fluctuations in groundwater on the overall stream signal. (In the detailed comments below, I have called out places in the paper where this contrast could be addressed.)

**We agree that different hydrologic settings could be important, but we are not aware of evidence that supports the statement that "this fundamentally changes the role of diurnal fluctuations in groundwater on the overall stream signal."  Indeed, our data suggest the opposite, because daily snowmelt and ET cycles in our granitic Independence Creek drainage are strikingly similar to those in the more groundwater-dominated Sagehen Creek basin.  We believe this is because the transmission of daily cycles depends on how groundwater and soil water respond to incremental additions and subtractions of water from the surface, not on the total volume of water in storage.  Note that, for example, S in Eq. (4) is transient storage (or, as the manuscript puts it, "storage above the stream"), so whether there is a large volume of water stored below the stream does not change the behavior of our conceptual model.  And again, the field data support this view, because the daily cycles in Independence Creek and Sagehen Creek are similar, even though one basin is granitic and the other has an extensive groundwater system.**

**To make this point clearer in the manuscript, we will add the following figures and text:**

[Figure]

*Figure A.  Snowmelt-driven daily cycles in stream water levels measured in April 2007 at three locations along Upper Independence Creek, underlain by glaciated granodiorites, and two locations along Sagehen Creek, underlain by thick volcanic and volcaniclastic deposits.  Stream stages were detrended using Eq. (3).  The shapes and phases of the daily cycles are similar, and all exhibit similar lags relative to the solar forcing, despite the marked geological differences between the two catchments.*

[Figure]

*Figure B.  Evapotranspiration-driven daily cycles in stream water levels measured in July 2007 at three locations along Upper Independence Creek, underlain by glaciated granodiorites, and two locations along Sagehen Creek, underlain by thick volcanic and volcaniclastic deposits.  Stream stages were detrended using Eq. (3).  The shapes and phases of the daily cycles are similar, and all exhibit similar lags relative to the solar forcing, despite the marked geological differences between the two catchments.*

*Although this conceptual model has been developed in the context of Sagehen Creek, which has an extensive groundwater aquifer, the mechanisms described here do not require substantial aquifer storage.  In the model, changes in storage equal changes in discharge multiplied by the characteristic response time τ.  This directly implies that the daily range of storage also equals τ times the daily range of discharge.  At the Sagehen main gauge, where we can measure daily cycles in units of discharge (at the other stations we lack rating curves and thus have only stage measurements), typical daily ranges of discharge during peak snowmelt were ~2-4 mm/day in 2006 (above-average SWE), 0.2-0.6 mm/day in 2007 (below-average SWE), and 0.4-1 mm/day in 2008 (roughly average SWE).  Even τ values as small as  ~0.2-0.5 days are sufficient to generate significant lags between peak snowmelt and peak streamflow, implying that these lags could be associated with storage changes of only 0.4-2 mm in 2006, 0.04-0.3 mm in 2007, and 0.08-0.5 mm in 2008 (the ET cycles, and their associated ranges of storage, are about 1-2 orders of magnitude smaller).  This simple calculation implies that significant dynamical phase lags can be generated from small daily variations in soil water and shallow groundwater, and that a substantial groundwater aquifer is not required.*

*This inference can be tested by comparing daily streamflow cycles in Sagehen Creek with those in Upper Independence Creek. The Upper Independence basin is dominated by glacially scoured granodiorites (Sylvester and Raines, 2017) and lacks the volcanic and volcaniclastic deposits that host Sagehen's extensive groundwater aquifer.  Despite this sharp contrast in hydrogeology, Figs. A and B show that snowmelt and ET cycles are similar in Upper Independence Creek and Sagehen Creek.  Streamflow cycles lag the solar flux curve by slightly more at the Sagehen main gauge than at the other four stations shown in Figs. A and B, reflecting the fact that the main gauge is farther downstream from its most distant headwaters (7.9 km, compared to 2.6-3.9 km for the other four stations) and integrates over a larger drainage area (27.6 km2 vs 4.7-7.7 km2 for the other stations), and thus accumulates commensurately larger kinematic wave lags.  The daily cycle amplitudes also differ, due to differences in drainage areas and channel cross-sections among the different stations.  Nevertheless, the clear conclusion from Figs. A and B is that the shapes of the daily cycles, and their phase lags relative to the solar flux, are strikingly similar between the granitic, glacially scoured Upper Independence basin, and the groundwater-dominated Sagehen basin.  This strongly suggests that*

*similar mechanisms shape the streamflow cycles in both basins, despite the marked differences between their geological settings.*

3. With regards to 2 above, the paper lightly addresses comparisons and contrast between Sagehen and Independence Creek. These could be strengthened through better consideration of dominant terms in different hydrogeologic settings and with further discussion of how these two sites relate to the sites in the literature. Sarah Godsey, the second author, has a nice paper on how geology relates to low flow sensitivity to snow across the Sierra, and it seems like this could be a nice tie in with this study and a discussion on hydrogeologic setting.

**A multi-site intercomparison study would certainly be interesting, but would be an entirely different study from the one we present here. Challenges confronting any such intercomparison would be the differences in which variables are measured at which sites, as well as the general problem of data availability (although Prof. Lundquist's Yosemite Hydroclimate Network, http://depts.washington.edu/mtnhydr/data/yosemite.shtml, is a good example of how this problem will be gradually overcome). But again, this would be an entirely different study. Beyond the comparisons between Sagehen and Independence outlined above, we do not want to make assumptions about "dominant terms in different hydrogeologic settings" for other basins without having a clear basis for those assumptions.**

If the authors have questions for me regarding these comments or would like to discuss, I can be reached at Jessica Lundquist, jdlund@uw.edu.

**Thanks for the very helpful discussion that we subsequently had by video conference. We particularly appreciate your suggestion that a conceptual diagram would be help readers to visualize the groundwater-stream interactions that underlie the lag in daily cycles. We will therefore add the following figure to the manuscript:**

[Figure]

*Figure X. Visualization of groundwater-stream coupling that leads to lagged evapotranspiration cycles in groundwater levels and streamflow (snowmelt cycles are similar but reversed). Streamflow is supplied by drainage from riparian groundwater, and this drainage rate is faster at higher levels of riparian groundwater storage (S). Riparian groundwater storage changes at a rate dS/dt that depends on the flux balance between streamflow (Q), evapotranspiration (ET), and groundwater recharge from surrounding uplands (G). The relative magnitudes of these fluxes in each panel are indicated by the number of arrows; upland recharge (G) is constant but the other fluxes vary from panel to panel. Inset figures show the corresponding phases of the daily cycle in streamflow and groundwater levels. In the morning (a), groundwater storage and streamflow reach their maximum and begin to decline as the evapotranspiration rate rises enough, relative to the difference between groundwater recharge and discharge, that the riparian aquifer reaches equilibrium and begins to decline. Around noon (b), high evapotranspiration fluxes lead to a strongly negative flux balance and a rapid draw-down of groundwater storage, and thus a rapid decline in streamflow (the dashed line indicates the morning high-stand of groundwater levels and stream stage, as a reference). Toward evening (c), riparian groundwater and stream stage reach their minimum and begin to rise when evapotranspiration rates and streamflows decline enough that the riparian aquifer reaches equilibrium and begins to refill. During the night (d) riparian groundwater levels (and thus stream stages) slowly rebound, because evapotranspiration is nearly zero and upland recharge exceeds stream discharge.*

I apologize for my time delay in getting this posted. *Citation: Lundquist, J. D. and M. D. Dettinger, 2003. Linking diurnal cycles in river discharge to interannual variations in climate. Proceedings, AMS 17th Conference on Hydrology. Long Beach, California. available at: https://ams.confex.com/ams/annual2003/webprogram/Paper55265.html

Specific Comments Follow:

The paper has a whole has a very nice literature review, but the intro seems to diminish, rather than highlight the work that went before.

**That is certainly not our intention. In several cases, though, we need to point out limitations and inconsistencies in the literature. We have tried to do that as gently as possible, consistent with the need to also be clear.**

line 91: What is an "integrodifferential relationship" ? This is confusing.

**Another reviewer also found this confusing, and we will change it. What we meant is a relationship that is described by a differential equation rather than an algebraic one.**

Lined 105-109: I think the Loheide and Lundquist paper is a link here. These two assumptions are compatible if the stream and the groundwater levels essentially rise and fall at the same time. Most papers state that ET flux variations are only true in this very linked riparian zone. I don't follow the argument that they must be separate hypotheses.

**We will try to clarify this argument in the manuscript, because the assumptions underlying the water table fluctuation approach and the missing streamflow approach are in fact mathematically inconsistent with one another. This inconsistency has nothing to do with whether groundwater and streamwater rise and fall at essentially the same time (which they also do in our conceptual model, and also in our data).**

**Here is the problem. "Missing streamflow" methods must assume that daily additions and removals of water from the catchment are transmitted 1:1 to the stream; otherwise, the change in streamflow does not quantitatively reflect the snowmelt or evapotranspiration rate. But daily additions and removals of streamflow are not transmitted 1:1 to the stream if groundwater levels also vary on a daily cycle, because in this case, the daily addition of snowmelt (for example) is partitioned between both the daily change in streamflow and the daily change in groundwater storage.**

**Conversely, water table fluctuation methods assume that daily removals of groundwater by ET are reflected 1:1 in changes in groundwater storage (net of an assumed constant input from upland recharge and constant output to streamflow); otherwise, the change in groundwater level does not quantitatively reflect the evapotranspiration rate. But daily removals of groundwater by ET are not reflected 1:1 in groundwater storage if streamflow (which is generated predominantly from groundwater…) also varies on a daily cycle.**

**Thus the "missing streamflow" approach assumes that ET and snowmelt will only change streamflow fluxes (and thus groundwater storage will be constant), and water table fluctuation approaches assume that ET and snowmelt will only change groundwater storage (and thus that streamflow fluxes will be constant). Thus these premises really are inconsistent with one another.**

Lines 115-120: Again, I must beg to differ here. The Lundquist papers focused on the early (snowmelt-dominated) season, in a granite-lined basin with a meadow/riparian system whose groundwater levels responded essentially in synch with the streamflow levels. Again, it's not incompatible, but it's also very nuanced. I think a better way to discuss this would be that the ideas may be system specific and not directly transferable across systems. I think most people are making simplifications that matter for their systems without explicitly discussing other possible systems. So yes, it makes sense to bring them all together, but the "incompatible" statements don't seem right to me.

**Our statement was that studies that attribute lags in daily cycles exclusively to travel times and flow velocities are incompatible with the assumptions that underlie water table fluctuation (WTF) methods. WTF methods assume that groundwater integrates its inputs, which will create a substantial phase lag even in the absence of any travel-time lags. If the assumptions of the WTF approach are correct, then the first several hours of the lag in groundwater or streamflow cycles cannot be attributed to travel times. Thus the two approaches do, in fact, make incompatible assumptions.**

**This has nothing to do with whether groundwater levels and streamflow levels are synchronized with one another (which they are in our conceptual model, and also often are in our data, as well as in your data from Tuolumne Meadow). Rather, the issue is (as stated in lines 111-121) whether the cyclic response by _both_ groundwater and streamflow lags the cyclic forcing by snowmelt or ET, and whether this lag arises from a phase lag due to integration in the shallow groundwater system, or due to travel times.**

**The phase lag that we have identified at Sagehen is also seen at Independence Creek, and it is also present in your own Yosemite data, including at Budd Creek, Delaney Creek, and Lyell Fork below Maclure (at 2940 m, and only about 4 km below the headwaters at Lyell Glacier). Thus these different geological settings exhibit a consistent pattern of behavior. Generating this same pattern of behavior, with roughly the same lag time, by travel-time delays would require the same rather particular set of conditions (depth of snowpack, distance to channel, length of channel network, etc.) to hold across these very different settings.**

Line 130: Loheide and Lundquist (2009) had observations as well.

**We will change this sentence to eliminate the distinction between modeling and observational studies.**

Also, with regards to "few studies have examined things together", it seems to me that there are few diurnal cycle studies in general, but it seems like about as many have looked at both as have looked at one.

**We agree that there are few diurnal cycle studies in general, but even among this group, studies of coupled groundwater/surface water cycles are relatively scarce – particularly those that actually measure both groundwater and surface water cycles, along with measurements of sapflow/ET as a driver. Even scarcer are studies that have looked at both groundwater/surface water cycles in response to both snowmelt and ET, in the spatial context of elevation gradients and the temporal context of seasonal shifts from snowmelt to ET and back again.**

Upper Independence Basin is more similar to the Tuolumne watershed (compare and contrast your results with the literature).

**As we point out above, the daily cycles in streamflow at Upper Independence, and in headwater streams in the Tuolumne basin, are similar to those at Sagehen despite the substantial difference in lithology.**

A fair bit of the literature is also concerned with how much of the riparian area actually takes part in diurnal fluctuations. Can you address this issue?

**We cannot, for the reasons described in Section 3.3. To do this by modeling requires assuming that we actually know the volume of the daily snowmelt or ET forcing, but as Section 3.3 makes clear, we cannot know this unless we also know the time constant (tau) of the riparian soil water/groundwater system. Alternatively, one could detect the spatial extent of groundwater fluctuations using direct measurements, but it would require a more extensive groundwater monitoring network than we have.**

line 240: Given the sharp rain-shadow gradient in these areas, I would recommend using the 800-m PRISM normals for distributing the Snotel rather than elevation weights (different locations at the same elevation can get quite different amounts of snow). However, I doubt that this would change any of your main results here, so this comment is mainly for future reference rather than a requirement to redo your precipitation mapping for this particular paper.

**Thanks, yes, this is an interesting point. The SNOTEL sites are sited along the rain-shadow gradient, so they capture the rain shadow effect rather well. In any case our results do not depend on mass balances, so they don't require that we have an ideal interpolation of the SNOTEL data.**

line 335: also in Lundquist and Cayan 2002

**We will add this reference.**

line 360: This discussion is relevant to your "incompatibility" argument, see notes above.

**We don't understand the point here. The Loheide and Lundquist study was conducted more than six weeks after snowmelt ended in Tuolumne Meadow, and the snowmelt signal in the stream was generated much higher up in the basin (where there are no groundwater measurements). Thus the Loheide/Lundquist paper does not bear directly on how snowmelt signals are transmitted to the stream, which is our focus here.**

Your Fig. 9 is in L&C 2002, see their Fig 14. This is also in Lundquist and Dettinger 2003, a preprint from a conference (https://ams.confex.com/ams/annual2003/webprogram/Paper55265.html, also attached here). See Figures 5 and 6, which essentially show what you are getting at here.

**We disagree with the claim that "Your Fig. 9 is in L&C 2002", which could be construed as accusing us of using prior work without attribution. We are sure that this was not your intention, but nonetheless we need to set the record straight. Below we show Figure 14 of L&C 2002 and our Figure 9, side by side.**

[Figure]

FIG. 14. (a) A 1996 hydrograph for the Little Bighorn River, illustrating how diurnal cycle changes as snowmelt forcing gives way to evapotranspiration/infiltration forcing. Periods illustrated in (b) and (c) were fit to a line, which was then subtracted out to accentuate the diurnal fluctuations.

[Figure]

Figure 9. Correlations between solar flux and rates of rise and fall of water levels (Sagehen Creek, B transect) during two example days, one when the catchment was snow-covered and the stream exhibited a strong snowmelt cycle (24 April 2007), and another when the catchment was snow-free and the stream exhibited a strong evapotranspiration cycle (4 June 2007). In the lower plot, the correlation coefficients (blue dots) for each day indicate the relative dominance of snowmelt or evapotranspiration as generators of daily cycles in Sagehen Creek, while the gray shading shows the amplitude of the detrended daily stage fluctuations.

**We are perplexed by the statement that the figure on the right is contained in the figure on the left. The figure on the left (from L&C 2002) shows only that streams can exhibit daily cycles driven by both snowmelt and evapotranspiration, and that these cycles have different shapes. Claiming that that our Figure 9 makes the same points is simply inconsistent with Sections 3.4 through 3.6 of our paper.**

**Below we show Figures 5 and 6 from Lundquist and Dettinger (2003). Again, we do not think that our Figure 9 is equivalent to these or contained in them. The figures below concern how the _asymmetry_ of the daily cycle changes between snowmelt and ET-dominated cycles. Our Figure 9 concerns the _phases_ of the cycles, and in particular, the phase relationship between time derivative of stream water levels and the solar flux (as a driver, through snowmelt and ET, of those rises and falls in stream water levels). This is quantified through the diel cycle index, which is nowhere mentioned in Lundquist and Dettinger (2003) or L&C 2002.**

[Figure]

**Figure 5:** (a) The 1996 hydrograph for the Little Bighorn River, illustrating how the diurnal cycle changes as snowmelt forcing gives way to evapotranspiration/infiltration forcing. Periods illustrated in (b) and (c) were fit to a line, which was then subtracted out to accentuate the diurnal fluctuations.

**Figure 6:** (a) Shifting of the flow in the Little Bighorn River from a longer diurnal decay time (snowmelt-dominated) to a longer rise time (evapotranspiration/infiltration-dominated), as the recent snowmelt season draws to a close. (b) The date of the shift as a function of the total discharge for each year. In general, drier years shift earlier in the summer.

Fig. 10: You're using straight sinusoids. We know that they're assymmetric. See Lundquist and Cayan 2002.

**We agree that they're asymmetric, and in fact our analysis explains _why_ they're asymmetric (see our Figure 7, and Section 3.3, particularly the paragraph beginning on line 480 – where we already cite two Lundquist papers). The point of this figure was to simply show that the disappearance of the diel cycle in the stream was due to the destructive interference between the snowmelt and ET cycles. The asymmetry in those cycles is not important to this point, but we will re-draw the figure to include it.**

Line 429: You mean Lundquist and Dettinger (2005) here (not Lundquist and Cayan 2005).

**You're right, sorry, that was just a typo. We'll fix it (we had it right in the reference list).**

Line 430: Again, I think it's worth comparing and contrasting how the assumptions made in these different systems really relate to the underlying geology. In a granitic system like Tuolumne, there isn't much of a riparian aquifer (unlike in Sagehen, with deep soils) Section 2.2 in Lundquist et al. 2005 discusses the hillslope/riparian flow paths.

**Yes, but Section 2.2 of Lundquist et al. (2005) discusses these flow paths in terms of the transit time of the water, which is not what controls how rapidly changes in flow rates will be transmitted to the stream. Changes in flow rates propagate at the kinematic wave velocity, not at the mean flow velocity. This is true in snowpacks, in unconfined groundwater systems, and in open-channel flow (in confined groundwater systems, changes in flow rates propagate even faster, at the pressure wave propagation velocity).**

Loheide and Lundquist 2009 goes on to show that for the Tuolumne system, the riparian groundwater levels are driven by the stream water levels and not vice versa.

**One cannot make that statement about riparian groundwater levels throughout "the Tuolumne system", but only about Tuolumne meadow, and only during the period studied by Loheide and Lundquist, more than a month after snowmelt ended there. The daily cycles studied by Loheide and Lundquist 2009 are driven by snowmelt many kilometers upstream from Tuolumne meadow (and understandably, by that late in the season, they drive the variations in the groundwater rather than the other way around, since there is no**

**remaining snowpack, and thus no locally-generated snowmelt, at the Tuolumne Meadow groundwater wells).**

**But the snowmelt cycles in the Tuolumne River entered the river somehow, and the headwater gauges (Lyell Fork below Maclure, Budd Creek, and Delaney Creek) indicate that snowmelt cycles enter the stream with several-hour lags relative to the snowmelt rate itself.  We show that similar lags can arise whenever inflow rates to the stream are coupled to riparian groundwater storage, which integrates the snowmelt input itself. (In principle, this lag can also arise by kinematic wave propagation through snowpacks, hillslopes, and channels, but it would take some rather special circumstances for the resulting kinematic wave lag to be so similar in so many different settings.)**

Again, you are correct that Sagehen should be modeled differently, but your paper as a whole would be a stronger contribution if you put your results in the context of the varying hydrogeology represented in the literature.

**We are not saying that Sagehen should be modeled differently, or saying that its geology differs in ways that are important for these purposes (although we recognize that this may be your view).  We are instead saying that it is essential to recognize that in all catchments, rates of streamflow are linked to the volume of water stored in the near-stream aquifer, which in turn integrates inputs from snowmelt and removals from ET.  This conceptual picture is consistent with the phase relationships observed not only at Sagehen, but also at Independence Creek and the headwater Yosemite sites.**

Line 584: This is illustrated in Lundquist and Dettinger 2003, see Figure 5.

**We don't understand the basis for this statement.  Figure 5 of Lundquist and Dettinger (2003), reproduced above, shows that snowmelt and evapotranspiration cycles have different shapes.  That is different from the point made here, which is that these two cycles will destructively interfere in the stream, resulting in the daily cycle becoming weak and reversing phase as it shifts from being snowmelt-dominated to ET-dominated.**

Line 620: Also, Independence Creek has more granitic geology and less groundwater reserves. It makes sense in the hydrogeologic context that this would have a snowmelt-dominated signal longer.

**We don't understand the rationale behind this statement.  The snowmelt-dominated signal will last as long as snow is melting in the riparian corridor, and melting in sufficient amounts that the snowmelt signal dominates over the competing evapotranspiration signal.  Having more or less groundwater will not change this balance, but having more snow and melting it later certainly will.  And our Figures 13 and 14 indicate that this is indeed the case here; the snow-covered fraction at Upper Independence Creek remains higher, for longer, than at Sagehen Creek.**

Lines 847-850: Data do not appear to be available at this time. Please do check that everything is publicly available and clearly interpretable (with readme files, metadata, etc) before final acceptance of the publication.

**We intend to do this.**

---

## Author Response (AR2)

Overall comments:

I appreciate the authors taking the time to carefully consider all of my comments and to improve the paper. Following a couple very minor changes (see below), I recommend the paper for publication, and I don't need to see it again. Jessica Lundquist, jdlund@uw.edu

**Thanks for your comments.**

The literature review is much more respectful than the prior version and more clearly illustrates how this work differs from what has been presented before. Thank you. I really like your addition of the Figure 5 conceptual figure – I think it helps a lot. Personally, I wish that the trees looked a bit more like the evergreen trees actually found in your basin, but that's a stylistic issue that you can decide on. I also like that you've added clearer comparisons of Independence Creek and Sagehen Creek. This adds greatly to the paper.

**Thanks for the discussion that led to the inclusion of this figure. We will leave it with the highly stylized and generic trees, because we find it hard to draw reasonable-looking evergreens.**

I really like the new version of Figure 9 – this is much clearer.

**Actually this isn't new. It is identical to Figure 8 in the original submission.**

For discussion, this (your Fig. 9) is actually great insight that ties to together the patterns shown in Lundquist and Dettinger (2003) [which I attached to my prior review]. Figure 3 of that paper shows that the phase of the diurnal signal shifts towards later in the day between the period of peak snowmelt and the end of the summer season, and that delay decreases more gradually in wetter years. These empirical results (which I struggled to find an explanation for in 2003), fit well with your statements (~line 562) that tau changes, becoming larger (line 805) as the catchment dries out. If you agree, I think around line 810 and/or after line 900, you could add a line or two to highlight that these earlier empirical findings actually fit well with your concepts and explanations here, and thus, there may be an interesting avenue for future research on how changes in the diurnal cycle characteristics may let us better understand changing aquifer storage and response times.

**The shifts in peak timing shown in Lundquist and Dettinger (2003) are much larger – about 12 hours – than can be explained by aquifer delays alone (roughly 6 hours). In their case (and also because the stream gauge is far downstream of the major snowmelt sources), the shifts in peak timing must reflect more than just the aquifer processes shown in Figure 9. There must also be either a lengthening of the distance between the gauge and the snowmelt sources (as snowmelt moves higher in the basin), and/or a decrease in the kinematic wave velocity as streamflow becomes shallower. We have nonetheless added this reference to the previous references that deal with lags in peak timing and their interpretation (in terms of both aquifer lags and kinematic wave lags).**

Minor comments:

Line 23, add comma after "aquifer),"

**Done.**

Line 36, add comma after weakens

**This comma is not needed for clarity, and would probably be removed during copy-editing.**

Lines 475+, Note that I did not have time to check all of the equations. However, given the propensity for typos to sneak into these types of equations, I request that the authors double check them all before final publication.

**They have \*already\* been double-checked.**

Line 874: add comma after "individual location"

**Done.**

[revised manuscript text omitted]